# METTL3/MYCN cooperation drives neural crest differentiation and provides therapeutic vulnerability in neuroblastoma

Ketan Thombare [1,2,11], Roshan Vaid [1,2,11], Perla Pucci [3,11], Kristina Ihrmark Lundberg [4], Ritish Ayyalusamy [1,2], Mohammad Hassan Baig [5], Akram Mendez[1,2], Rebeca Burgos-Panadero [1,2], Stefanie Höppner[6,7,8], Christoph Bartenhagen[6,7,8], Daniel Sjövall[9], Aqsa Ali Rehan [1,2], Sagar Dattatraya Nale[10], Anna Djos[1], Tommy Martinsson[1], Pekka Jaako [9], Jae-June Dong[5], Per Kogner[4], John Inge Johnsen [4], Matthias Fischer [6,7,8], Suzanne D Turner[3] & Tanmoy Mondal [1,2✉]

## Abstract

Neuroblastoma (NB) is the most common extracranial childhood cancer, caused by the improper differentiation of developing trunk neural crest cells (tNCC) in the sympathetic nervous system. The $N^6$-methyladenosine (m⁶A) epitranscriptomic modification controls post-transcriptional gene expression but the mechanism by which the m⁶A methyltransferase complex METTL3/METTL14/WTAP is recruited to specific loci remains to be fully characterized. We explored whether the m⁶A epitranscriptome could fine-tune gene regulation in migrating/differentiating tNCC. We demonstrate that the m⁶A modification regulates the expression of *HOX* genes in tNCC, thereby contributing to their timely differentiation into sympathetic neurons. Furthermore, we show that posterior *HOX* genes are m⁶A modified in *MYCN*-amplified NB with reduced expression. In addition, we provide evidence that sustained over-expression of the MYCN oncogene in tNCC drives METTL3 recruitment to a specific subset of genes including posterior *HOX* genes creating an undifferentiated state. Moreover, METTL3 depletion/inhibition induces DNA damage and differentiation of MYCN overexpressing cells and increases vulnerability to chemotherapeutic drugs in *MYCN*-amplified patient-derived xenografts (PDX) in vivo, suggesting METTL3 inhibition could be a potential therapeutic approach for NB.

**Keywords** Neuroblastoma; MYCN; METTL3; m⁶A; *HOX* Genes
**Subject Categories** Cancer; Chromatin, Transcription & Genomics; Neuroscience

## Introduction

RNA modification (also known as epitranscriptomics) can control important steps in RNA biogenesis such as RNA stability and RNA transport. One of the most abundant modifications of cellular RNA is $N^6$-methyladenosine (m⁶A). m⁶A is deposited on cellular RNA co-transcriptionally by the enzyme complex METTL3/METTL14/WTAP/RBM15 (Huang et al, 2019). A role for m⁶A RNA modification has been shown in chromatin regulation, stress response, DNA damage repair, as well as in viral infection by regulating gene expression both at transcriptional and post-transcriptional levels (Akhtar et al, 2021; Fu and Zhuang, 2020; Vaid et al, 2023; Zhang et al, 2020). Active chromatin modification H3K36me3, RNA binding proteins such as RBFOX2, and transcription factors have been implicated in METTL3 recruitment (Barbieri et al, 2017; Bertero et al, 2018; Dou et al, 2023). Despite these studies, the mechanism that drives the locus-specific recruitment of METTL3 remains unclear.

Neuroblastoma (NB) is a heterogeneous disease, it can spontaneously regress but in cases of high-risk disease, even with intensive therapy, disease relapse is frequently observed. As such, a better understanding of this disease and novel therapeutic strategies are urgently required. Amplification of the MYCN oncogene is one of the major genetic alterations found in NB and correlates with poor survival (Ackermann et al, 2018; Huang and Weiss, 2013).

[1]Department of Laboratory Medicine, Institute of Biomedicine, University of Gothenburg, Gothenburg, Sweden. [2]Region Västra Götaland, Sahlgrenska University Hospital, Department of Clinical Chemistry, Gothenburg, Sweden. [3]Division of Cellular and Molecular Pathology, Department of Pathology, University of Cambridge, Cambridge, UK. [4]Childhood Cancer Research Unit, Department of Women's and Children's Health, Karolinska Institute, and Pediatric Oncology, Astrid Lindgren Children's Hospital, Karolinska University Hospital, Stockholm, Sweden. [5]Department of Family Medicine, Yonsei University College of Medicine, Gangnam Severance Hospital, 211 Eonju-Ro, Gangnam-Gu, Seoul 06273, Republic of Korea. [6]Department of Experimental Pediatric Oncology, University Children's Hospital of Cologne, Medical Faculty, Cologne, Germany. [7]Center for Molecular Medicine Cologne (CMMC), University of Cologne, Cologne, Germany. [8]Department of Pediatric Oncology and Hematology, University of Cologne, Cologne, Germany. [9]Sahlgrenska Center for Cancer Research, Department of Microbiology and Immunology, Institute of Biomedicine, Sahlgrenska Academy, University of Gothenburg, SE-40530 Gothenburg, Sweden. [10]BNJ Biopharma, Memorial Hall, 85, Songdogwahak-ro, Yeonsu-gu, Incheon 21983, Republic of Korea. [11]These authors contributed equally: Ketan Thombare, Roshan Vaid, Perla Pucci. ✉E-mail: tanmoy.mondal@gu.se

MYCN amplification creates an undifferentiated state in NB (Huang and Weiss, 2013; Weiss et al, 1997), but detailed molecular mechanisms are lacking particularly as to how deregulation of MYCN creates an undifferentiated state in early developing human trunk neural crest cells (tNCC). MYCN overexpression has been shown to induce the transformation of neural crest cells into NB cells in humanized mouse models (Cohen et al, 2020; Saldana-Guerrero et al, 2024; Weng et al, 2022). Recently tNCC were derived from human embryonic stem cells (hESC) in vitro and these tNCC can be driven further to sympathoadrenal progenitors (SAP) and sympathetic neurons (SN) (Frith et al, 2018; Frith and Tsakiridis, 2019). Whether m⁶A modification has any role in this differentiation process has not been investigated.

METTL3 has been shown to have a tumor-promoting role in many cancers (Deng et al, 2018) and its inhibition through small molecules has recently been proposed as a therapeutic strategy for acute myeloid leukemia (AML) (Yankova et al, 2021). A role for METTL3-mediated m⁶A modification was recently reported in Alternative Lengthening of Telomeres-positive (ALT + ) NB (Vaid et al, 2024) but its function is unknown in other types of high-risk NB. High expression levels of METTL3 are predictive of an inferior outcome for NB patients and METTL3 is expressed in both ALT+ and *MYCN*-amplified (MNA) high-risk NB tumors, suggesting it may have broader relevance (Vaid et al, 2024).

In this study, we show that the expression level of METTL3 is higher in tNCC compared to hESC, correlating with an increase in overall m⁶A peaks in tNCC. We also found that METTL3 regulates the timely differentiation of tNCC by regulating *HOX* gene expression. We observed that MYCN overexpression can lead to an undifferentiated state in tNCC by downregulating posterior *HOX* gene expression. This MYCN-mediated undifferentiated state can be reversed by METTL3 depletion suggesting that METTL3 inhibition could be a novel therapeutic option for high-risk NB.

## Results

### METTL3 regulates posterior *HOX* genes expression during differentiation of tNCC

We have established a protocol for in vitro differentiation of hESC to tNCC adapting the previously described methodology (Frith et al, 2018) (Fig. 1A). The tNCC can be further differentiated into sympathoadrenal progenitors (SAP) and then into sympathetic neurons (SN). To confirm differentiation, using immunofluorescence (IF), robust expression of HOXC9, a posterior *HOX* gene was seen in tNCC whereas SAP cells expressed PHOX2B, and the differentiated SN were positive for peripherin (PRPH) (Fig. 1B). At the RNA level the pluripotency markers, *OCT4* and *NANOG* decreased during differentiation whereas *NGFR* and *SOX10* were upregulated at the tNCC stage (Appendix Fig. S1A). In addition, SAP showed upregulation of *ASCL1* and *ISL1,* whereas typical SN markers (*DBH* and *TH*) were upregulated from the SAP-stage of differentiation onwards with expression maintained in SN (Appendix Fig. S1A). Global gene expression changes comparing those of hESC and tNCC by RNA-seq showed that differentially expressed genes (DEGs) are enriched with pathways related to anterior-posterior pattern formation, epithelial to mesenchymal transition (EMT), and neural crest differentiation (Appendix Fig. S1B). Upregulation of *HOX* genes could also be

detected in tNCC using RNA-seq data (Appendix Fig. S1C). Overall, these data suggest that the differentiation of hESC to tNCC, SAP, and SN had been achieved in our model system.

Having established cellular differentiation, we sought to examine expression levels of m⁶A writer complex member proteins METTL3, METTL14, RBM15 and WTAP which peaked at the tNCC stage and then gradually decreased as the cells transitioned through the SAP to SN-stage (Fig. 1C). However, the regulation of expression of main m⁶A writer proteins METTL3 and METTL14 at the hESC to tNCC stage is likely regulated post-transcriptionally as RNA levels of *METTL3* and *METTL14* were unchanged (Appendix Fig. S1D). Indeed, METTL3 protein was more stable at the tNCC stage as confirmed by a cycloheximide chase experiment conducted with cells at both the hESC and tNCC stages of differentiation (Appendix Fig. S1E).

Next, we characterized the pattern of m⁶A modifications in hESC and tNCC using m⁶A RIP-seq (Vaid et al, 2024). We observed that consistent with the upregulation of m⁶A writer complex proteins, a higher number of m⁶A peaks was seen in tNCC compared to hESC (Fig. 1D), and these were enriched with DRACH-like motifs (Fig. 1E). tNCC showed higher relative m⁶A peak density in comparison with hESC and m⁶A peaks were enriched at the 3' UTR (Fig. 1F). Analysis of the m⁶A peaks in hESC and tNCC suggests both commonality and cell type-specific nature of m⁶A-positive genes (Fig. 1G).

The top enriched terms of the genes associated with m⁶A peaks from both hESC and tNCC showed pathways related to RNA splicing consistent with the role of m⁶A modification in RNA metabolism (Appendix Fig. S1F). Furthermore, we observed that the genes that were modified by m⁶A in tNCC and differentially expressed between hESC and tNCC were enriched in pathways related to anterior-posterior pattern formation, axonogenesis, and EMT (Fig. 1H). Overall, higher expression of METTL3/14, differential gene expression along with m⁶A RIP-seq data in hESC and tNCC suggest that m⁶A may have a role in the tNCC differentiation.

To validate the role of m⁶A modification in neural crest cell differentiation, we used an alternative previously described protocol to generate multipotent neural crest stem cells (NCSC) from hESC (Menendez et al, 2013). The NCSC identity of these cells was validated by robust expression of neural crest lineage markers (Appendix Fig. S1G). During the NCSC differentiation protocol, the cells were harvested at different stages of differentiation (days 3, 7, and 14). The expression of METTL3, METTL14, RBM15, and WTAP was upregulated during NCSC differentiation compared to hESC. (Appendix Fig. S1H). Consistent with the increase in the METTL3 level at day 3 NCSC progenitors, the stability of the METTL3 protein was higher compared to hESC (Appendix Fig. S1E). We also performed m⁶A RIP-seq of day 7 NCSC progenitors and NCSC at day 14 and detected 10,723 and 7250 m⁶A peaks, respectively, suggesting a role of m⁶A modification during NCSC differentiation (Appendix Fig. S1I).

To further characterize the role of METTL3-mediated m⁶A modification in tNCC, we generated METTL3 knockdown (KD) hESC which were then differentiated to tNCC or NCSC (Fig. 1I; Appendix Fig. S1J). Gene expression analysis of the METTL3 KD tNCC and day 7 NCSC showed robust upregulation of *HOX* genes (Appendix Fig. S1K). Furthermore, the genes that were deregulated following METTL3 KD and had m⁶A peaks in tNCC were associated with pathways related to anterior–posterior pattern

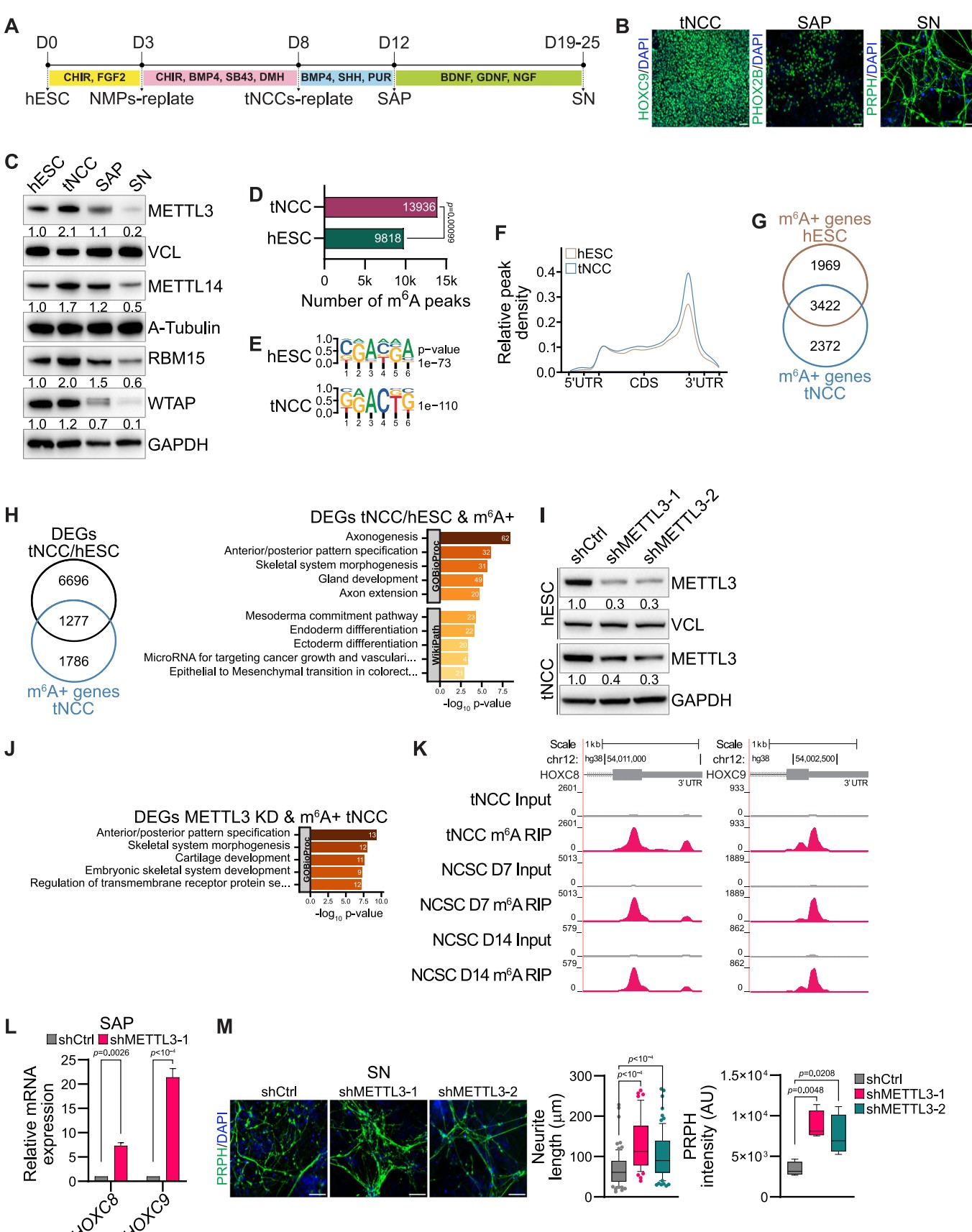

**Figure 1. METTL3 regulates posterior *HOX* genes expression during differentiation of tNCC.**

(A) Schematic diagram showing key steps involved in the differentiation process of human embryonic stem cells (hESC) into trunk neural crest cells (tNCC), followed by their further differentiation into sympathoadrenal progenitors (SAP) and, ultimately, into sympathetic neurons (SN). (B) Representative immunofluorescence (IF) images illustrating the expression of distinct lineage markers at different stages of differentiation: HOXC9 for tNCC, PHOX2B for SAP, and PRPH for SN. The scale bar is indicative of 50 μm. (C) Representative immunoblot shows the levels of METTL3, METTL14 RBM15, and WTAP across various stages of differentiation, including hESC, tNCC, SAP, and SN. Vinculin, A-tubulin, and GAPDH were loading controls. The values below the blots indicate the fold change (normalized to loading control) in the levels of METTL3, METTL14, RBM15, and WTAP. The experiments were repeated three times. (D) The total number of m⁶A peaks in hESC and tNCC, the p value was calculated using a permutation test, and the number of permutations was set to 1000. (E) Identified motifs from de novo motif analysis of m⁶A peaks enriched in hESC and tNCC and P values were obtained using the HOMER tool. (F) Metagene analysis showing relative m⁶A peak density at genes in hESC and tNCC. (G) Venn diagram showing overlap of the m⁶A positive (m⁶A +) [containing at least one m⁶A peak] genes in hESC and tNCC. (H) Left: Venn diagram comparison of differentially expressed genes (DEGs) [hESC vs. tNCC] and m⁶A +. Right: Top enriched terms associated with m⁶A-containing DEGs (hESC vs. tNCC) were identified using enrichGO, with p values obtained through Fisher's exact test. (I) Representative immunoblot shows the levels of METTL3, in hESC and tNCC with control (shCtrl) or stable METTL3 KD (shMETTL3-1, shMETTL3-2). Vinculin and GAPDH were loading control. The values below the blots indicate the fold change (normalized to loading control) in the levels of METTL3. The experiments were repeated three times. (J) Top enriched terms associated with DEGs (shCtrl vs. shMETTL3-1) that are m⁶A+ in tNCC were identified using enrichGO, with P values obtained through Fisher's exact test. (K) Genome browser screenshots of *HOXC8* and *HOXC9* 3′UTR, showing the presence of m⁶A in tNCC, neural crest stem cells (NCSC) at day 7 and at day 14. (L) RT-qPCR data showing the relative expression of *HOXC8* and *HOXC9* in SAP following METTL3 KD. *GAPDH* was used to normalize the qPCR data. Data are shown as mean ± SEM of three independent biological replicates. Two-way ANOVA with Šídák's multiple comparisons test was used. (M) shCtrl, shMETTL3-1, and shMETTL3-2 hESC were differentiated to SN, followed by IF with PRPH antibody to assess neurite length and PRPH signal intensity. Data are represented by box-whisker plots where the median is indicated by a horizontal line, the boxes represent the 25th to 75th percentiles, the whiskers show the 10th to 90th percentiles, and any outliers beyond this range are displayed as individual dots. This analysis was conducted across three independent biological replicates and statistical significance was determined using one-way ANOVA with Dunnett's multiple comparisons test. Scale bar represents 100 μm. Source data are available online for this figure.

specification (Fig. 1J). In particular, the posterior *HOX* genes *HOXC8* and *HOXC9* were enriched with m⁶A in tNCC, day 7 NCSC progenitors, and day 14 NCSC (Fig. 1K). RNA-seq data showed METTL3 KD resulted in upregulation of *HOXC8* and *HOXC9* in day 7 NCSC progenitors and *HOXC8* in tNCC (Appendix Fig. S1K). In addition, METTL3 KD resulted in upregulation of both, *HOXC8* and *HOXC9* as detected by RT-qPCR at SAP, suggesting m⁶A-dependent regulation of these genes (Fig. 1L). METTL3 KD SAP showed deregulation in the expression of several SAP markers such as *PHOX2B*, *ASCL1*, *ISL1*, and *GATA2*, suggesting METTL3 KD created a change in the differentiation potential of these cells (Appendix Fig. S1L). On further differentiation to the SN-stage, METTL3 KD promoted higher differentiation to SN as visualized by an increase in neurite length, and PRPH intensity (Fig. 1M).

In addition, conditional KD of METTL3 using the TetO system (Dox-induced METTL3 KD from day 5 onwards) specifically at the tNCC also led to enhanced differentiation at the SN-stage as visualized by increased PRPH intensity (Appendix Fig. S1M,N). These data suggest that the differentiation phenotype we observed is not due to METTL3 KD at the hESC stage but rather due to a reduced level of METTL3 during the differentiation of tNCC. Conditional METTL3 KD at the tNCC stage also led to the upregulation of HOXC8 and HOXC9 expression in the differentiated SAP (Appendix Fig. S1O). These data suggest that METTL3 plays an important role in regulating the timely transition of tNCC to SAP through regulating the expression of posterior *HOX* genes such as *HOXC8* and *HOXC9* via m⁶A modification.

## METTL3-mediated m⁶A modification controls HOXC8 and HOXC9 expression in *MYCN*-amplified NB tumors and in NB cell lines

Using RNA-seq data from 498 NB tumor samples, we observed that low expression of *HOXC8* and *HOXC9* correlates with poor survival, and this is consistent with an earlier report (Fig. 2A) (Kocak et al, 2013). We also observed that *HOXC8* and *HOXC9* expression was downregulated in *MYCN*-amplified (MNA)

compared to *MYCN* non-amplified (non-MNA) tumors (Fig. 2B). In line with that finding, HOXC9 protein levels were reduced in MNA tumors compared to non-MNA in the available public dataset (Hartlieb et al, 2021) and our validation cohort (Fig. 2C). To explore this further, we performed m⁶A RIP-seq of RNA derived from MNA NB tumors and observed m⁶A modification in *HOXC8* and *HOXC9* transcripts (Fig. 2D). The m⁶A peaks in the NB tumors were enriched at the stop codon and 3′UTR regions and with DRACH-like motifs as reported earlier (Meyer et al, 2012) (Appendix Fig. S2A). The common m⁶A enriched genes detected in MNA tumors were related to pathways such as axonogenesis and dendrite development/morphogenesis (Fig. 2E). These data were further strengthened by analyzing m⁶A RIP-seq of MNA SK-N-BE(2) cells, where *HOXC8*/*HOXC9* were likewise modified by m⁶A (Fig. 2F). Next, we examined whether the depletion of METTL3 in SK-N-BE(2) cells leads to the upregulation of *HOXC8* and *HOXC9*. Stable shRNA-mediated depletion of METTL3 was not possible with repeated attempts as METTL3-depleted cells did not survive, suggesting METTL3 was essential for the survival of SK-N-BE(2) cells. Hence, a doxycycline (Dox) inducible TetO shRNA system was employed to deplete METTL3 (TetO shM3) (Appendix Fig. S2D), and cells were analyzed by RNA-seq. These data showed that posterior *HOXC* locus genes such as *HOXC8*, *HOXC9*, and *HOXC10* were upregulated and m⁶A modified (Fig. 2F,G). We observed a moderate increase in the stability of *HOXC8* and *HOXC9* mRNA following induction of METTL3 KD by Dox addition for both 3 and 6 days in SK-N-BE(2) cells (Fig. 2H). Globally m⁶A positive genes tend to show upregulation at the RNA level following METTL3 depletion (Appendix Fig. S2B) and this observation is consistent with earlier reports on the role of m⁶A modification in RNA stability (Wang et al, 2014). METTL3-mediated m⁶A modification has been implicated in RNA translation (Shan et al, 2023). To investigate if METTL3 depletion results in a change in the translation efficiency of *HOXC8* and *HOXC9* genes, we checked their enrichment in polysome bound fraction following polysome profiling in METTL3 KD cells. Polysome enrichment of *HOXC8* and *HOXC9* mRNA were unchanged in METTL3 KD, suggesting that the translation efficiency of these

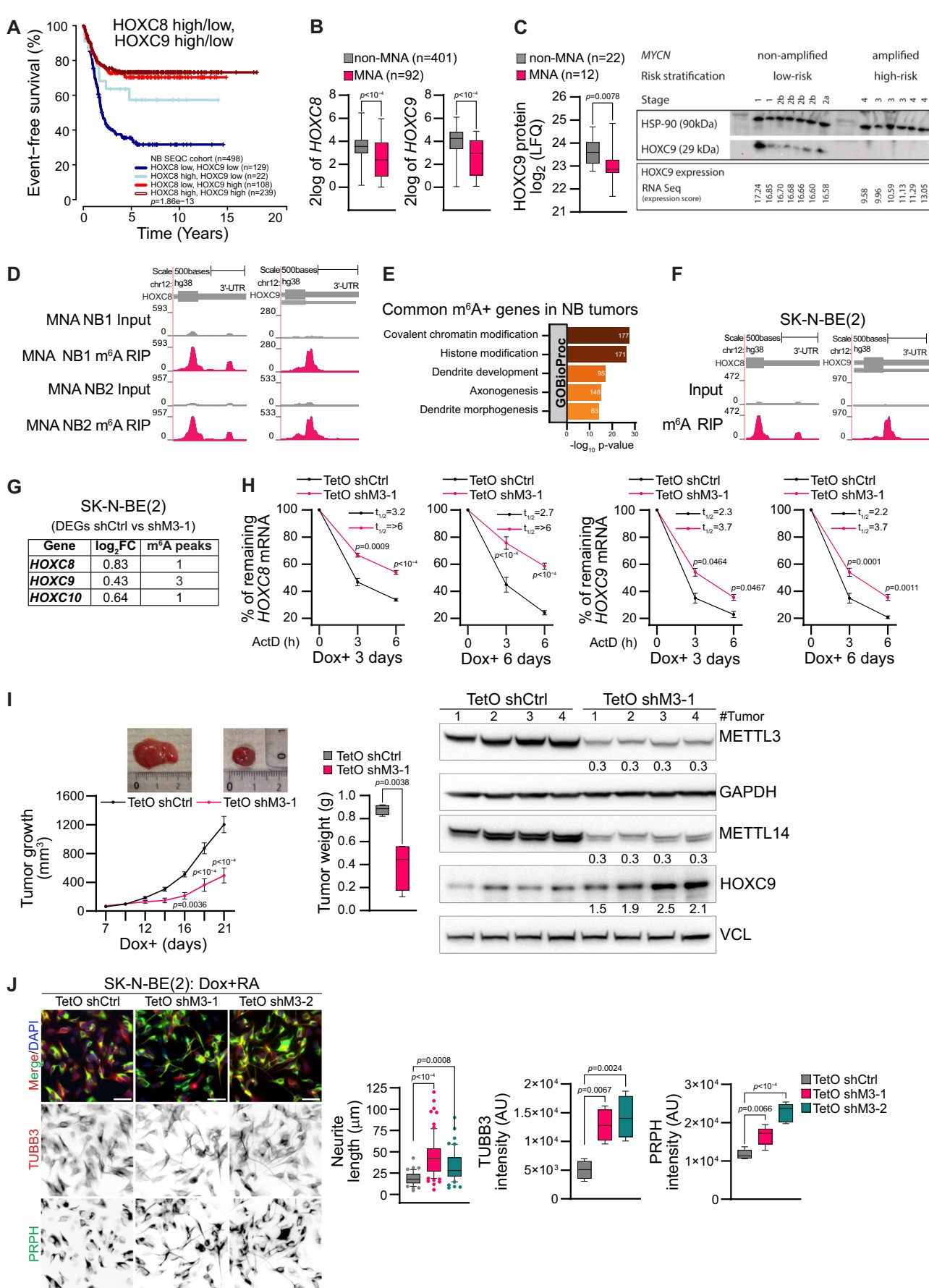

**Figure 2.  METTL3-mediated m⁶A controls *HOXC8/HOXC9* in *MYCN*-amplified NB.**

(A) Kaplan–Meier plot illustrates event-free survival in neuroblastoma (NB) patients ($n = 498$, SEQC cohort) with either low or high expression of *HOXC8* and *HOXC9*. Statistical analysis of survival was performed with a log-rank test. (B) Box-whisker plots show *HOXC8* and *HOXC9* expression in NB patients from the SEQC cohort, classified based on *MYCN* amplification status (non-MNA: non-*MYCN*-amplified, $n = 401$; MNA: *MYCN*-amplified, $n = 92$). The centerlines of the boxes represent the medians, the boxes extend from the 25th to 75th percentiles, and the whiskers depict the minimum and maximum values. Statistical analysis was performed using a two-tailed unpaired $t$ test. (C) Left: Box-whisker plots show HOXC9 protein levels in non-MNA ($n = 22$) and MNA ($n = 12$) NB patients (Hartlieb et al, 2021). The centerlines of the boxes represent the medians, the boxes extend from the 25th to 75th percentiles, and the whiskers depict the minimum and maximum values. Statistical significance was determined using a two-sided Mann-Whitney test. Right: Immunoblot shows the levels of HOXC9 in NB patient samples. *MYCN* status, risk stratification, stage, and *HOXC9* expression levels determined by RNA sequencing (expression score) are also provided. HSP90 was used as a loading control. The experiments were repeated three times. (D) Browser screenshot of m⁶A RIP-seq tracks at 3′UTR of *HOXC8* and *HOXC9* genes in MNA NB tumors. (E) Top enriched terms associated with m⁶A+ genes in both MNA NB tumors were identified using enrichGO, with $P$ values obtained through Fisher's exact test. (F) Genome browser screenshot showing the presence of m⁶A enrichment at 3′UTR of *HOXC8* and *HOXC9* genes in MNA NB cell line, SK-N-BE(2). (G) Differentially expressed posterior *HOXC* genes between control and METTL3 KD SK-N-BE(2) cells, and the number of m⁶A peaks identified using MACS peak caller in these genes are indicated. (H) Stability of *HOXC8* and *HOXC9* transcripts detected by RT-qPCR following Actinomycin D (10 μg/ml) mediated transcription blocking for the time points indicated in control (TetO shCtrl) and METTL3 KD (TetO shM3-1) SK-N-BE(2). Assay was conducted following 3 and 6 days of doxycycline (Dox) addition. Line plots present the quantification of remaining levels of *HOXC8* and *HOXC9* transcript at the indicated time points. Half-life ($t_{1/2}$) values are also denoted. Experiments were performed in three independent biological replicates. Data are presented as mean ± SEM. Two-way ANOVA with Šídák's multiple comparisons test was employed. (I) Left: Line plots showing tumor volume in control (TetO shCtrl) and METTL3 KD (TetO shM3-1) SK-N-BE(2) mouse xenograft with representative tumors from each group ($n = 4$ mice per group). Data are presented as mean ± SEM. Middle: Box-whisker plots show tumor weight in control and METTL3 KD xenograft tumors. The median is indicated by a horizontal line, the boxes represent the 25th to 75th percentiles, the whiskers show the 10th to 90th percentiles, and any outliers beyond this range are displayed as individual dots. Two-way ANOVA with Šídák's multiple comparisons test was employed to compare tumor volumes and two-tailed unpaired $t$ test for tumor weights. Right: Immunoblot showing expression of METTL3, METTL14, and HOXC9 in control and METTL3 KD xenografted tumors. GAPDH and vinculin were used as loading controls. The values below indicate the fold change (normalized to loading control) in the individual METTL3 KD xenografts compared to the mean expression of the control xenografts for METTL3, METTL14, and HOXC9. The experiments were repeated three times. (J) Representative IF showing PRPH (green), TUBB3 (red) staining in control (TetO shCtrl) and METTL3 KD (TetO shM3-1, TetO shM3-2) SK-N-BE(2) cells were pretreated with Dox for 1 day followed by 3 days of Dox and retinoic acid (RA) mediated differentiation. Box-whisker plots show the quantification of the neurite length, TUBB3, and PRPH intensity. The median is indicated by a horizontal line, the boxes represent the 25th to 75th percentiles, the whiskers show the 10th to 90th percentiles, and any outliers beyond this range are displayed as individual dots. Scale bar represents 50 μm. Experiments were performed in three independent biological replicates and statistical significance was determined using one-way ANOVA with Dunnett's multiple comparisons test. Source data are available online for this figure.

transcripts was unaffected (Appendix Fig. S2C). Furthermore, Dox-induced METTL3 KD in SK-N-BE(2) and IMR-32 (both MNA NB cell lines) led to reduced proliferation in these cells (Appendix Fig. 2D). To further verify the effects of METTL3 KD in combination with MYCN overexpression we have used SHEP cells (low MYCN expressing NB cells), using Dox inducible TetO system (SHEP^MYCN). We observed that METTL3 KD in combination with MYCN overexpression showed a decrease in cell viability, whereas METTL3 KD alone in SHEP cells had little effect on cell viability (Appendix Fig. S2E,F). Moreover, injection of Dox inducible METTL3 KD SK-N-BE(2) cells into immunocompromised nude mice administered with doxycycline, led to reduced xenografted tumor growth (Fig. 2I). We validated METTL3 KD in xenografted SK-N-BE(2) tumors with consequent upregulation in HOXC9 expression (Fig. 2I). Interestingly, METTL3 KD also resulted in downregulation of METTL14 expression (Fig. 2I). We verified a decrease in METTL14 expression by siRNA mediated KD of METTL3 in SK-N-BE(2) cells (Appendix Fig. S2G). This observation is consistent with a recent report that suggested METTL3 protects METTL14 by preventing its ubiquitination and degradation (Zeng et al, 2023). METTL3 KD also reduced colony formation in SK-N-BE(2), in line with our in vivo findings (Appendix Fig. S2H; Fig. 2I). The genes that were deregulated after METTL3 KD and m⁶A positive in SK-N-BE(2) cells were enriched in pathways related to axonogenesis suggesting m⁶A dependent role of METTL3 in the differentiation of NB cells (Appendix Fig. S2I). To explore this further, we performed retinoic acid (RA) mediated differentiation of control and Dox inducible METTL3 KD SK-N-BE(2) cells. We observed that METTL3 KD could promote differentiation of SK-N-BE(2) cells as visualized by increased neurite length and TUBB3/PRPH intensities (Fig. 2J).

## MYCN overexpression creates an undifferentiated state in tNCC

Interestingly, MYCN expression is high in hESC and tNCC but is downregulated as differentiation progresses to SAP, and the expression becomes almost undetectable in SN cells (Fig. 3A). Given that MYCN is amplified in a subset of high-risk NB, our tNCC to SN differentiation model allows us to explore how MYCN deregulation may contribute to the improper differentiation characteristic of NB. To enforce MYCN expression throughout differentiation, we created a Dox inducible expression system by introducing MYCN into hESC using the inducible PiggyBac system (Randolph et al, 2017). To induce the MYCN overexpression Dox was added from day 5 of differentiation and continued until the end of the experiment (Fig. 3B). We validated the overexpression of MYCN following induction with Dox at the tNCC, SAP, and SN stages of differentiation by immunoblot (Fig. 3C). To determine whether MYCN overexpression affects differentiation towards the SN-stage, we harvested cells at day 22 when the SN phenotype is normally observed and noted that the cells forcibly expressing MYCN lacked expression of PRPH and TUBB3 as detected by IF (Fig. 3D,E). We wanted to check if HOXC8 and HOXC9 expression was altered following MYCN overexpression, and we observed HOXC8 and HOXC9 were downregulated in MYCN overexpressing SAP cells (Fig. 3F,G; Appendix Fig. S3A). Downregulation of *HOX* genes, such as *HOXC8* and *HOXC9*, was detected in MYCN overexpression SAP cells using RNA-seq and RT-qPCR (Fig. 3H). Consistent with the downregulation, the stability of *HOXC8* and *HOXC9* transcripts was also reduced in MYCN overexpressed SAP cells (Fig. 3I). MYCN overexpression in SAP also resulted in the increased level of m⁶A methylation in *HOXC8* and *HOXC9*

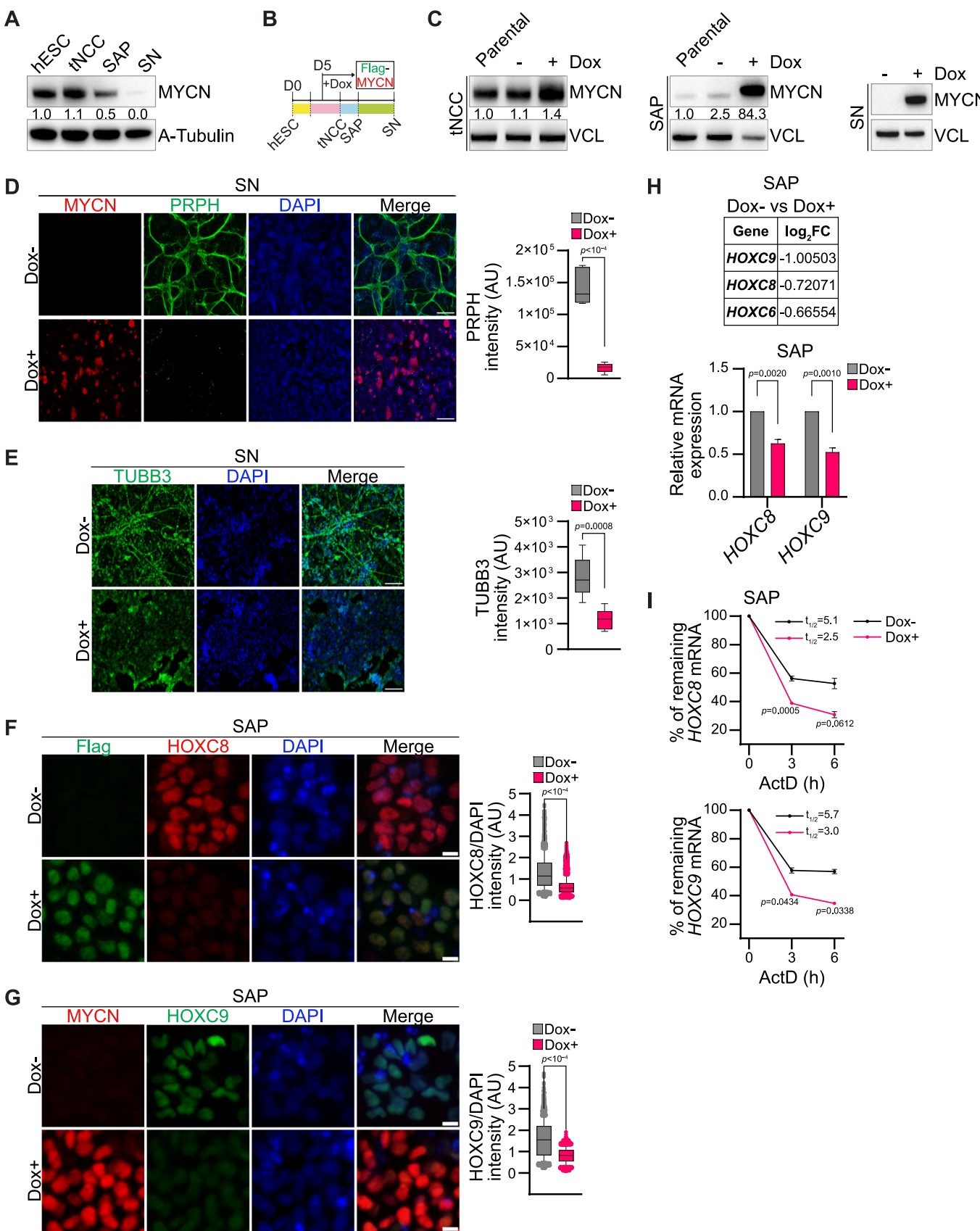

**Figure 3.   MYCN overexpression creates an undifferentiated state in tNCC.**

(**A**) Representative immunoblot showing expression of MYCN at hESC, tNCC, SAP, and SN stages of differentiation. A-tubulin was used as a loading control. The values below indicate the fold change in levels of MYCN. The experiments were repeated three times. (**B**) Schematic diagram showing the time of FLAG-tagged MYCN (Flag-MYCN) induction in a Dox-dependent manner from day 5 during tNCC differentiation. (**C**) Immunoblot showing MYCN overexpression in tNCC (left), SAP (middle), and SN (right), following Dox-induced Flag-MYCN from day 5 of differentiation. Vinculin was used as a loading control. The values below indicate the fold change in levels of MYCN. The experiments were repeated three times. (**D**) Control (Dox-), and Flag-MYCN overexpressed (Dox + , from day 5 onwards) tNCC were differentiated to SN and IF was performed with PRPH (green) and MYCN (red) antibodies. Box-whisker plots show PRPH signal intensity. The median is indicated by a horizontal line, the boxes represent the 25th to 75th percentiles, the whiskers show the 10th to 90th percentiles, and any outliers beyond this range are displayed as individual dots. Data are from three independent experiments and statistical analysis was performed using a two-tailed unpaired $t$ test. Scale bar represents 500 μm. (**E**) Control (Dox-), and Flag-MYCN overexpressed (Dox + , from day 5 onwards) tNCC were differentiated to SN and IF was performed with TUBB3 (green) antibody. Box-whisker plots show TUBB3 intensity. The median is indicated by a horizontal line, the boxes represent the 25th to 75th percentiles, the whiskers show the 10th to 90th percentiles, and any outliers beyond this range are displayed as individual dots. Data are from three independent experiments and statistical analysis was performed using a two-tailed unpaired $t$ test. Scale bar represents 100 μm. (**F**) Representative IF showing expression of HOXC8 (red), and MYCN (green) visualized with an anti-FLAG antibody in control (Dox-) and Flag-MYCN overexpressed (Dox + , from day 5 onwards) SAP-stage cells. Box-whisker plots show HOXC8 signal intensity normalized to DAPI intensity. The median is indicated by a horizontal line, the boxes represent the $25^{th}$ to $75^{th}$ percentiles, the whiskers show the $10^{th}$ to $90^{th}$ percentiles, and any outliers beyond this range are displayed as individual dots. Signal intensity measurements were taken from over 3900 cells. Data are from three independent experiments, and statistical analysis was performed using a two-tailed unpaired $t$ test. Scale bar represents 10 μm. (**G**) Representative IF showing expression of HOXC9 (green), and MYCN (red) visualized with MYCN antibody in control (Dox-) and Flag-MYCN overexpressed (Dox + , from day 5 onwards) SAP-stage cells. Box-whisker plots show HOXC9 intensity normalized to DAPI intensity. The median is indicated by a horizontal line, the boxes represent the 25th to 75th percentiles, the whiskers show the 10th to 90th percentiles, and any outliers beyond this range are displayed as individual dots. Signal intensity measurements were taken from over 2650 cells. Data are from three independent experiments. Statistical analysis was performed using a two-tailed unpaired $t$ test. Scale bar represents 10 μm. (**H**) Top: Differentially expressed posterior *HOXC* genes between control (Dox−) and Flag-MYCN overexpression (Dox + , from day 5 onwards) in SAP. The expression values were determined from RNA-seq data. Bottom: Relative mRNA expression of *HOXC8* and *HOXC9* in SAP following Flag-MYCN overexpression (Dox + , from day 5 onwards) and in control (Dox-). *GAPDH* was used to normalize the qPCR data. Data are shown as mean ± SEM of three independent biological replicates. Two-way ANOVA with Šídák's multiple comparisons test was employed. (**I**) Stability of *HOXC8* and *HOXC9* transcripts detected by RT-qPCR after Actinomycin D (10 μg/ml) mediated transcription blocking for the time points indicated in MYCN overexpressed SAP. Line plots present the quantification of remaining levels of *HOXC8* and *HOXC9* transcript at the indicated time points and $t_{1/2}$ values are also denoted. Data are shown as mean ± SEM of three independent biological replicates. Two-way ANOVA with Šídák's multiple comparisons test was employed. Source data are available online for this figure.

transcripts (Appendix Fig. S3B). Along with *HOX* genes, we observed that several other transcription factors involved in SAP differentiation, such as *ASCL1* and *ISL1*, were also altered following MYCN overexpression in SAP as detected by RNA-seq (Appendix Fig. S3C). To further investigate transcriptional changes on MYCN overexpression at the SN-stage, we conducted RNA-seq as well. Consistent with the undifferentiated phenotype (Fig. 3D,E) we observed that downregulated genes from both MYCN over-expressed SAP and SN-stage cells were enriched in pathways related to axonogenesis, axon guidance, and neuronal projection guidance (Appendix Fig. S3D,E). In contrast, upregulated genes following MYCN overexpression in SAP were enriched with terms such as metabolic process and in SN were related to the RNA metabolic process (Appendix Fig. S3D,E), which is consistent with the earlier reported function of MYCN oncogene (Wang et al, 2023).

## MYCN cooperates with METTL3 to regulate m6A levels of HOXC8 and HOXC9

Our data show that the alteration in MYCN and METTL3 levels could regulate HOXC8 and HOXC9 expression during the differentiation of tNCC (Fig. 3F,G; Appendix S1O). This suggests that MYCN might cooperate with METTL3 to regulate m6A levels during tNCC differentiation and thereby gene expression. To explore this further, we performed Co-immunoprecipitation (Co-IP) with MYCN antibody in SHEP cells following Dox-induced MYCN overexpression and found METTL3 to be interacting with MYCN (Fig. 4A). This was also true vice versa, i.e., when METTL3 was immunoprecipitated and MYCN was immunoblotted (Fig. 4A). We further performed PLA and observed an interaction between MYCN and METTL3 in tNCC (Fig. 4A). In addition, in SHEP cells following Dox-induced MYCN overexpression an interaction between MYCN and METTL3 was also observed by PLA (Fig. 4A;

Appendix Fig. S2E). Furthermore, IF for MYCN and METTL3 in tNCC showed co-localization of these proteins in the cells (Appendix Fig. S4A). To check if the transcriptional elongation process has any contributing role in the MYCN and METTL3 interaction, we treated the MYCN overexpressed SHEP cells with Flavopiridol (FP). MYCN overexpression in SHEP cells resulted in a global increase in transcriptional output as visualized by labeling of the nascent RNA and this MYCN-mediated transcriptional effect could be effectively inhibited by FP treatment. We observed that treatment with FP did not alter MYCN and METTL3 interaction as detected by Co-IP in SHEP^MYCN cells suggesting the transcription elongation-independent nature of this interaction (Appendix Fig. S4B; Fig. 4B). The mechanisms by which METTL3 is recruited to chromatin remain largely unknown. Our data whereby MYCN and METTL3 interact (Fig. 4A; Appendix Fig. S4A) suggests that MYCN might guide METTL3 to distinct genomic regions. To explore this further, the ChIP-seq of both MYCN and METTL3 was conducted in hESC and tNCC. We observed an overlap between MYCN and METTL3 chromatin-bound regions genome-wide in both hESC and tNCC (Appendix Fig. S4C; Fig. 4C). The MYCN and METTL3 co-bound regions were enriched with active chromatin modification H3K27ac in tNCC and were located around gene promoters (Fig. 4D; Appendix Fig. S4D,E). The METTL3 and MYCN co-bound genes were enriched with pathways related to neuronal projection and axonogenesis (Appendix Fig. S4F). We observed that METTL3/MYCN co-bound gene promoters had higher enrichment of METTL3 compared to only METTL3-bound, suggesting that MYCN binding could facilitate METTL3 recruitment (Fig. 4E). Approximately 20% of MYCN and METTL3 promoter-bound genes in tNCC contained at least one m6A peak (Appendix Fig. S4G). Interestingly, the MYCN and METTL3 promoter-bound and m6A-modified genes which were deregulated upon METTL3 KD in tNCC were related to axon guidance

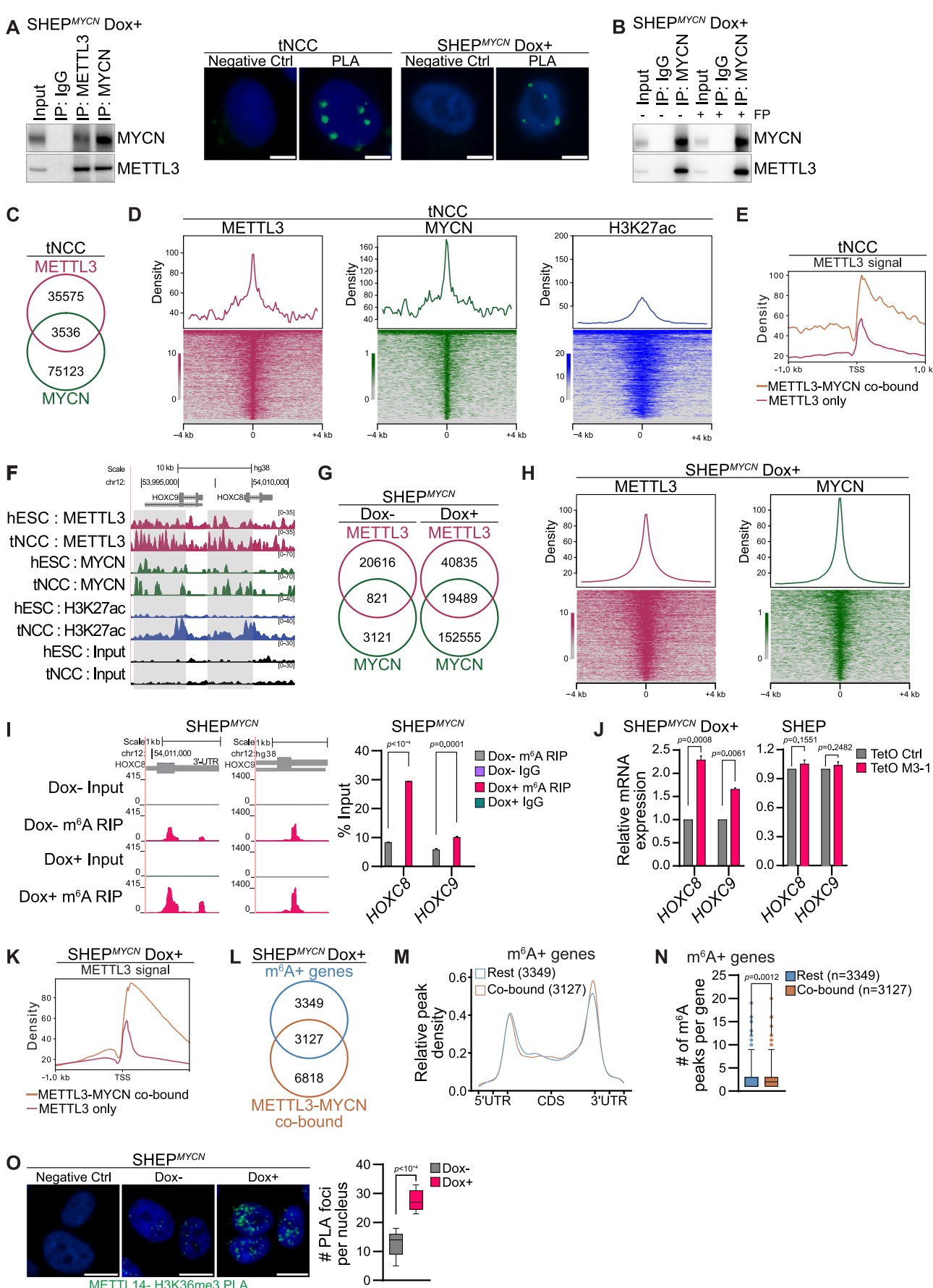

**Figure 4. MYCN cooperates with METTL3 to regulate m⁶A levels of *HOXC8/HOXC9*.**

(A) Left: Co-IP of METTL3 or MYCN from lysates of Flag-MYCN overexpressing SHEP cells (SHEP^MYCN^) after Dox induction for 24 h, blotted with MYCN or METTL3 antibodies. IgG served as a negative control. Right: Proximity ligation assay (PLA) in tNCC and 24 h Dox-induced SHEP^MYCN^ cells depicting METTL3 and MYCN PLA signal (green) in the nucleus (marked by DAPI). The negative control shows PLA with only the METTL3 antibody. Scale bar is 5 µm. The experiments were repeated three times. (B) Co-IP of METTL3 or MYCN from lysates of SHEP^MYCN^ after 24 h of Dox induction and 1 h treatment with 300 nM flavopiridol (FP), followed by western blotting with MYCN or METTL3 antibodies. IgG was used as a negative control. The experiments were repeated three times. (C) Venn diagram comparison of METTL3 and MYCN binding sites determined from ChIP-seq experiments performed in tNCC. (D) Distribution and heatmaps of normalized ChIP-seq reads for METTL3, MYCN, and H3K27ac over the MYCN and METTL3 overlapping peak coordinates. The data is centered on MYCN peaks (− 4 kb to +4 kb). (E) Distribution of METTL3 ChIP signal in a metagene profile. The data is centered at the transcription start site (TSS) [−1 kb to +1 kb], at genes that are co-bound by METTL3 and MYCN or bound by METTL3 only in tNCC. (F) Genome browser screenshot showing METTL3, MYCN, and H3K27ac ChIP-seq signals in hESC and tNCC over the *HOXC8* and *HOXC9* gene locus. (G) Venn diagram comparison of METTL3 and MYCN binding sites determined from ChIP-seq experiments performed in SHEP^MYCN^ cells before and after Dox induction for 24 h. (H) Distribution and heatmaps of normalized ChIP-seq reads for METTL3 and MYCN overlapping peaks centered on MYCN peaks (− 4 kb to +4 kb) in SHEP^MYCN^ cells after Dox induction. (I) Left: Browser screenshot showing m⁶A RIP-seq tracks at 3′UTR of *HOXC8* and *HOXC9* genes in SHEP^MYCN^ cells before and after Dox induction for 24 h. Right: m⁶A RIP-qPCR data showing enrichment of both *HOXC8* and *HOXC9* in SHEP^MYCN^ cells before and after Dox induction for 24 h. Data are represented as a percentage of input. IgG was used as a negative control. Data are from three independent experiments and shown as mean ± SEM. Two-way ANOVA with Tukey's multiple comparisons test was used. (J) RT-qPCR data showing the expression of *HOXC8* and *HOXC9* in SHEP^MYCN^ cells Left: SHEP cells Right: with either control (TetO shCtrl) or METTL3 KD (TetO shM3-1) after Dox induction for 6 days. *GAPDH* was used to normalize the qPCR data. Data are shown as mean ± SD of three independent biological replicates. Two-way ANOVA with Šídák's multiple comparisons test was used. (K) Left: Distribution of METTL3 ChIP signal in a metagene profile. The data are centered at the transcription start site (TSS) [−1 kb to +1 kb], at genes that are co-bound by METTL3 and MYCN or bound by METTL3 only in SHEP^MYCN^ cells after 24 h Dox induction. (L) Venn diagram comparing m⁶A+ and METTL3/MYCN co-bound genes SHEP^MYCN^ cells after Dox 24 h induction. METTL3 and MYCN co-bound regions were determined using the ChIP-seq experiments. (M) Metagene analysis showing relative m⁶A peak density at genes co-bound by METTL3 and MYCN or the rest of m⁶A-containing genes in SHEP^MYCN^ cells after Dox induction. (N) Box-whisker plots showing the number of m⁶A peaks/genes that are co-bound by METTL3 and MYCN (median = 2) or the rest of m⁶A-containing genes (median = 1) in SHEP^MYCN^ cells after Dox induction. The number of co-bound peaks and Rest peaks used for this analysis are 3127 and 3349, respectively. Whiskers indicate the 1st to 99th percentiles, and any outliers beyond this range are shown as individual dots. Statistical analysis was performed using the Wilcoxon matched-pairs signed rank test. (O) PLA in SHEP^MYCN^ cells with or without Dox induction for 24 h depicting METTL14 and H3K36me3 PLA signal (green) in the nucleus (marked by DAPI). The negative control shows PLA with only the H3K36me3 antibody. Signal intensity measurements were taken from over 50 cells. Data are from three independent experiments and presented as box-whisker plots where the median is indicated by a horizontal line, the boxes represent the 25th to 75th percentiles, the whiskers show the 10th to 90th percentiles and any outliers beyond this range are displayed as individual dots. Statistical analysis was performed using a two-tailed unpaired *t* test. Scale bar represents 10 µm. Source data are available online for this figure.

(Appendix Fig. S4G). Furthermore, higher enrichment of METTL3 and MYCN was seen over the *HOXC8* and *HOXC9* gene loci in tNCC compared to hESC, and these genes were also enriched with the active histone modification H3K27ac in tNCC but not in hESC, consistent with their expression at this stage of differentiation (Fig. 4F).

To determine whether MYCN expression can influence METTL3 binding and m⁶A modification, we utilized the Dox inducible SHEP^MYCN^ system (Appendix Fig. S2E). Mapping of METTL3 and MYCN binding before and after MYCN over-expression in SHEP cells showed an expected increase in MYCN binding genome-wide on Dox induction (Fig. 4G). In addition, a threefold increase in the number of METTL3 peaks was also observed in comparison with MYCN non-induced (Dox-) SHEP cells (Fig. 4G). MYCN and METTL3-binding sites frequently overlapped in Dox-induced SHEP^MYCN^ cells and again most of the overlapping peaks were associated with gene promoters (Fig. 4G,H; Appendix Fig. S4H). Furthermore, MYCN overexpression resulted in increased METTL3 recruitment to *HOXC8* and *HOXC9* genes (Appendix Fig. S4I). To check how MYCN overexpression can influence m⁶A methylation in SHEP cells we have performed m⁶A RIP-seq. This m⁶A RIP-seq data shows a higher level of m⁶A enrichment at *HOXC8* and *HOXC9* genes which correlates with increased METTL3 recruitment over these genes in MYCN overexpressed SHEP cells (Fig. 4I). Higher level of m⁶A enrichment at HOXC8 and HOXC9 genes was further validated by m⁶A RIP-qPCR (Fig. 4I). Consistent with this, MYCN overexpression combined with METTL3 KD resulted in increased level of *HOXC8* and *HOXC9* whereas only METTL3 KD had no significant effect on expression of these genes in SHEP cells (Fig. 4J). METTL3/MYCN co-bound gene promoters in MYCN overexpressed SHEP cells had

higher enrichment of METTL3 compared to only METTL3-bound promoters (Fig. 4K), and 30% of the MYCN/METTL3 co-bound genes were m⁶A positive (Fig. 4L). Metagene analysis further suggested that MYCN/METTL3 co-bound genes had a higher density of m⁶A peaks in the 3′ UTR compared to the rest of the m⁶A positive genes, and the number of m⁶A peaks per gene was higher in METTL3/MYCN co-bound genes (Fig. 4M,N). We also show that blocking transcriptional elongation by FP in MYCN overexpressed SHEP cells did not affect METTL3 and MYCN recruitment over *HOXC8/HOXC9* and the known MYCN target gene *NPM1* (Appendix Fig. S4J). This data is consistent with METTL3 and MYCN Co-IP data in the presence of FP, as FP treatment did not disrupt METTL3 and MYCN interaction (Appendix Fig. S4J; Fig. 4B). We next aimed to explore further how higher METTL3 recruitment in MYCN bound genes can drive m⁶A modification. The METTL3/METTL14 complex has previously been shown to be guided by H3K36me3 (Huang et al, 2019), a transcription elongation-specific chromatin mark, for co-transcriptional m⁶A deposition. We observed that MYCN over-expression in SHEP cells led to higher interaction between H3K36me3 and METTL14 (Fig. 4O). These data suggest that MYCN-mediated METTL3 recruitment could enhance the inter-action of the METTL3/METTL14 complex with H3K36me3 for co-transcriptional m⁶A deposition. METTL3 has recently been reported to regulate global transcription by upregulating MYCN expression in paused mouse ES cells (Collignon et al, 2023). However, we did not observe such an effect of METTL3 KD on MYCN expression levels in tNCC or NB cells (Appendix Fig. S4K), which is consistent with the earlier report (Hagemann et al, 2023). Altogether, our data suggest that MYCN could act as a guiding factor for METTL3 recruitment to drive m⁶A modification in

specific sets of developmental genes, including *HOXC8* and *HOXC9*.

## METTL3-mediated m⁶A modification of HOXC9 regulates differentiation of MNA NB

We observed transcription factor binding motifs such as HOXC9 and SOX10 (top 10 motifs) were enriched in promoters of genes deregulated following MYCN overexpression in SAP. (Fig. 5A; Appendix Fig. S3D). This indicates that the downregulation of HOXC9 in MYCN overexpressed SAP cells could contribute to the observed undifferentiated phenotype (Fig. 3D,E,G). To further validate, we overexpressed HOXC9 in MYCN overexpressed SAP cells, which led to the rescue of differentiation, as indicated by increased neurite length and elevated levels of TUBB3, and PRPH-positive SN (Fig. 5B; Appendix Fig. S5A). Next, we investigated the contribution of m⁶A modification specifically on the *HOXC9* transcript to the differentiation of MNA NB. To this end, we utilized the dCasRX-FTO system, where the m⁶A demethylase FTO can be recruited to specific mRNA using guide RNA (gRNA) (Vaid et al, 2024). We first validated that the two targeted gRNAs located close to *HOXC9* m⁶A peaks could deplete *HOXC9* mRNA levels, using catalytically active CasRx and RT-qPCR (Appendix Fig. S5B). Next, using these gRNAs we recruited either wild-type (WT) or catalytically dead (mutant) FTO fused with dCasRx (catalytically inactive CasRx) using these gRNAs. We observed recruitment of dCasRx-FTO^WT resulted in decreased m⁶A level over *HOXC9* transcript (Appendix Fig. S5C,D). We differentiated dCasRx-FTO^WT and dCasRx-FTO^Mutant cells expressing HOXC9 gRNAs using RA and observed that the recruitment of WT FTO on HOXC9 resulted in differentiation of SK-N-BE(2) cells but not recruitment of mutant FTO (Fig. 5C; Appendix Fig. S5C). Consistent with the differentiation phenotype, we also observed increased expression of HOXC9 only in WT FTO recruitment, suggesting that the level of m⁶A determines HOXC9 expression during RA-mediated differentiation (Fig. 5D). We wanted to know further if increased HOXC9 expression in METTL3 KD SK-N-BE(2) cells contributes to the observed enhanced differentiated phenotype (Fig. 2J). Increased HOXC9 level following METTL3 KD could be detected using immunoblot, and this upregulation could be reversed by using shRNA against HOXC9 (Fig. 5E). METTL3 KD induced differentiation of SK-N-BE(2) cells, as observed earlier, but simultaneous KD of HOXC9 in these cells drastically perturbed differentiation (Fig. 5F). We found that around 20% of HOXC9 target genes identified by ChIP-seq (Mao et al, 2011), overlapped with METTL3 deregulated genes in SK-N-BE(2) cells (Appendix Fig. S5E). These overlapping genes (such as *PRPH*, *SEMA3D*, and *NRCAM*) were enriched in pathways related to neuronal differentiation, further suggesting that HOXC9 upregulation in METTL3 KD cells contributes to the observed differentiation phenotype (Appendix Fig. S5E).

As both *HOXC8* and *HOXC9* were upregulated following METTL3 KD and were m⁶A positive, we investigated if these transcription factors together contribute to the METTL3 KD mediated differentiation phenotype. We performed network analysis, which predicted physical interaction between HOXC8 and HOXC9 (Appendix Fig. S5F). Using proximity ligation assay (PLA) we validated HOXC8 and HOXC9 interaction in MNA NB cells (Appendix Fig. S5F). Stable overexpression of both HOXC8

and HOXC9 induced spontaneous differentiation of SK-N-BE(2) cells without RA addition (Appendix Fig. S5G) (Kocak et al, 2013; Mao et al, 2011). We also observed a stronger RA-mediated differentiation phenotype after transient overexpression of combined HOXC8 and HOXC9, compared to a single *HOX* gene in SK-N-BE(2) cells (Appendix Fig. S5H). Using ChIP-qPCR we detected enrichment of both HOXC8 and HOXC9 at the selected HOXC9 targets genes which were deregulated following METTL3 KD (Appendix Fig. S5E,I). Together, these data suggest that deregulation of posterior *HOX* genes is critical in the observed differentiation phenotype in both METTL3-depleted and MYCN overexpressing cells.

## METTL3 inhibition restores differentiation and sensitizes NB cells to chemotherapeutic drug

Our data show that MYCN overexpression downregulates and METTL3 KD upregulates expression of the posterior *HOX* genes *HOXC8* and *HOXC9* suggesting a potential antagonistic regulation (Appendix Fig. S1L and S3B,C; Fig. 3F,G). We show that HOXC8 and HOXC9 overexpression promoted differentiation of the MNA SK-N-BE(2) cells (Appendix Fig. S5G,H). Furthermore, m⁶A removal using the dCasRx-FTO system was sufficient to induce differentiation of MNA SK-N-BE(2) cells (Fig. 5C). These data suggest that pharmacologic inhibition of METTL3 could promote differentiation of MYCN overexpressing tNCC. We therefore treated MYCN overexpressing tNCC with STM2457, a small molecule inhibitor of METTL3 (Yankova et al, 2021). HOXC8 and HOXC9 expression was restored in STM2457-treated MYCN overexpressing SAP-stage cells (Fig. 6A,B). Indeed, treatment of MYCN overexpressing tNCC with STM2457 rescued the differentiation phenotype of these cells, as observed by PRPH IF (Fig. 6C). Concurrently, we knocked down METTL3 (from day 5 onwards) in the MYCN overexpressing tNCC (Appendix Fig. S6A). As expected, METTL3 KD rescued the differentiation of MYCN overexpressing cells as evidenced by PRPH and TUBB3 expression (Appendix Fig. S6A). Hence, our data suggest that MYCN cooperates with METTL3 to create an undifferentiated state that can be reversed by METTL3 inhibition or KD. To further assess the effect of STM2457 on differentiation, MNA NB cells (SK-N-BE(2) and NGP) were treated with STM2457 in combination with RA, leading to an increase in differentiation (Fig. 6D; Appendix Fig. S6B). STM2457 treatment in SK-N-BE(2) cells resulted in the increased expression and stability of both *HOXC8* and *HOXC9* transcript, suggesting that they contribute to STM2457-mediated enhanced differentiation in these cells (Appendix Fig. S6C,D).

Given the low expression of *HOXC9* at the SN compared to tNCC and SAP (Appendix Fig. S6E), we explored whether factors beyond *HOX* genes contributed to the restoration of the differentiation phenotype observed following METTL3 inhibition in cells overexpressing MYCN (Fig. 6C). For this purpose, we conducted gene expression analysis on MYCN overexpressing SN-stage cells treated with either DMSO or STM2457. The gene expression profiles showed that genes related to DNA damage and repair were upregulated during differentiation following METTL3 inhibition (Appendix Fig. S6F). We also profiled gene expression of the METTL3 KD SK-N-BE(2) cells following RA-mediated differentiation for 5 days and we observed upregulation of DNA repair-related pathways in this system as well (Appendix Fig. S6G).

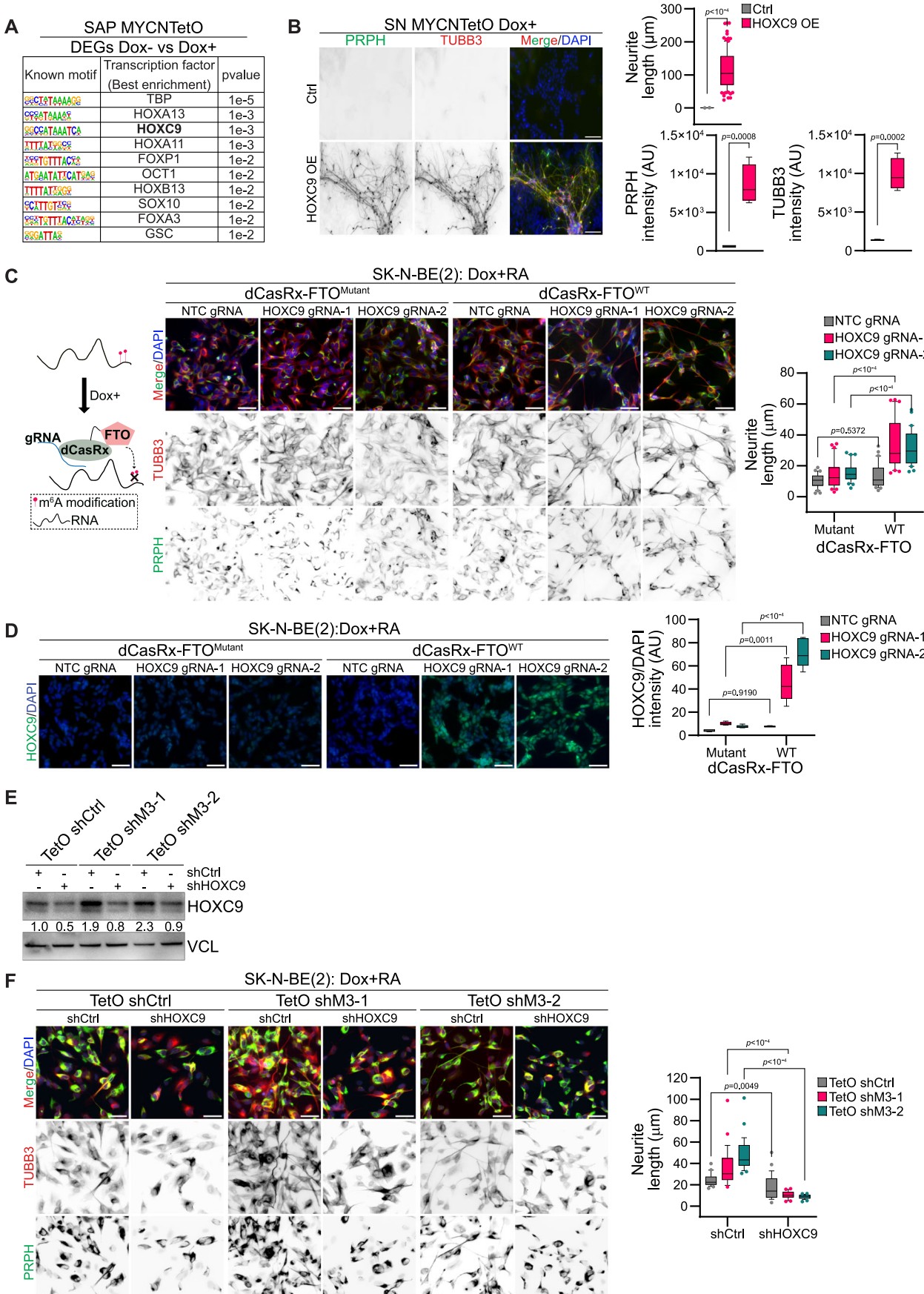

**Figure 5.    m⁶A modification of *HOXC9* regulates differentiation of MNA NB.**

(A) Top 10 transcription factor binding motifs enriched in the promoter region of the DEGs (Flag- MYCN overexpressed Dox- vs. Dox + ) in SAP. *P* values were obtained using the HOMER tool. (B) IF showing expression of PRPH (green) and TUBB3 (red) in Flag-MYCN overexpressed (Dox +, from day 5 onwards) SN-stage cells with either control (Ctrl) or HOXC9 overexpression (OE). HOXC9 OE was performed from day 9 of differentiation. Box-whisker plots show the quantification of the neurite length, TUBB3, and PRPH intensity. The median is indicated by a horizontal line, the boxes represent the 25th to 75th percentiles, the whiskers show the 10th to 90th percentiles, and any outliers beyond this range are displayed as individual dots. Experiments were performed in three independent biological replicates. Two-tailed unpaired *t* test was used. Scale bar represents 100 µm. (C) Illustration describing the recruitment of dCasRx-FTO at target RNA. Representative IF showing expression of PRPH (green) and TUBB3 (red) in SK-N-BE(2) cells expressing dCasRx-FTO^Mutanat (catalytically dead-H231A and D233A mutant)/dCasRx-FTO^WT (wild-type) with either non-template control (NTC gRNA) or HOXC9 guide RNAs (HOXC9 gRNA-1, HOXC9 gRNA-2). Dox induction was performed for 72 h followed by 3 days of RA-mediated differentiation in the presence of Dox. Box-whisker plots show neurite length. The median is indicated by a horizontal line, the boxes represent the 25th to 75th percentiles, the whiskers show the 10th to 90th percentiles, and any outliers beyond this range are displayed as individual dots. Data are from three independent experiments. Two-way ANOVA with Šídák's multiple comparisons test was used. Scale bar represents 50 µm. (D) Representative IF showing expression of HOXC9 (green) in SK-N-BE(2) cells in the same condition as detailed above in (C). Box-whisker plots show mean HOXC9 intensity normalized with DAPI intensity. The median is indicated by a horizontal line, the boxes represent the 25th to 75th percentiles, the whiskers show the 10th to 90th percentiles, and any outliers beyond this range are displayed as individual dots. Signal intensity measurements were taken from over 2000 cells. Data are from three independent experiments. Two-way ANOVA with Šídák's multiple comparisons test was used. Scale bar represents 50 µm. (E) Representative immunoblot showing expression of HOXC9 in control (TetO shCtrl) and METTL3 KD (TetO shM3-1, TetO shM3-2) SK-N-BE(2) cells, along with shRNA-mediated KD of HOXC9 (shHOXC9). Dox induction was performed for 72 h. Vinculin was loading control. The values below indicate the fold change in levels of HOXC9. The experiments were repeated three times. (F) IF showing expression of PRPH (green) and TUBB3 (red) in TetO shCtrl, TetO shM3-1, and TetO shM3-2 SK-N-BE(2) cells in similar conditions as described in (E), except Dox was added for 24 h after HOXC9 shRNA transduction followed by 3 days RA-mediated differentiation in the presence of Dox. Box-whisker plots show the quantification of the neurite length. The median is indicated by a horizontal line, the boxes represent the 25th to 75th percentiles, the whiskers show the 10th to 90th percentiles, and any outliers beyond this range are displayed as individual dots. Experiments were performed in three independent biological replicates. Two-way ANOVA with Šídák's multiple comparisons test was used. Scale bar represents 50 µm. Source data are available online for this figure.

METTL3 has been shown to regulate DNA double-strand break repair (Zhang et al, 2020). We therefore reasoned that METLL3 inhibition would result in the accumulation of DNA damage caused by compromised DNA repair. Indeed, MYCN overexpressing, STM2457-treated cells at the SN-stage showed an accumulation of the DNA damage markers RPA32 and gamma-H2AX (Fig. 6E). An increase in DNA damage was also detected in MYCN overexpressing SN cells post METTL3 KD (Appendix Fig. S6H). We also performed RPA32 IF in METTL3 KD SK-N-BE(2) cells, and again these cells showed accumulation of DNA damage (Fig. 6F). MYCN overexpression creates transcriptional and replication stress thereby promoting DNA damage (Papadopoulos et al, 2022). These DNA damages are required to be repaired efficiently for the survival of MNA NB cells (Szydzik et al, 2021). As METTL3 inhibition enhances DNA damage in MYCN overexpressing cells, our data suggest that this accumulating DNA damage may drive proliferating MYCN overexpressing cells to differentiate. DNA damage-dependent differentiation has been observed in leukemic cells where creating exogenous double-strand breaks by restriction enzymes was sufficient to induce differentiation (Santos et al, 2014). Consistent with this double-strand break repair, related pathways were identified as top deregulated pathways in both STM2457-treated MYCN overexpressed SN-stage cells and in METTL3 KD RA treated SK-N-BE(2) cells (Appendix Fig. S6F,G).

We further explored METTL3 inhibition-induced DNA damage as a possible combination therapy against NB. We observed that the METTL3 inhibitor STM2457 enhanced the activity of the DNA intercalating anthracycline doxorubicin in MNA SK-N-BE(2) cells (Fig. 6G). MNA NB cells treated with a combination of STM2457, and doxorubicin accumulated higher levels of DNA damage as indicated by enhanced RPA32 IF (Fig. 6G). We next tested this combination of drugs in a patient-derived xenograft (PDX) cell line (COG-N-496h). We found that these drugs acted synergistically resulting in reduced cell viability in the PDX cell line but had no significant effect on tNCC (Appendix Fig. S6I,J).

## METTL3 inhibition in combination with doxorubicin suppresses the growth of NB in vivo

We finally aimed to test the efficacy of STM2457 and doxorubicin combination in an NB xenograft model in vivo. For this purpose, we first utilized a zebrafish xenograft model of MNA SK-N-BE(2) cells. Combined treatment of STM2457 and doxorubicin was well tolerated by the zebrafish larvae, as the chosen concentrations did not cause any observable changes in morphology or mortality. Consistent with our in vitro data, combination treatment showed an overall better treatment response (Appendix Fig. S6K). To validate the efficacy of the combination treatment using another in vivo approach, we used the NSG mice with tumors resulting from the subcutaneous injection of the MNA PDX cell line (COG-N-415). Xenografted mice were treated with either vehicle control, a combination of STM2457 and doxorubicin, or single drugs. We found that STM2457 along with doxorubicin was significantly more potent in reducing tumor volume than single drugs (Fig. 6H). Consistent with reduced tumor size the combination treatment also showed higher expression of DNA damage markers (gamma-H2AX and RPA32) and reduced levels of the proliferation marker Ki67 (Appendix Fig. S6L). We also observed that none of the drugs or the combination had any significant effect on the mouse body weight (Fig. 6H), suggesting that the combination treatment was well tolerated in this treatment model. Overall, these data suggest that METTL3 inhibition may represent an efficacious therapeutic approach in the treatment of MNA NB.

## Discussion

Although conventionally MYCN has been shown to regulate gene expression by influencing the transcriptional machinery (Zeid et al, 2018), our study highlights the role of MYCN in m⁶A-mediated gene regulation. We provide evidence that MYCN and METTL3 co-occupy promoter regions of the m⁶A-modified genes. We further elucidate the

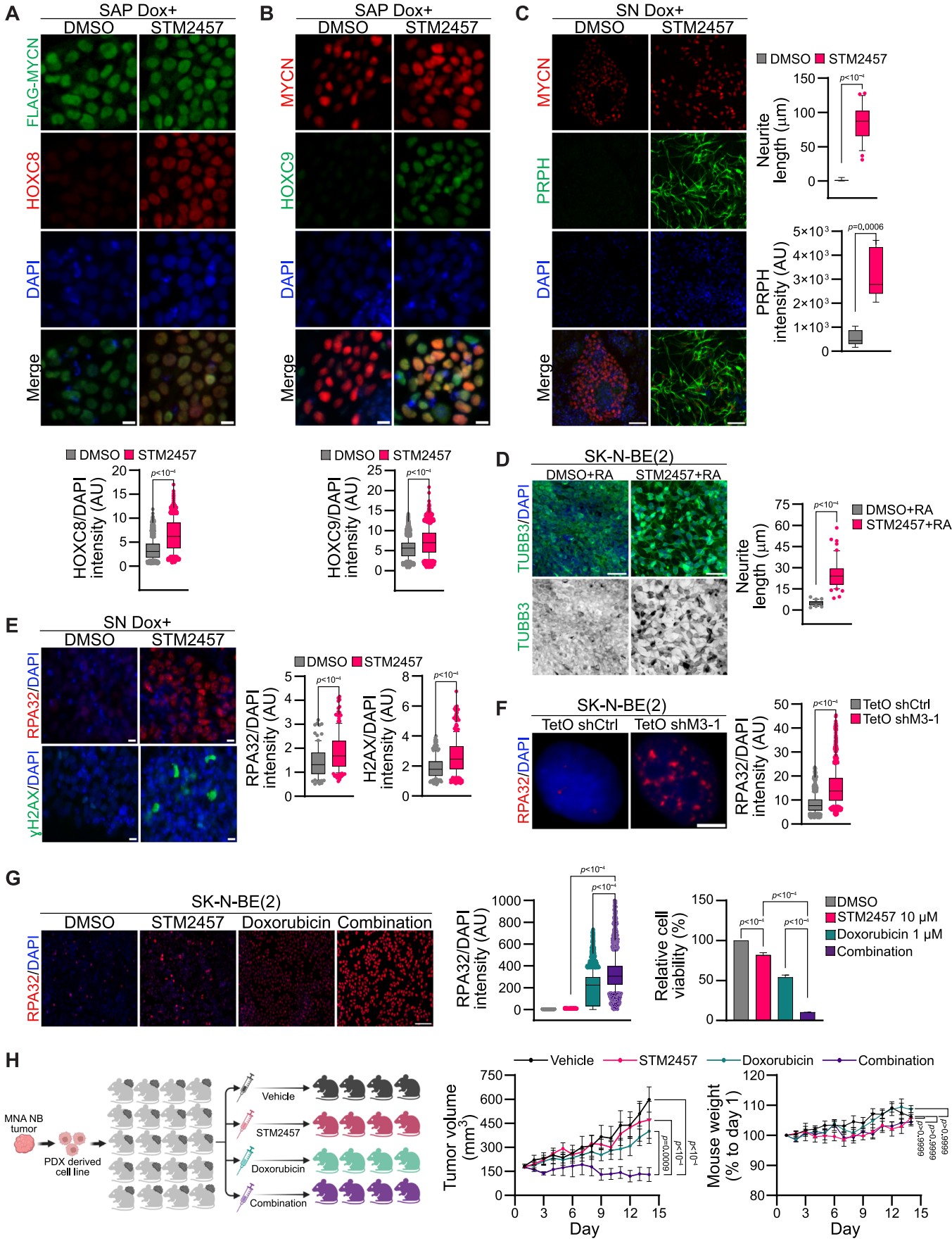

Figure 6. METTL3 inhibition restores differentiation and sensitizes NB to chemotherapy.

(A) HOXC8 (red), and Flag (green) IF were performed in Flag- MYCN overexpressed (Dox +, from day 5 onwards) SAP after DMSO or STM2457 (10 μM) treatment. STM2457 or DMSO was added on day 9 of differentiation. Box-whisker plots show HOXC8 signal intensity normalized to DAPI intensity. The median is indicated by a horizontal line, the boxes represent the 25th to 75th percentiles, the whiskers show the 10th to 90th percentiles, and any outliers beyond this range are displayed as individual dots. Signal intensity measurements were taken from over 2200 cells and data are from three independent experiments. Statistical analysis was performed using a two-tailed unpaired $t$ test. Scale bar represents 10 μm. (B) HOXC9 (green), and MYCN (red) IF were performed in Flag- MYCN overexpressed (Dox +, from day 5 onwards) SAP after DMSO or STM2457 (10 μM) treatment. STM2457 or DMSO was added on day 9 of differentiation. Box-whisker plots show HOXC9 intensity normalized to DAPI intensity. The median is indicated by a horizontal line, the boxes represent the 25th to 75th percentiles, the whiskers show the 10th to 90th percentiles, and any outliers beyond this range are displayed as individual dots. Signal intensity measurements were taken from over 1500 cells and data from three independent experiments. Statistical analysis was performed using a two-tailed unpaired $t$ test. Scale bar represents 10 μm. (C) PRPH (green), and MYCN (red) IF were performed in Flag- MYCN overexpressed (Dox +, from day 5 onwards) SN-stage cells after DMSO or STM2457 (10 μM) treatment. STM2457 or DMSO was added from day 9 of differentiation. Box-whisker plots show quantification of PRPH signal intensity and neurite length. The median is indicated by a horizontal line, the boxes represent the 25th to 75th percentiles, the whiskers show the 10th to 90th percentiles, and any outliers beyond this range are displayed as individual dots. Data are from three independent experiments. Statistical analysis was performed using a two-tailed unpaired $t$ test. Scale bar is 50 μm. (D) Representative IF images of TUBB3 (green) in SK-N-BE(2) cells that were pretreated with either DMSO or STM2457 (10 μM) for 24 h, followed by RA treatment for another 3 days. Box-whisker plots show the quantification of neurite length. The median is indicated by a horizontal line, the boxes represent the 25th to 75th percentiles, the whiskers show the 10th to 90th percentiles, and any outliers beyond this range are displayed as individual dots. Data are from three independent experiments. Statistical analysis was performed using a two-tailed unpaired $t$ test. Scale bar is 50 μm. (E) RPA32 (red) [top] and gamma-H2AX (green) [bottom] IF were performed in Flag-MYCN overexpressed (Dox +, from day 5 onwards) SN-stage cells after DMSO or STM2457 (10 μM) treatment. STM2457 or DMSO was added from day 13 of differentiation. Box-whisker plots show either RPA32 or gamma-H2AX signal intensity normalized to DAPI intensity. The median is indicated by a horizontal line, the boxes represent the 25th to 75th percentiles, the whiskers show the 10th to 90th percentiles, and any outliers beyond this range are displayed as individual dots. Signal intensity measurements were taken from over 90 cells and data are from three independent experiments. Statistical analysis was performed using a two-tailed unpaired $t$ test. Scale bar is 10 μm. (F) RPA32 (red) IF was performed in SK-N-BE(2) cells with TetO shCtrl or TetO shM3-1 after 48 h Dox induction. Box-whisker plots show RPA32 signal intensity normalized to DAPI intensity. The median is indicated by a horizontal line, the boxes represent the 25th to 75th percentiles, the whiskers show the 10th to 90th percentiles, and any outliers beyond this range are displayed as individual dots. Signal intensity measurements were taken from over 800 cells and data are from three independent experiments. Statistical analysis was performed using a two-tailed unpaired $t$ test. Scale bar is 5 μm. (G) Left: Representative IF showing expression of RPA32 (red) in SK-N-BE(2) cells treated either with DMSO, STM2457 (10 μM), doxorubicin (1 μM), or a combination of STM2457 with doxorubicin for 24 h. Scale bar is 100 μm. Middle: Box-whisker plots show RPA32 signal intensity normalized to DAPI intensity. The median is indicated by a horizontal line, the boxes represent the 25th to 75th percentiles, the whiskers show the 10th to 90th percentiles, and any outliers beyond this range are displayed as individual dots. Signal intensity measurements were taken from over 1900 cells. Data are from three independent experiments. Right: Bar plots show relative cell viability in SK-N-BE(2) cells treated for 72 h with DMSO, STM2457, Doxorubicin, and a combination of STM2457 with doxorubicin. Data are presented as mean ± SEM from three independent experiments. Statistical analysis was conducted using a one-way ANOVA with Tukey's post hoc test. (H) Left: Cartoon demonstrating the experimental strategy used for the mouse in vivo experiment performed with patient-derived xenograft (PDX) cells. MNA COG-N-415x, PDX cells were injected into NSG mice. Once tumors reached 170 mm³ mice were randomly allocated into four treatment groups ($n$ = 4–6 mice per group) and treated for 14 days with either vehicle (20% hydroxypropyl-beta cyclodextrin) daily, STM2457 (50 mg/kg in vehicle) daily, doxorubicin (0.2 mg/kg in vehicle) every three days or a combination of STM2457 and doxorubicin at the same doses. Line plots show tumor volume (middle) and body weight (right) in the treatment groups. Data are presented as mean ± SEM. Statistical analysis was conducted using a two-way ANOVA with Tukey's post hoc test. Source data are available online for this figure.

mechanistic insight into how MYCN interaction with the m⁶A writer complex could bring m⁶A modification in developmentally regulated genes such as *HOXC8* and *HOXC9*. A similar mechanism has also been described in the case of SMAD2 which interacts with METTL3 to regulate m⁶A deposition in mRNA (Bertero et al, 2018). However, sustained MYCN overexpression, as observed in MNA NB tumors, results in aberrant epitranscriptomic regulation and deregulation of critical genes such as *HOXC8* and *HOXC9*. We need further understanding of how METTL3 recruitment in gene promoter could guide m⁶A methyltransferase complex to the specific locations in the RNA, such as the 3′end of the transcript (Barbieri et al, 2017). DRACH motifs are the predominant m⁶A sites but the number of DRACH motifs is higher than the detected m⁶A sites present in the RNA. Exon junction complexes (EJC) have been proposed as one of the guiding factors for m⁶A site selection by inhibiting the occupancy of the m⁶A methyltransferase complex close to exon junctions (He et al, 2023; Uzonyi et al, 2023; Yang et al, 2022). We provide evidence that METTL3 recruitment by MYCN can enhance the interaction of the m⁶A methyltransferase complex with active chromatin modification H3K36me3, normally present in the gene-body region of active genes (Huang et al, 2019). We propose that once the m⁶A methyltransferase complex is recruited to gene promoters by transcription factors, such as MYCN and SMAD2, interaction with H3K36me3 further guides co-transcriptional m⁶A deposition, whereas exon junctional complex facilitates m⁶A site selection by sterically blocking METTL3 (Yang et al, 2022).

We show that METTL3 KD or inhibition can promote differentiation in MYCN overexpressed tNCC and MNA NB cells (Fig. 6). The m⁶A+ genes in NB tumors and MYCN/METTL3 co-bound genes that were m⁶A+ in tNCC were related to axon guidance (Fig. 2E; Appendix Fig. S4G). This suggests that MYCN and METTL3-mediated epitranscriptomic regulation might play a broader role apart from *HOX* gene regulation and could be a key player in MYCN-induced oncogenic transformation of the tNCC (Cohen et al, 2020; Saldana-Guerrero et al, 2024), which requires further investigation. Our study paves the way for future studies on the mechanistic understanding of how METTL3 can be guided to specific genomic locations by MYCN or other oncogenic transcription factors to drive m⁶A modification in cancers.

Using both METTL3 KD and MYCN overexpression approaches, we demonstrated that HOXC9 expression is altered, and levels of HOXC9 are critical during neural crest differentiation. Furthermore, the enrichment of the HOXC9 motif at promoters of gene deregulated following MYCN overexpression in SAP suggests that there is a cascade of gene regulation, and this is tightly regulated by the levels of HOXC9. When the HOXC9 levels were restored in MYCN overexpressed cells, we rescued the differentiation phenotype, hinting that genes regulated by HOXC9 are vital for tNCC differentiation. HOXC9 has been previously reported as the top significantly downregulated gene in high-risk NB (Kocak et al, 2013). Analysis of epigenetic regulation, such as DNA methylation, however, did not explain the downregulation of

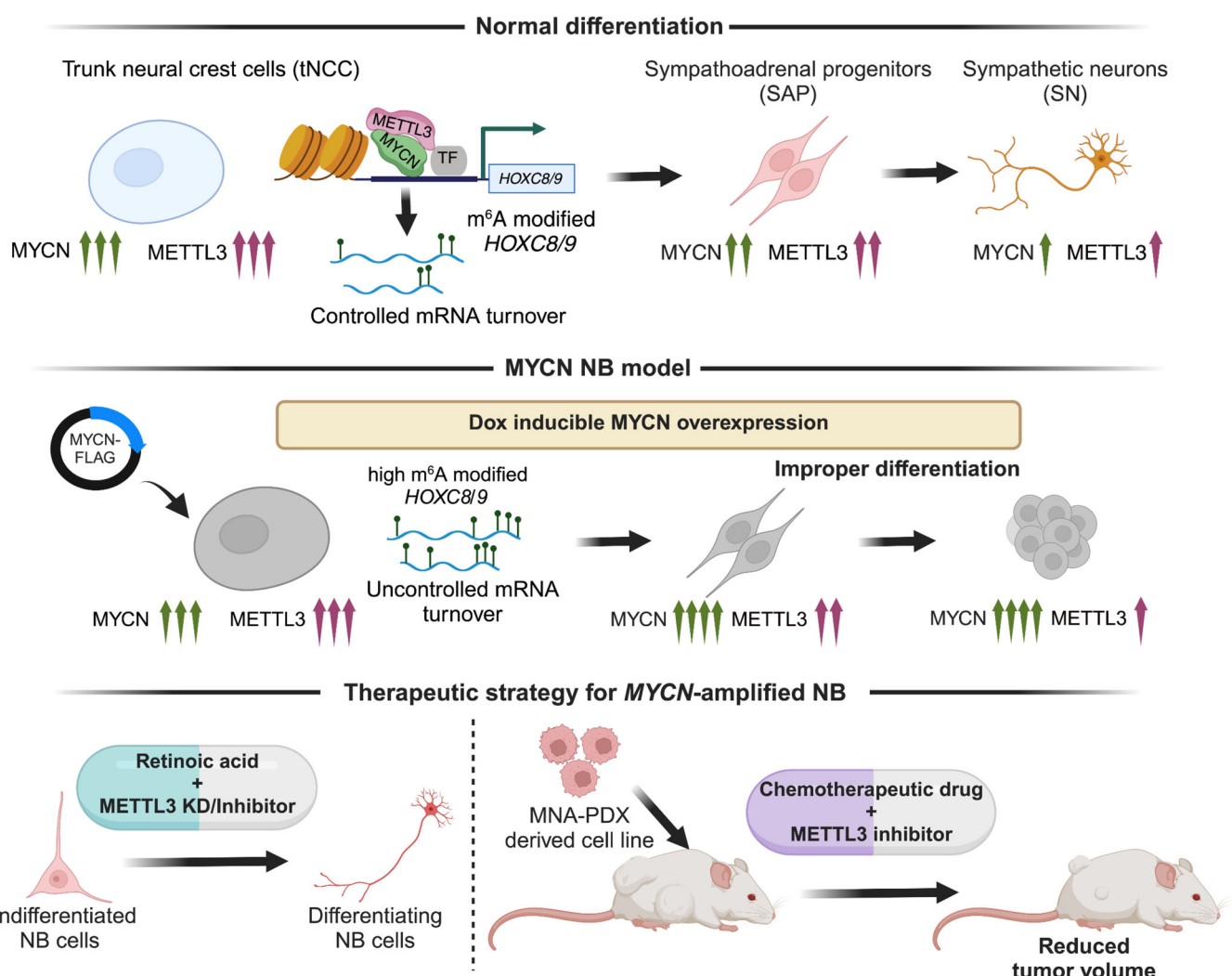

**Figure 7. Working model of MYCN and METTL3 cooperation in *HOX* gene regulation and therapeutic implications of METTL3 inhibition for MNA NB.**

The schematic illustrates (top) the dynamic fine-tuning of developmentally important posterior *HOX* genes, during differentiation of tNCC and further to SN by the cooperation of MYCN and METTL3. METTL3 deposits m⁶A RNA modification on *HOX* genes, thereby facilitating controlled mRNA turnover leading to a normal differentiation process. Middle: To better comprehend the role of METTL3 and MYCN during early differentiation, we created an MYCN NB model by overexpressing MYCN during the tNCC-SN differentiation process. The MYCN overexpression led to an increase in m⁶A modification of *HOX* genes and further the tNCC failed to differentiate to SN. Bottom: As the NB cells failed to differentiate, we utilized RA along with METTL3 inhibitor (STM2457) and observed restoration of differentiation phenotype. Finally, we designed a therapeutic strategy using METTL3 inhibitor to treat MNA high-risk NB tumors. Combining doxorubicin and STM2457 had a synergistic effect on tumor volume in the MNA NB in vivo PDX model.

*HOXC9* gene expression in high-risk NB (Kocak et al, 2013). We here uncover an unexpected role of METTL3-mediated m⁶A modification in controlling *HOX* gene expression in NB. We provide evidence that m⁶A epitranscriptomic modification could explain deregulation in the expression of posterior *HOX* genes in NB. Differentiation of tNCC needs to be regulated tightly but also dynamically. The migrating tNCC, once delaminate from the neural tube, differentiate at various developmental time points. We propose that the m⁶A epitranscriptome-mediated gene regulation provides flexibility by rapidly regulating important lineage-specific transcription factors during the differentiation of the crest cells. Epigenetic regulation of *HOX* genes is well studied (Schuetten-gruber et al, 2017). Here we for the first time provide evidence of an m⁶A epitranscriptomic modification-dependent regulation of the

*HOX* genes in NB. Several studies have implicated the phenotypic plasticity of NB tumor cells. Epigenetic mechanisms along with external environmental cues have been implicated in such phenotypic plasticity (Thirant et al, 2023; van Groningen et al, 2017). Given that m⁶A modification can regulate epithelial to mesenchymal transition (Lin et al, 2019), further studies are required to reveal if epitranscriptomic-based regulation contributes to phenotypic plasticity in NB tumors.

Apart from its role in the regulation of post-transcriptional gene expression such as RNA stability, recent studies have shown the role of m⁶A modification in gene transcription (Liu et al, 2021). METTL3-mediated m⁶A modification of promoter-associated RNA can recruit polycomb repressive complex 2 (PRC2) in a YTHDC1-dependent manner (Dou et al, 2023). Our data on promoter-bound

METTL3 suggest that m⁶A modification can control the epigenetic state in differentiating tNCC, and this might contribute to a widespread deregulation in gene expression observed following METTL3 KD. We speculate that the observed METTL3 and MYCN co-binding at the gene promoter can drive m⁶A modification of promoter-associated transcripts, thereby affecting the epigenetic state of NB cells, and this requires further investigation.

Our gene expression data suggest that genes related to the DNA damage response were upregulated when METTL3 was inhibited using a small molecule inhibitor in MYCN overexpressing SN-stage cells. Consistent with this, we detected an increase in DNA damage markers following METTL3 KD and/or inhibition in MYCN overexpressing SN cells. We hypothesize that induction of DNA damage following pharmacologic inhibition of METTL3 acts as a further trigger for differentiation of the MYCN overexpressing cells apart from *HOXC8/HOXC9* upregulation. Differentiation induced by DNA damage has been previously reported in several other experimental models (Santos et al, 2014; Sherman Bassing and Teitell, 2011). We explored this idea further in MNA NB cells and PDX cell lines where METTL3 pharmacologic inhibition combined with doxorubicin was effective in inhibiting cell viability. Consistent with our in vitro data, treatment of MNA PDX mice with a combination of doxorubicin and STM2457 led to a significant reduction of tumor volumes compared to single agents. We propose that treating tumor cells with a METTL3 inhibitor to sensitize them to chemotherapeutic drugs could be an effective treatment strategy. There is a growing interest in developing more effective METTL3 inhibitors, and recently STM3006 was described as 20 times more potent than STM2457 in cell-based assays. However, the bioavailability of STM3006 was limited, because it was rapidly metabolized, highly reducing the drug's effectiveness in vivo (Guirguis et al, 2023).

Overall, our findings reveal that MYCN can cooperate with METTL3 to establish an m⁶A epitranscriptomic signature over developmentally regulated *HOXC8* and *HOXC9* genes. We provide evidence that pharmacological inhibition of METTL3 could be a novel therapeutic approach for high-risk NB, by inducing differentiation and increasing the efficacy of the chemotherapeutic drugs (Fig. 7).

# Methods

### Reagents and tools table

| Reagent/resource | Reference or source | Identifier or catalog number |
|---|---|---|
| **Experimental models** | | |
| SK-N-BE(2) | DSMZ | ACC 632, RRID:CVCL_0528 |
| IMR-32 | DSMZ | 300148, RRID:CVCL_0346 |
| NGP | DSMZ | ACC 676, RRID:CVCL_2141 |
| SHEP | Gift from Dr. Marie Arsenian-Henriksson, Karolinska Institute, Sweden | RRID:CVCL_0524 |
| COG-N-496h | Children's Oncology Group, Texas, USA | |
| COG-N-415x | Children's Oncology Group, Texas, USA | |

| Reagent/resource | Reference or source | Identifier or catalog number |
|---|---|---|
| WA09 (H9) | Gift from Dr. Fredrik H. Sterky, Sahlgrenska University Hospital, Gothenburg, Sweden | RRID:CVCL_9773 |
| Mouse: Crl:NU(NCr)-Foxn1nu | Charles River | |
| **Antibodies** | | |
| Mouse monoclonal anti-Alpha-tubulin | Sigma-Aldrich | Cat# T5168 |
| Mouse monoclonal anti-beta-tubulin Isotype III | Sigma-Aldrich | Cat# T5076, RRID:AB_532291 |
| Mouse monoclonal anti-FLAG | Sigma-Aldrich | Cat# F1804, RRID:AB_262044 |
| Rabbit polyclonal anti-GAPDH | Cell Signaling Technology | Cat# 5174 |
| Rabbit polyclonal anti-H3K27ac | Abcam | Cat# ab4729 |
| Mouse monoclonal anti-H3K36me3 | Activ Motif | Cat# 61021 |
| Rabbit monoclonal anti-HA-tag | Cell Signaling Technology | Cat# 3724 |
| Rabbit polyclonal anti-HOXC8 | Atlas Antibodies | Cat# HPA028911, RRID:AB_10602236 |
| Mouse monoclonal anti-HOXC9 | Santa Cruz Biotechnology | Cat# sc-81100, RRID:AB_2279855 |
| Mouse monoclonal anti-HOXC9 | Abcam | Cat# ab50839 |
| Rabbit monoclonal anti-HSP90 | Cell Signaling Technology | Cat# 4877S |
| Rabbit IgG | Santa Cruz Biotechnology | Cat# sc-2027 |
| Rabbit Monoclonal anti-Ki67 | ThermoFisher | Cat# MA5-14520 |
| Rabbit polyclonal anti-m⁶A | Synaptic Systems | Cat# 202 003, RRID:AB_2279214 |
| Rabbit polyclonal anti-METTL14 | Atlas Antibodies | Cat# HPA038002 |
| Rabbit monoclonal anti-METTL3 | Abcam | Cat# ab195352, RRID:AB_2721254 |
| Mouse monoclonal anti-Myc-tag | DSHB | Cat# 9E10 |
| Rabbit monoclonal anti-N-Myc (D4B2Y) | Cell Signaling Technology | Cat# 51705, RRID:AB_2799400 |
| Mouse monoclonal anti-N-Myc antibody [NCM II 100] | Abcam | Cat# ab16898, RRID:AB_443533 |
| Mouse monoclonal anti-Oct-3/4 (C-10) | Santa Cruz Biotechnology | Cat# sc-5279, RRID:AB_628051 |
| Mouse monoclonal anti-Phospho-Histone H2A.X (Ser139) | ThermoFisher | Cat# MA1-2022, RRID:AB_559491 |
| Rabbit Polyclonal anti-Phospho-Histone H2A.X (Ser139) | Abcam | Cat# ab11174 |
| Rabbit polyclonal anti-Phospho-RPA32 | ThermoFisher | Cat# A300-246A |
| Rabbit polyclonal anti-Phospho-RPA32 (Ser33) | ThermoFisher | Cat# A300-246A |
| Mouse monoclonal anti-Phox2b (B-11) | Santa Cruz Biotechnology | Cat# sc-376997, RRID:AB_2813765 |

| Reagent/resource | Reference or source | Identifier or catalog number |
|---|---|---|
| Mouse monoclonal anti-PRPH (A-3) | Santa Cruz Biotechnology | Cat# sc-377093 |
| Rabbit polyclonal anti-RBM15 | Atlas Antibodies | Cat# HPA019824, RRID:AB_1856113 |
| Mouse monoclonal anti-Vinculin (7F9) | Santa Cruz Biotechnology | Cat# sc-73614, RRID:AB_1131294 |
| Rabbit monoclonal anti-WTAP [EPR18744] | Abcam | Cat# ab195380 |
| Goat polyclonal anti-Mouse IgG (H + L), Alexa Fluor Plus 488 | ThermoFisher | Cat# A32723, RRID:AB_2633275 |
| Goat polyclonal anti-Mouse IgG (H + L), Alexa Fluor Plus 555 | ThermoFisher | Cat# A32727, RRID:AB_2633276 |
| Goat polyclonal anti-Rabbit IgG (H + L), Alexa Fluor Plus 488 | ThermoFisher | Cat# A32731, RRID:AB_2633280 |
| Goat polyclonal anti-Rabbit IgG (H + L), Alexa Fluor Plus 555 | ThermoFisher | Cat# A32732, RRID:AB_2633281 |
| **Oligonucleotides and other sequence-based reagents** | | |
| **Oligos used for cloning** | | |
| NTC gRNA F | Sigma-Aldrich | AAACACAGCAAATA TTGCAGAACAGCC |
| NTC gRNA R | Sigma-Aldrich | AAAAGGCTGTTCT GCAATATTTGCTGT |
| HOXC9 gRNA-1 F | Sigma-Aldrich | AAACGAAACCAGA TTTTGACCTGCCGC |
| HOXC9 gRNA-1 R | Sigma-Aldrich | AAAAGCGGCAGGT CAAAATCTGGTTTC |
| HOXC9 gRNA-2 F | Sigma-Aldrich | AAACTGAAACCAGA TTTTGACCTGCCG |
| HOXC9 gRNA-2 R | Sigma-Aldrich | AAAACGGCAGGTCA AAATCTGGTTTCA |
| **Chemicals, enzymes, and other reagents** | | |
| 2-Mercaptoethanol | ThermoFisher | 31350010 |
| Accutase | STEMCELL Technologies | 7920 |
| Actinomycin D | Sigma-Aldrich | A9415 |
| B-27 Supplement (50X) | ThermoFisher | 12587010 |
| BrainPhys Neuronal Medium | STEMCELL Technologies | 5790 |
| CellTracker CM-DiI Dye | ThermoFisher | C7000 |
| Click-iT RNA Alexa Fluor 488 Imaging Kit | ThermoFisher | C10329 |
| Cycloheximide (CHX) | Sigma-Aldrich | 1810 |
| Dimethyl sulfoxide (DMSO) | Sigma-Aldrich | D8418 |
| DMH-1 | Tocris | 4126 |
| Doxorubicin (hydrochloride) | MedChemExpress | HY-15142 |
| Doxycycline (Dox) | Sigma-Aldrich | D3447 |
| Flavopiridol hydrochloride hydrate | Sigma-Aldrich | F3055 |
| HP-β-CD vehicle | MedChemExpress | HY-101103 |
| Human BDNF | Miltenyi Biotec | 130-093-811 |
| Human BMP-4 | Miltenyi Biotec | 130-111-168 |

| Reagent/resource | Reference or source | Identifier or catalog number |
|---|---|---|
| Human FGF-2 (bFGF) | Miltenyi Biotec | 130-093-840 |
| Human NGF-β | Miltenyi Biotec | 130-127-431 |
| Human SHH (C24II) | Miltenyi Biotec | 130-095-727 |
| Human GDNF | Miltenyi Biotec | 130-096-291 |
| Matrigel Matrix Growth Factor Reduced | Corning | 354230 |
| N-2 Supplement (100X) | ThermoFisher | 17502048 |
| NaveniFlex MR | Navinci Diagnostics | NC.MR.100 |
| Neurobasal Medium | ThermoFisher | 21103049 |
| ProLong Gold Antifade Mountant with DNA Stain DAPI | ThermoFisher | P36931 |
| Retinoic acid | Sigma-Aldrich | R2625 |
| StemMACS CHIR99021 | Miltenyi Biotec | 130-106-539 |
| StemMACS Cryo-Brew | Miltenyi Biotec | 130-109-558 |
| StemMACS iPS-Brew XF, human | Miltenyi Biotec | 130-104-368 |
| StemMACS Purmorphamine | Miltenyi Biotec | 130-104-465 |
| StemMACS SB431542 | Miltenyi Biotec | 130-106-543 |
| StemMACS Y27632 | Miltenyi Biotec | 130-106-538 |
| STM2457 | MedChemExpress | HY-134836 |
| **Software** | | |
| Prism | GraphPad | https://www.graphpad.com/features |
| ggplot2 | Wickham, 2016 | https://ggplot2.tidyverse.org |
| ImageLab software | BioRad | https://www.bio-rad.com/en-il/product/image-lab-software?ID=KRE6P5E8Z |
| R 4.3.1 | R-project | https://cran.r-project.org/bin/windows/base/ |
| ChIPpeakAnno | Zhu et al, 2010 | https://bioconductor.org/packages/ChIPpeakAnno/ |
| ClusterProfiler | Wu et al, 2021 | https://bioconductor.org/packages/clusterProfiler/ |
| DESeq2 | Love Huber and Anders, 2014 | https://bioconductor.org/packages/DESeq2/ |
| MACS (2.2.6) | Zhang et al, 2008 | https://pypi.org/project/MACS2/ |
| DiffBind | Ross-Innes et al, 2012 | https://bioconductor.org/packages/DiffBind/ |
| HISAT2 | Kim et al, 2019 | http://daehwankimlab.github.io/hisat2/ |

| Reagent/resource | Reference or source | Identifier or catalog number |
|---|---|---|
| deepTools2 v3.3.2 | Ramirez et al, 2016 | https://test-argparse-readoc.readthedocs.io/en/latest/content/installation.html |
| Homer v4.11 | Heinz et al, 2010 | http://homer.ucsd.edu/homer/download.html |
| ggpubr | R-Project | https://cran.r-project.org/web/packages/ggpubr/index.html |
| plyranges | Lee Cook and Lawrence, 2019 | https://bioconductor.org/packages/ReactomePA/ |
| picard v2.23.4 | https://github.com/broadinstitute/picard | https://github.com/broadinstitute/picard/releases/tag/2.23.4 |
| samtools v1.12 | | http://github.com/samtools/ |
| Trim Galore | https://zenodo.org/records/7598955 | https://github.com/FelixKrueger/TrimGalore/tree/0.6.10 |

**Other**

**qPCR primers for human**

| Primer | Forward sequence 5'–3' | Reverse sequence 5'–3' |
|---|---|---|
| NPM1 | TTCACCGGGAAGCATGG | CACGCGAGGTAAGTCTACG |
| PRPH | CCCTGGGGATTAGGGAGAGT | AGCTACCCCTCCTTCACCAC |
| SEMA3D | GGGACAAGAGGGGACAGTTT | AATACCGTGGGTCACAGAGG |
| NRCAM | CCGGTGTTATGAGAGCTTGG | AGAGGGCGCTTGTATTAGCA |
| HOXC8 | GGACTGACCGAGAGACAAGT | CACCTTCTCTCATCTCGGG |
| HOXC9 | AAAATACCCCAACACAGGCG | AACCCTCCCAAATCGCAAG |
| HOXC13 | TGTTAAGGAAAGAGAAGAACCGC | GGGATGGGATAGGGAGTTGG |
| ASCL1 | CGACTTCACCAACTGGTTCTG | ATGCAGGTTGTGCGATCA |
| DBH | TCTCCATGCACTGCAACAA | GGCTGCAGGTTCCATTCA |
| GAPDH | TTAAAAGCAGCCCTGGTGAC | CTCTGCTCCTCCTGTTCGAC |
| GATA2 | GGTGCTAGGGTCAGGAGACA | GGAGGCCCACTCTCTGTGTA |
| HOXC8 | AACTCGTCTCCCAGCCTCATGT | TCTAGTTCCAAGGTCTGATACCG |
| HOXC9 | GCTGGAACTGGAGAAGGAGT | AACCAGATTTTGACCTGCCG |
| ISL1 | TTGCCTCGGGAGCCCTAATC | ATCATATTTCAGCCTCGCCGC |
| METTL14 | CTGAAAGTGCCGACAGCATTGG | CTCTCCTTCATCCAGATACTTACG |
| METTL3 | ATTTCTTGGCTGCTCCTTT | GCTGACCATTCCAAGCTCTC |

| Reagent/resource | Reference or source | Identifier or catalog number |
|---|---|---|
| NANOG | TTCCTTCCTCCATGGATCTG | TCTGCTGGAGGCTGAGGTAT |
| NGFR | GAGGCACCTCCAGAACAAGA | AGACAGGGATGAGGTTGTCG |
| OCT4 | AGGGACCGAGGAGTACAGTG | AGCGATCAAGCAGCGACTAT |
| PHOX2B | GGTTGGGATTGGGACCTG | CTCCTCGGGCAAAAAGTCT |
| SOX10 | ATAGGGTCCTGAGGGCTGAT | AGCCCAGGTGAAGACAGAGA |
| TFAP2A | ATGGCGTGAGGTAAGGAGTG | GATCCTCGCAGGGACTACAG |
| TH | ACGCCAAGGACAAGCTCA | AGCGTGTACGGGTCGAACT |

**siRNAs and shRNAs**

| | | |
|---|---|---|
| shCtrl | Sigma-Aldrich | Cat# SHC016 |
| Control-siRNA | ThermoFisher Scientific | Cat# AM4611 |
| shMETTL3-2 | Sigma-Aldrich | Cat# TRCN0000289814 |
| shMETTL3-1 | Sigma-Aldrich | Cat# TRCN0000289812 |
| METTL3-siRNA 1 | ThermoFisher Scientific | Cat# AM16708, ID 132906 |
| METTL3-siRNA 2 | ThermoFisher Scientific | Cat# AM16708, ID 132907 |
| METTL3-siRNA 3 | Thermo Fisher Scientific | Cat# AM16708, ID 132908 |
| TetO shCtrl (pLKO-Tet-On-shRNA-Control) | Addgene | Cat# 98398 |
| TetO shM3-2 | This study | CGTCAGTATCTTGGGCAAGTT |
| TetO shM3-1 | This study | GCTGCACTTCAGACGAATTAT |

**Plasmids**

| | | |
|---|---|---|
| CasRx-HA | Addgene | Cat# 138149 |
| dCasRx-FTO$^{Mutant}$-HA | In-house | Vaid et al, 2024 |
| dCasRx-FTO$^{WT}$-HA | Gift from Dr. Wenbo Li, University of Texas Health Science Center, Houston, USA | NA |
| HOXC8 (NM_022658) Human Tagged ORF Clone | Origene | Cat# RC208810 |
| HOXC9 (NM_006897) Human Tagged Lenti ORF Clone | Origene | Cat# RC208833L3 |
| HOXC9 (NM_006897) Human Tagged ORF Clone | Origene | Cat# RC208833 |
| PB-TRE3G-MYCN | Addgene | Cat# 104542 |
| pLKO-Tet-On-shRNA-Control plasmid | Addgene | Cat# 98398 |
| Tet-pLKO-puro vector | Addgene | Cat# 21915 |
| XLone-GFP | Addgene | Cat# 96930 |

## Neuroblastoma (NB) cell lines and culture conditions

*MYCN*-amplified NB cell lines SK-N-BE(2), IMR-32, and NGP were used in this study. SK-N-BE(2) and NGP were procured from DSMZ, whereas IMR-32 was from CLS Cell Lines Service. SHEP cells were a gift from Dr. Marie Arsenian-Henriksson (Karolinska Institute, Sweden). SK-N-BE(2) cells were cultured in DMEM/F-12 media supplemented with 10% FBS, 1× GlutaMAX, and penicillin/streptomycin. NGP and SHEP cells were cultured in Dulbecco's modified Eagle's medium (DMEM) supplemented with 10% fetal bovine serum (FBS) and penicillin/streptomycin. IMR-32 cells were cultured in Minimum Essential Medium (MEM) supplemented with 10% FBS, 1 mM sodium pyruvate, 1× GlutaMAX (Gibco), and penicillin/streptomycin. Patient-derived xenografts (PDX) cells, COG-N-496h (*MYCN*-amplified, ALK WT, P53 WT) and COG-N-415x (*MYCN*-amplified, ALK mutant, P53 WT) were obtained from the Children's Oncology Group (Texas, USA) and cultured in Iscove's Modified Dulbecco's Medium (IMDM) plus 20% FBS, 4 mM L-Glutamine, 1× ITS (5 μg/mL insulin, 5 μg/mL transferrin, 5 ng/mL selenous acid). Cells were confirmed to be mycoplasma-negative using the MycoAlert Mycoplasma detection assay (Lonza, LT07218) and maintained in a humidified incubator at 37 °C with 5% $CO_2$.

## hESC culture and differentiation

Human embryonic stem cell line WA09 (H9) obtained from Dr. Fredrik H. Sterky (Sahlgrenska University Hospital, Gothenburg, Sweden) were cultured on Matrigel-coated plates with iPS-Brew XF (Miltenyi) media. To differentiate hESC to trunk neural crest cells (tNCC), hESC were dissociated using Accutase and seeded (day 0) on Matrigel-coated plates to induce differentiation to neuromesodermal progenitor cells (NMP, day 3) which were then driven to tNCC (days 7–9), sympathoadrenal progenitors (SAP, day 12), and further towards sympathetic neurons (SN, day 19–25) as described previously (Frith et al, 2018; Frith and Tsakiridis, 2019). For neural crest stem cells (NCSC) induction, hESCs were dissociated using Accutase and seeded (day 0) on Matrigel-coated plates at a density of $2 \times 10^4$ cells/cm² in iPS-Brew XF media with ROCK inhibitor (Y27632 10 μM). The following day, media was replaced with NCSC differentiation medium as previously described (Menendez et al, 2013). NCSC differentiation medium was replaced every 2 days, and cells were passaged on Matrigel-coated plates every 3–4 days of reaching 90% confluency.

## Transient gene silencing and overexpression

METTL3 siRNAs were used to induce transient knockdown in SK-N-BE(2) cells with RNAiMAX (ThermoFisher) reagent, following the manufacturer's guidelines. HOXC8 and HOXC9 plasmids were transfected using Lipofectamine 3000 reagent (ThermoFisher), following the manufacturer's guidelines.

## METTL3 shRNA lentiviral packaging and viral transduction

shRNA constructs targeting METTL3 and control shRNA in the pLKO.1 vector were procured from Sigma. To generate lentiviral particles for each shMETTL3 and control shRNA construct, HEK293T cells were co-transfected with these plasmids, along with pMD2.G and psPAX2 packaging plasmids, employing the

CalPhos mammalian transfection kit (Takara Bio). Supernatants were harvested at 48 and 72 h post-transfection and subsequently stored at −80 °C. The METTL3 shRNA sequences were subsequently cloned into a tet-pLKO-puro vector and packaged to produce doxycycline (Dox)-inducible viral particles. Before selection with 1 μg/ml puromycin, NB cells, and hESC were transduced with shMETTL3 or shCtrl viral particles for 48 h. For inducible METTL3 KD, Dox was administered at a concentration of 2 μg/ml during hESC differentiation and 200 ng/ml for NB cells.

## MYCN overexpression

PB-TRE3G-MYCN and XLone-GFP (Randolph et al, 2017) were acquired from Addgene. PB-TRE3G-MYCN was used to amplify MYCN sequence with 1× Flag-tag followed by cloning into XLone vector replacing GFP to get Dox inducible Flag-MYCN. SHEP cells were co-transfected (1:1) with Flag-MYCN and pCYL43 piggyBac transposase using Lipofectamine 3000 as per the manufacturer's protocol. Stably transfected cells were selected by treatment with 5 μg/ml Blasticidin. hESC H9 were nucleofected with two plasmids (1:1) Flag-MYCN and pCYL43 piggyBac transposase to generate stable cells. Amaxa 4D-Nucleofector system (Lonza) was employed for nucleofection as per the manufacturer's instructions. Nucleofected cells were selected for a week using 2.5 μg/ml Blasticidin. Cells were then sparsely seeded, and colonies were picked, expanded, and screened for MYCN expression. Overexpression of MYCN was confirmed on the protein level using immunoblotting after 48 h Dox induction. The clone which showed consistently high MYCN expression was used in further experiments. Dox was used at a concentration of 2 μg/ml during hESC differentiation and 200 ng/ml in NB cells for MYCN overexpression. Flag-MYCN cells were differentiated following the tNCC differentiation protocol as described above with or without Dox treatment. DMSO or METTL3 inhibitor STM2457 (10 μM) was introduced during the differentiation protocol as described in the figure legends. For the rescue experiment, a Ctrl/HOXC9 OE lentivirus was transduced on day 9. HOXC9 OE was confirmed by IF against Myc-tag in D12 SAP-stage cells and differentiated further towards SN.

## Retinoic acid (RA) mediated differentiation of NB cells

SK-N-BE(2) or NGP cells were seeded at a density of $1 \times 10^5$ or $2 \times 10^5$ per well, respectively. The following day, cells were pretreated with DMSO or STM2457 (10 μM) for 24 h and followed by RA (10 μM) for an additional 3 days. A similar procedure was followed in Dox inducible METTL3 KD cells, where cells were induced with Dox for 24 h before the addition of RA for the following 3 days. For differentiation following simultaneous METTL3 and HOXC9 KD, TetO shCtrl/shM3-1 and shM3-2 cells were seeded and transduced with either shCtrl or shHOXC9. The following day, media was replaced to induce METTL3 KD with Dox for 24 h. Cells were then detached and plated on coverslips in RA differentiation media for 3 days in the presence of Dox. A similar procedure was used for the differentiation of dCasRx-FTO cells.

## Immunofluorescence staining (IF)

Cells were fixed in 4% formaldehyde for 10 min at room temperature (RT) and then rinsed twice with PBS. Following this,

cells were permeabilized with 0.25% Triton X-100 in PBS for 10 min, followed by two washes with 0.1% Tween 20 in PBS (PBST). Subsequently, cells were blocked for 1 h at RT in 3% BSA in PBST. Primary antibodies, including METTL3 (1:300), phospho-Histone H2A.X (Ser139) (1:500), phospho-RPA32 (Ser33) (1:300), peripherin (PRPH) (1:100), PHOX2B (1:100), Oct-3/4 (1:100), HOXC9 (1:100), anti-HOXC8 (1:300), FLAG (1:500), MYCN (1:1000), and beta-Tubulin Isotype III (TUBB3) (1:500), were applied and incubated overnight at 4 °C. After primary antibody exposure, cells were washed three times for 5 min each with PBST and then incubated for 1 h in the dark at RT with secondary antibodies labeled with Alexa Fluor 488 and Alexa Fluor 555 fluorochromes (1:800). Following secondary antibody incubation, cells were washed again three times for 5 min each with PBST. Prolong Gold with DAPI (ThermoFisher) was added to each coverslip, mounted on a slide, and air-dried in the dark for nuclei detection. Slides were imaged using a fluorescence microscope EVOS FL Auto (ThermoFisher), and image analysis was performed with ImageJ. METTL3, HOXC8, HOXC9, RPA32, and gamma-H2AX signal intensities were normalized to the DAPI signal.

## Immunoblotting and Co-IP

Cells were lysed in RIPA buffer (ThermoFisher) with a protease inhibitor cocktail (ThermoFisher). Protein concentrations were determined by bicinchoninic acid assay (ThermoFisher). Equal protein amounts were loaded onto a 4–12% Bis-Tris gel (Thermo-Fisher) and transferred to a nitrocellulose membrane with a Trans-Blot Turbo system (BioRad). Membranes were blocked for 1 h in 5% non-fat dry milk in PBST and probed overnight with primary antibodies diluted in 5% blocking solution: METTL3 (1:200), METTL14 (1:2000), MYCN (1:1000), Vinculin (1:5000), GAPDH (1:5000), A-tubulin (1:5000). Membranes were then incubated with HRP-linked secondary antibodies (Cell Signaling) for 1 h, and signals were detected using SuperSignal West Pico PLUS Chemi-luminescent Substrate (ThermoFisher). Blots were developed and quantified using a ChemiDoc system and ImageLab software (BioRad). For primary NB tumors, immunoblots were prepared as described previously (Kocak et al, 2013). Co-IP was performed with METTL3 and MYCN antibodies as described before (Mondal et al, 2018). To test whether the MYCN-METTL3 interaction is a consequence of a co-transcriptional process, SHEP$^{MYCN}$ cells were induced with Dox for 24 h. Following this induction, the cells were treated with 300 nM Flavopiridol (FP) for 1 h before conducting the Co-IP.

## Proximity ligation assay (PLA)

PLA was performed in tNCC or SHEP$^{MYCN}$ cells after 24 h Dox-induced overexpression of MYCN using a Duolink PLA kit (Sigma, DUO92014) according to the manufacturer's protocol. As a background control, a single antibody was used in this assay. Briefly, cells were fixed for 10 min in 4% PFA at RT before being blocked with a blocking solution. The cells were treated with primary antibodies targeting METTL3-MYCN/HOXC8-HOXC9/METTL14-H3K36me3 for 1 h at 37 °C, followed by incubation with PLA probes for 1 h at 37 °C in a humidified chamber. After three washes, a ligation-ligase solution was added and incubated for 30 min at 37 °C. The slides were incubated for 100 min in an

amplification solution containing polymerase at 37 °C in the dark. Finally, the cells were stained with Prolong Gold containing DAPI, and coverslips were mounted on a slide and air-dried. Fluorescence microscopy was used to capture the fluorescence images.

## Proliferation assay

In total, $1 \times 10^4$ cells per well (for SK-N-BE(2), $5 \times 10^3$ cells per well) were seeded on a 96-well plate to assess cellular proliferation. SK-N-BE(2), IMR-32, and SHEP$^{MYCN}$ Dox inducible shCtrl or shMETTL3 cells were seeded and induced on the following day by adding Dox 200 ng/ml up to 6 days. For the combination, experiment in SK-N-BE(2) treatment with STM2457 (10 μM) or with doxorubicin was performed at indicated concentrations for 72 h. CellTiter 96 non-radioactive cell proliferation assay kit (Promega, G4000) was used to determine cell growth, and the manufacturer's instructions were followed. Absorbance was measured using a microplate reader Infinite 50 (Tecan, Austria).

For dose–response matrices, cells were treated with log-scale concentrations of STM2457 in addition to log-scale concentrations of doxorubicin, and the DMSO concentration was maintained at <0.2%. The potential synergy between STM2457 and doxorubicin was evaluated by calculating the synergy score based on the Loewe model (Loewe, 1953) using the synergy finder web application, https://synergyfinder.fimm.fi (Ianevski Giri and Aittokallio, 2022). The synergy score is calculated for each combination of drug concentrations and also as an overall value and is defined as: >10 = synergistic; Between -10 and 10 = additive; <-10 = antagonistic.

## RNA sequencing (RNA-seq) and m⁶A RNA Immunoprecipitation sequencing (m⁶A RIP-seq)

RNA was isolated from cells using TRIZOL reagent (Thermo-Fisher) and Direct-zol RNA Miniprep (ZYMO research). NB cell line, hESC, NCC, and tNCC RNA (15 μg) were spiked-in with Bacterial RNA (10 ng) before fragmentation using RNA Fragmentation Reagents (ThermoFisher). For NB tumors, RNA samples were anonymously provided, and the study was conducted per the Declaration of Helsinki. In total, 3 μg of total RNA was used for fragmentation without bacterial spike-in RNA. This fragmented RNA was either used for RNA-seq or m⁶A RIP-seq. m⁶A RIP was performed as previously described (Vaid et al, 2023; Vaid et al, 2024; Zeng et al, 2018) with m⁶A antibody. Input and m⁶A RIP–RNA were used to generate sequencing libraries using SMARTer-Stranded Total RNA-Seq kit V2, Pico Input Mammalian (Takara Bio). All the libraries were single-end sequenced (1 × 88 bp) on the Illumina NextSeq 2000 platform at the BEA Core Facility (Stockholm, Sweden). Details of the RNA-seq and m⁶A RIP-seq samples used in the study are listed in Dataset EV1.

## RT-qPCR and m⁶A RIP-qPCR

For m⁶A RIP-qPCR, 3 μg of cellular RNA was utilized. RIP assays were conducted using 1 μg of m⁶A antibody or IgG antibody as described above. Both input and m⁶A RIP–RNA were reverse-transcribed to cDNA employing the High-Capacity RNA-to-cDNA kit (ThermoFisher) with random primers. Subsequent qPCR was performed on a Quant Studio 3 instrument (ThermoFisher), utilizing gene-specific PCR primers mixed with Power SYBR Green

Master Mix (ThermoFisher) and diluted cDNA as a template. The resulting data were expressed as percentage input values. For RT-qPCR, RNA was directly converted into cDNA and subjected to qPCR. The expression values for each gene were normalized to *GAPDH* using the delta-delta Ct method.

## RNA stability assay

Transcriptional inhibitor actinomycin D (10 µg/ml) was used to inhibit RNA synthesis. After treatment with actinomycin D, cells were harvested at 3, and 6 h time points, and the RNA was then extracted. The mRNA levels of *HOXC8* and *HOXC9* were detected through RT-qPCR. mRNA half-life was calculated by fitting measurements to a one-phase exponential decay equation using GraphPad Prism. The half-life ($t_{1/2}$) was determined using the equation $t_{1/2} = 0.693/k$, where k is the rate constant for mRNA decay.

## Chromatin immunoprecipitation-qPCR/sequencing (ChIP-qPCR/seq)

ChIP was performed as described before (Vaid Wen and Mannervik, 2020). In brief, cells were fixed with formaldehyde (1% final concentration) for 15 min at RT and quenched using Glycine. After fixation, the cells were subjected to cold PBS wash, followed by lysis, and chromatin shearing using a Bioruptor (Diagenode). The chromatin was sheared until the fragments reached an average size range of 200–500 bp. Subsequently, immunoprecipitation of the solubilized chromatin was conducted using 3 µg of METTL3, MYCN, or H3K27ac antibodies overnight at 4 °C. The immunoprecipitated complex was then captured using a combination of Protein A and G Dynabeads (Invitrogen), washed and RNase A treated. The samples were next incubated at 68 °C for at least 4 h to reverse the cross-links and further treated with proteinase K for 2 h at 37 °C. Finally, the ChIP DNA was eluted using the ChIP DNA Clean & Concentrator™ kit (Zymo Research, D5205). Eluted ChIP DNA was either directly used as a template for qPCR as described earlier with primers or was sequenced. ChIP-seq libraries were prepared using the NEBNext Ultra II DNA Library Prep Kit (NEB, E7645L) and single-end (1 × 75 bp) sequenced on Illumina NextSeq platform at BEA core facility, Stockholm. Details of the ChIP-seq samples used in the study are listed in Dataset EV1.

## Analysis of RNA-seq data

Paired-end RNA-seq reads obtained from BGI DNBseq were analyzed using FastQC v0.11.9 for quality control (https://www.bioinformatics.babraham.ac.uk/projects/fastqc/) using default quality filtering parameter (-q 20). Adapters were removed using Trim Galore v0.6.6 with a minimal length threshold of 20 bp. Trimmed reads were mapped to the GRCh38 human reference genome obtained from GENCODE (Release 36 GRCh38.p13) using HISAT2 v2.2.1 (Kim et al, 2019) with parameters (--sensitive --no-discordant --no-mixed -I 1 -X 1000). After mapping, duplicate alignments were labeled using *markDuplicates* from Picard v2.23.4. Marked alignment files were further processed using Sambamba v0.7.1 (Tarasov et al, 2015) keeping uniquely mapping reads separated by strand after duplicate removal. Aligned reads were

quantified using Salmon v1.4.0 (Patro et al, 2017) with GRCh38 Gencode v36 annotation, following differential expression analysis with DESeq2 (Love Huber and Anders, 2014) using two replicates per condition. Genes were considered differentially expressed if their |log2 fold change| > 1 and adjusted *P* value < 0.01. Normalized TPM counts were calculated to account for differences in gene length and library sizes for downstream analysis and visualization. Functional enrichment analysis for differentially expressed genes was performed using clusterProfiler (Wu et al, 2021) and enrichR (Chen et al, 2013) packages. Data visualization was carried out using custom scripts with the ggplot2 R package (Wickham, 2016).

### Motif analysis

Differential gene expression analysis was performed on RNA-seq data from SAP cells as described above. The differentially expressed genes were selected considering a threshold of $\log_2$ fold change ($\log_2$FC) greater than or less than ±1 and a significance threshold of adjusted *P* value (padj) <0.05. The HOMER software package, version 4.11, was employed for motif discovery analysis (Heinz et al, 2010). The findMotifs.pl function within HOMER was utilized to identify known motifs within the promoter of selected differentially expressed genes. Analysis was conducted using the default human genome assembly (hg38) as the reference.

## Analysis of m⁶A RIP data

Single-end m⁶A RIP sequencing data obtained from SMARTer-Stranded Total RNA-Seq Kit v2 was processed using FastQC v0.11.9 and Trim Galore v0.6.6 for quality control. Trimmed reads were mapped to the GRCh38 reference genome using HISAT2 v2.2.1 preserving strand information. Uniquely mapping reads were filtered after duplicate removal and filtering using *markDuplicates* from Picard v2.23.4 and Sambamba v0.7.1, respectively. To control the systematic variation across m⁶A RIP experiments, the amount of spiked-in bacterial RNA was estimated by counting the total number of reads uniquely mapped to the E. coli K-12 reference genome using Sambamba v0.7.1 (Tarasov et al, 2015). E. coli spike-in Bacterial counts were further used to calculate scaling factors for each batch of m⁶A RIP-seq samples. Computed scaling factors were then used to normalize the processed alignment files using DownsampleSam tool from Picard v2.23.4. m⁶A modifications were identified following peak calling with MACS2 v2.2.26 (Zhang et al, 2008) on m⁶A RIP and input processed alignments with parameters "--nomodel –bdg –extsize 75 –keep-auto –call-summits" and effective genome size $3.7 \times 10^8$. Called peaks were annotated according to their nearest genomic feature with *annotatePeaks.pl* from HOMER v4.11 (http://homer.ucsd.edu/homer/). Peaks per gene counts were calculated using custom scripts from the annotated peak files. Motif analysis on m⁶A modified peaks was carried out using *findMotifsGenome.pl* from HOMER.

## Metagene analysis

To analyze the genome-wide distribution of m⁶A, a metagene analysis of m⁶A peak density distribution was performed by overlapping the peak coordinates with the genomic features of 5'UTR, CDS, and 3'UTR plus 1 kb upstream and downstream coordinates obtained from GTF genome annotation files from

GENCODE v36, the longest isoform for each gene was considered. Each transcript was scaled to fixed-size metagene bins according to their respective reference genome coordinates. m⁶A peak density distribution profiles were generated after mapping the m⁶A peaks to the metagene coordinates using the plyranges R package (Lee Cook and Lawrence, 2019). In order to compare multiple conditions, the relative m⁶A density distributions were calculated using the relative density function from ggmulti package (https://cran.r-project.org/web/packages/ggmulti/index.html). The relative density function calculates the sum of the density estimate area of all conditions, where the total sum is scaled to a maximum of 1 and the area of each condition is proportional to its own count.

## Analysis of ChIP-seq data

Single-end METTL3, MYCN, and H3K27ac ChIP-seq data were processed using FastQC v0.11.9 and Trim Galore v0.6.6 for quality control. Trimmed reads were mapped to the GRCh38 reference genome using HISAT2 v2.2.1. Alignment files were further processed using markDuplicates from Picard v2.23.4 and Sambamba v0.7.1 to retrieve mapped reads after duplicate removal. Genome-wide peaks of METTL3, MYCN, and H3K27ac ChIP-seq datasets were called using MACS2 v2.2.26 (Zhang et al, 2008) with genome size parameters "-p 1e-5 –nomodel, keep-dup=auto, gsize=2.7e9" for METTL3 ChIP-seq data and "-p 1e-9 –nomodel keep-dup=auto, gsize=2.7e9" for MYCN and H3K27ac ChIP-seq datasets as reported in (Durbin et al, 2018; Xu et al, 2022), respectively. Genome-wide coverage tracks were further calculated using bamCoverage from deepTools v3.3.2 (Ramirez et al, 2016). Identified peaks were further annotated according to their nearest genomic feature using annotatePeaks.pl from HOMER v4.11 (http://homer.ucsd.edu/homer/).

To generate Venn diagrams representing the overlap of ChIP-seq peaks, the DiffBind R package was employed (Ross-Innes et al, 2012). The dba.plotVenn function within the DiffBind object, encompassing narrowPeak files along with the corresponding mask, facilitated the calculation of overlapping peaks displayed in the Venn diagram. For the visualization of binding patterns and comparative analysis of raw signals, the ChIPpeakAnno package in R was utilized (Zhu et al, 2010). Specifically, the featureAlignedHeatmap and featureAlignedDistribution functions were employed. To construct the heatmaps, the peaks obtained from the Venn diagram's overlapping regions were centered on the genomic coordinates of the MYCN obtained from the EnsDb.Hsapiens.v86 R package (Rainer, 2020). The peak widths were adjusted and recentred accordingly. Subsequently, the relevant datasets containing BigWig files were processed to create an RleList, which was then utilized to generate the featureAligned signal. The parameter upper.extreme was set to define the upper limit of the color scale, allowing precise control over the visualization of signal intensities. This signal encapsulated the intensity values corresponding to the processed peaks.

### Plot profile analysis

ChIP-seq data, represented as bigWig files, were obtained for the SHEP and tNCC. Additionally, BED files containing genomic coordinates of the regions of interest were taken. Analysis was conducted using deepTools version 3.3.2. deepTools was used to compute matrices of signal intensities around the reference point (TSS). For each set of regions, a matrix of signal intensities was computed using the computeMatrix function. The regions of interest were analyzed with a window extending 1000 base pairs upstream and downstream from the TSS. Signal intensity data from the ChIP-seq experiments, stored in the bigWig files, were integrated into the matrix computation process using the -S option in computeMatrix. Profile plots were generated using the plotProfile command. The plots give the summary of signal patterns around TSS for the regions of interest.

## Animal studies

For the in vivo experiments, we established tumor xenografts by injecting inducible control (shCtrl) or shMETTL3 SK-N-BE(2) cells subcutaneously into the right dorsal flank of 5-week-old female nude mice (Crl:NU(NCr)-Foxn1nu, Charles River) at a concentration of $5 \times 10^6$ cells in a 200 μL mixture of Matrigel and PBS (1:3 ratio, $n = 4$ per group). To induce METTL3 KD, doxycycline (2 mg/mL) and sucrose (2%) were added to the drinking water approximately 4-5 days after cell injection. We monitored the mice's weight weekly and measured tumor volume every 2–3 days using a digital caliper, calculated using the formula Volume (mm³) = (w² × l × π)/6, where 'w' represents the width (shortest diameter) and 'l' represents the length (longest diameter) of the tumor. Mice were euthanized either when the tumors reached 1000 mm³ in volume or if they experienced a weight loss of ≥10% of their initial weight. Upon conclusion of the experiment, we collected, weighed, and processed the tumors for subsequent analysis. All experiments were carried out as per the standards approved by the Institutional Ethical Committee of Animal Experimentation, Gothenburg, Sweden (ethical permit no. 3722/21).

For the drug combination experiments in vivo, NSG mice were obtained from Charles River and housed in groups of 2–5 mice per cage in individually ventilated cages with a 12 h light/dark cycle. All procedures were carried out under UK Home Office license P4DBEFF63 according to the Animals (Scientific Procedures) Act 1986 and were approved by the University of Cambridge Animal Welfare and Ethical Review Board (AWERB). COG-N-415x patient-derived xenograft (PDX) cells were obtained from the Childhood Cancer Repository maintained by the Children's Oncology Group (COG). Cells were suspended in Matrigel (Corning) diluted 1:2 with PBS, and $3 \times 10^5$ cells (300 μL) were injected into the left flank of NSG mice at an average of 8 weeks of age. Tumors were measured daily with manual calipers, and tumor volumes were estimated using the modified ellipsoid formula: V = ab2/2, where a and b (a > b) are length and width measurements, respectively. Once tumors reached ~170 mm³, mice were randomly allocated into four treatment groups ($n = 4$–6 per group, with the same number of females and males in each study) and treated for 14 days with the following agents by intraperitoneal injection at 10 μl/g body weight: vehicle (20% hydroxypropyl-beta cyclodextrin) daily, STM2457 (50 mg/kg in a vehicle) daily, doxorubicin (0.2 mg/kg in vehicle) every three days or a combination of STM2457 and doxorubicin at the same doses. Mice were euthanized at the end of treatment or once tumors reached 15 mm in any direction (what came first). The maximal tumor size permitted by our Project Licence (15–20 mm) was not exceeded in any of the studies. METTL3 inhibitor STM2457 used in this experiment was synthesized in-house (Appendix Supplementary Methods).

## Graphics

Some of the graphics were created with BioRender.com.

## Statistical analysis

All data were represented as mean ± SEM unless stated otherwise. Comparisons between two groups were performed using a two-tailed unpaired Student's *t* test, and comparisons among more than two groups were performed using one-way or two-way ANOVA (indicated in figure legends). *P* values less than 0.05 were considered statistically significant. The Interaction Factor package in ImageJ was used to randomize the METTL3 signal distribution to investigate the potential effects of random cluster overlap (Bermudez-Hernandez et al, 2017). All graphs were generated using GraphPad Prism software (version 10) or R with ggplot2 (Wickham, 2016).

# Data availability

The data supporting the findings of this article are accessible through the NCBI Gene Expression Omnibus (GEO) at https://www.ncbi.nlm.nih.gov/geo/. These data are associated with the GSE244473 accession number. (https://www.ncbi.nlm.nih.gov/geo/query/acc.cgi?acc=GSE244473). All data analysis was performed using customized packages available online. The code used is available upon reasonable request.

The source data of this paper are collected in the following database record: biostudies:S-SCDT-10_1038-S44318-024-00299-8.

# Peer review information

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

## Acknowledgements

This work was funded by grants from the Swedish Research Council (Vetenskapsrådet, 2018- 02224) and project grants from Barncancerfonden (PR2019-0077 and PR2023-0079), Cancerfonden (22-2341), Svenska Läkaresällskapet, Åke Wibergs Stiftelse, and Kungl Vetenskaps-och Vitterhets-Samhället (KVVS) grant to TM, 4 years research position grant from Barncancerfonden (TJ2019-0077) to TM, Tore Nilsons Stiftelse to RV and Assar Gabrielsson Fond to RV and KT. RV received a postdoctoral fellowship from Cancerfonden (230753PT). The authors would like to thank the core facility at Novum, BEA, Bioinformatics and Expression Analysis, which is supported by the board of research at the Karolinska Institute and the research committee at the Karolinska Hospital for help with sequencing. The computations/data handling were enabled under the project (NAISS2024-22-275 and NAISS2024-23-131) by resources provided by the National Academic Infrastructure for Supercomputing in Sweden (NAISS), partially funded by the Swedish Research Council through grant agreement no. 2022-06725.

## Author contributions

**Ketan Thombare**: Conceptualization; Resources; Data curation; Software; Formal analysis; Validation; Investigation; Visualization; Methodology; Writing—original draft; Project administration; Writing—review and editing.
**Roshan Vaid**: Conceptualization; Resources; Data curation; Software; Formal analysis; Supervision; Funding acquisition; Validation; Investigation; Visualization; Methodology; Writing—original draft; Project administration; Writing—review and editing. **Perla Pucci**: Resources; Data curation; Formal analysis; Validation; Visualization; Methodology; Writing—original draft; Writing—review and editing. **Kristina Ihrmark Lundberg**: Data curation; Formal analysis; Validation; Visualization; Methodology; Writing—review and editing. **Ritish Ayyalusamy**: Resources; Software; Formal analysis; Validation; Investigation; Visualization; Methodology; Writing—review and editing. **Mohammad Hassan Baig**: Resources; Methodology; Writing—review and editing. **Akram Mendez**: Resources; Data curation; Software; Formal analysis; Investigation; Visualization; Methodology; Writing—review and editing. **Rebeca Burgos-Panadero**: Resources; Data curation; Formal analysis; Validation; Investigation; Visualization; Methodology; Writing—review and editing. **Stefanie Höppner**: Data curation; Formal analysis; Validation; Visualization. **Christoph Bartenhagen**: Resources; Data curation; Software; Formal analysis; Visualization; Methodology; Writing—review and editing. **Daniel Sjövall**: Data curation; Formal analysis; Validation; Visualization; Methodology; Writing—review and editing. **Aqsa Ali Rehan**: Resources; Validation; Methodology. **Sagar Dattatraya Nale**: Resources; Methodology; Writing—review and editing. **Anna Djos**: Resources; Writing—review and editing. **Tommy Martinsson**: Resources; Writing—review and editing. **Pekka Jaako**: Data curation; Validation; Methodology; Writing—review and editing. **Jae-June Dong**: Resources; Methodology; Writing—review and editing. **Per Kogner**: Resources; Writing—review and editing. **John Inge Johnsen**: Data curation; Formal analysis; Validation; Visualization; Methodology; Writing—review and editing. **Matthias Fischer**: Resources; Validation; Investigation; Visualization; Methodology; Writing—original draft; Writing—review and editing. **Suzanne D Turner**: Resources; Data curation; Formal analysis; Validation; Investigation; Visualization; Methodology; Writing—original draft; Writing—review and editing. **Tanmoy Mondal**: Conceptualization; Resources; Data curation; Software; Formal analysis; Supervision; Funding acquisition; Validation; Investigation; Visualization; Methodology; Writing—original draft; Project administration; Writing—review and editing.

Source data underlying figure panels in this paper may have individual authorship assigned. Where available, figure panel/source data authorship is listed in the following database record: biostudies:S-SCDT-10_1038-S44318-024-00299-8.

## Funding

## Disclosure and competing interests statement

The authors declare no competing interests.

