## [Peer Review File · The EMBO Journal]

METTL3/MYCN cooperation drives neural crest differentiation and provides therapeutic vulnerability in neuroblastoma

Ketan Thombare, Roshan Vaid, Perla Pucci, Kristina Ihrmark Lundberg, Ritish Ayyalusamy, Mohammad Baig, Akram Mendez, Rebeca Burgos-Panadero, Stefanie Höppner, Christoph Bartenhagen, Daniel Sjövall, Aqsa Rehan, Sagar Dattatraya Nale, Anna Djos, Tommy Martinsson, Pekka Jaako, Jae-June Dong, Per Kogner, John Inge Johnsen, Matthias Fischer, Suzanne Turner, and Tanmoy Mondal

Corresponding author: Tanmoy Mondal (tanmoy.mondal@medkem.gu.se)

Review Timeline:

Submission Date:	18th Oct 23
Editorial Decision:	1st Dec 23
Appeal:	12th Dec 23
Editorial Decision:	21st Dec 23
Revision Received:	5th Jun 24
Editorial Decision:	25th Sep 24
Revision Received:	9th Oct 24
Accepted:	24th Oct 24

Editor: Daniel Klimmeck

Transaction Report:

Dear Dr Mondal,

Thank you again for submitting your manuscript (EMBOJ-2023-115931) for consideration by the EMBO Journal. Please accept my sincere apologies for the unusual protraction with the evaluation of your study, which is due to delayed referee feedback and detailed discussion in the editorial team. Your study has been sent to three referees with expertise in neuroscience and cancer and we have received reports from all of them, which I copy below. In light of their comments, I am afraid we decided that we cannot offer publication in The EMBO Journal.

As you will see the referees state that the results will as such be of interest to the field. However, i.p. reviewers #2 and #3 at the same time raise major concerns with the analysis that I am afraid preclude publication here. In more detail, referee #1 states that substantial ambiguities remain about the causal linkage between MYCN-METTL3 cooperation, m6A modifications and HOX gene activity in described setting (ref#1, standfirst). This expert also states that proof of target specificity for MYCN-METTL3 is not provided, and mechanistic insight into this connection remains too limited. Reviewer #2 agrees in that important claims are not sufficiently supported by the data, which dampens his-her enthusiasm for the findings (ref#2, standfirst, pts. 1,5,6,10). In addition, this expert has major concerns about the translational relevance of the results (ref#2, pt. 12).

Given these overall negative opinions from good experts in the field, and considering that we need strong support from the referees to move on, I am afraid we cannot offer to publish your study in The EMBO Journal.

I regret to not have more positive feedback for you and hope you will view the possibility of a transfer favourably. If this is the case, please use the link below to transfer the manuscript directly. I again apologise for the delay.

with
Kind regards,

Daniel Klimmeck

Daniel Klimmeck, PhD
Senior Editor
The EMBO Journal

Referee #1

(Report for Author)

In their manuscript, Thombare and colleagues explore the functions of the N6-methyladenosine (m6A) complex in neuroblastoma development. The authors identified a role of m6A in regulating HOX gene expression to allow differentiation into sympathetic neurons. Over-expression of MYCN in trunk neural crest cells (tNCC) leads to an undifferentiated cell state due to recruitment of enhanced deposition of m6A specifically into posterior HOX genes. Inhibition of m6A in both MYCN-overexpressing cells and neuroblastoma cells with MYCN amplifications induces differentiation and DNA damage.

Several recent studies demonstrated that all internal adenosines within consensus DRACH motifs are converted to m6A during transcription, unless they are close to a splice site, where exon junction complexes prevent methylation (He et al, 2023; Uzonyi et al, 2022; Yang et al, 2022). Thus, it is currently largely unknown how the core METTL3/METTL14-containing m6A modifying complex exhibits gene-specific regulatory functions. Unfortunately, the question of how the m6A complex is recruited to specific MYCN targets is not directly addressed in this manuscript. As described below in more detail, my main concerns are two-fold: First, the functional relevance of m6A in the HOX genes in the presence or absence of MYCN could simply reflect differentiation stages and therefore, it remains unclear whether loss of m6A in the HOX genes is cause or consequence of differentiation. Second, why HOXC8 and HOXC9 are particularly important and whether and how they interact to induce differentiation is not addressed.

Specific comments:

1. As a model system this study used tNCC derived from human embryonic stem cells (hESC). The tNCC can be further differentiated into sympathoadrenal progenitors (SAP) and sympathetic neuron (SN). This differentiation process correlated on with down-regulation of METLL3 and METTL14 on the protein level. The authors should also show the expression levels of the other complex members of the protein complex.
2. Why are HOXC8 and HOXC9 more strongly affected than other HOX posterior HOX genes? Is the distribution of DRACH motifs different across HOX genes?
3. Figure 3F,G: The down-regulation of HOXC8 and HOXC9 is not very convincing by IF. The Western blots should be shown as well and both differentiation stages (SAP and SN) should be shown. It is also unclear why and how efficiently the MYCN over-expressing cells can differentiate into SAPs and SNs.
4. Figure 4: Since m6A is co-transcriptionally installed nearly by default, provided the adenosine is located within a DRACH motif, it is not surprising that MYCN and METTL3 at least partially co-localize. To show that MYCN guides m6A deposition, the authors should test whether MYCN-targets are enriched in DRACH motifs. It is also unclear why MYCN needs to recruit the METTL3 to reduce HOX gene expression in undifferentiated cells. It seems that they should be repressed by default in these cells.
5. The finding that inhibition of m6A deposition using the small inhibitor STM2457 is sufficient to enhance differentiation in MYCN over-expressing or MYCN-amplified neuroblastoma cells is interesting. The authors should provide some information with regard to toxicity of the inhibitor and the combination of STM2457 and doxorubicin in some tested normal populations of tNCC, SAP and SN.

He PC, Wei J, Dou X, Harada BT, Zhang Z, Ge R, Liu C, Zhang LS, Yu X, Wang S et al (2023) Exon architecture controls mRNA m(6)A suppression and gene expression. *Science*: eabj9090

Uzonyi A, Dierks D, Nir R, Kwon OS, Toth U, Barbosa I, Burel C, Brandis A, Rossmannith W, Le Hir H et al (2022) Exclusion of m6A from splice-site proximal regions by the exon junction complex dictates m6A topologies and mRNA stability. *Mol Cell*

Yang X, Triboulet R, Liu Q, Sendinc E, Gregory RI (2022) Exon junction complex shapes the m(6)A epitranscriptome. *Nat Commun* 13: 7904

Referee #2

(Report for Author)

The paper by Thombare and colleagues investigates the impact of the m6A modification on neuroblastoma, the most common extracranial childhood cancer. Focusing on the METTL3 methyltransferase m6A writer, the study suggests that m6A modification plays a crucial role in regulating the expression of HOX genes in neural crest cells, influencing their differentiation into sympathetic neurons. Furthermore, the authors suggest a potential therapeutic approach for neuroblastoma, particularly in cases with MYCN amplification/overexpression, as inhibiting/knocking down METTL3 induces differentiation, and may increase the vulnerability to chemotherapy. Finally, the authors suggest that MYCN and METTL3 physically interact to methylate MYCN targets and control HOX transcript stability via m6A. Overall, the findings have the potential to advance our understanding of the molecular mechanisms in neuroblastoma and the authors propose a possible avenue for therapeutic interventions. However, there are multiple shortcomings in this manuscript that need to be addressed. They are as follows:

- 1) The authors write: "Overall, higher expression of METTL3/14, differential gene expression along with m6A RIP- seq data in hESC and tNCC suggest a critical role for m6A in tNCC." However, based on the levels of the m6A writers and number of m6A peaks, one cannot suggest a critical role for m6A - please tone down this statement.
- 2) Regarding Figure 1D - please show the P value for this comparison. Furthermore, while the authors suggest that the number

of m6A peaks is higher in tNCC compared to hESCs, they do not clearly state whether this can be simply explained by a higher number of peaks in the same/overlapping transcripts or in fact some transcripts that are not m6A modified in hESC are modified in tNCCs (or other way round) upon lineage commitment. Please present a Venn diagram for m6A-modified transcripts in hESC and tNCCs - I cannot find these data in the current manuscript.

3) The experiments presented throughout the manuscript are preformed using only one shRNA. Given that key claims are being made based on METTL3 knockdown data, it would be important to preform the key experiments using another independent hairpin.

4) In Figure 2B, the authors suggest that HOXC8/9 are downregulated in tumours with MYCN amplifications. It would be important to test whether or not the HOXC8 and HOXC9 protein levels are also affected.

5) In Figure 2 the authors go on to perform RNA-seq in SK-N-BE cells upon METTL3 knockdown as well as m6A-seq in SK-N-BE cells. However, they do not provide any global analyses of the data. It would be important to show graphs/plots depicting the expression change between METTL3 KD against control, for non-modified and m6A-modified transcripts. Is there a statistically significant upregulation of m6A-modified transcripts upon METTL3 KD? Is there a difference in expression of non-modified transcripts upon METTL3 KD?

6) Is global m6A-transcript stability and translational efficiency affected upon METTL3 depletion in SK-N-BE cells? It would be important to perform SLAM-seq and RIBO-seq experiments to address this.

7) The authors state: "In particular, we observed the increased stability of HOXC8 and HOXC9 mRNA following METTL3 KD in SK-N-BE(2) cells (Fig. 2G)". The stability of HOXC9 is very marginally affected and the conclusions need to be toned down.

8) It is suggested that upregulation of HOXC8 and HOXC9 may mediate differentiation of SK-N-BE cells upon METTL3 knockdown. Can knockdown of HOXC8 and/or HOXC9 rescue the phenotype resulting from METTL3 knockdown?

9) The data shown in Figure 3F-G appear to have huge error bars. How can these differences be statistically significant? Please explain how this was achieved? The same comment pertains to results shown in Figure 5E.

10) The data presented in Figure 4 suggest that MYCN and METTL3 physically interact- please validate this interaction by performing immunoprecipitation experiments. Can MYCN pull down METTL3, and vice versa?

11) They conclude that: "...METTL3 inhibitors may represent efficacious therapeutic agents in the treatment of NB". However, in Figure 5 the authors use unusually high concentration of METTL3 inhibitor (i.e. 10 microM). Why? It is much higher compared to the concentration that compromises AML cells (Nature 593, 597-601; 2021).

12) The authors suggest that METTL3 inhibition combined with chemotherapy may be a therapeutic strategy in neuroblastoma. However, METTL3 inactivation alone causes hematopoietic stem cell failure and doxorubicin is toxic too. If there is a synergistic interaction between METTL3 inhibition and doxorubicin, as the authors suggest, then such treatment is highly likely to have deleterious consequences to normal tissues/organs. What is the therapeutic window for such treatment? It would be important to test this in normal (non-cancer) counterparts of neuroblastoma cells. It would also be important to perform a pilot study with METTL3 inhibition + doxorubicin to investigate whether this impacts on mouse health and survival.

Referee #3

(Report for Author)

Overall this manuscript is well written, presents novel mechanistic data and in my opinion is suitable for publication in EMBO.

Minor comments:

Suppl 1J-Q - the role of metal KD in hESC & NC - this feels like important data that is very relevant and should not be buried in the supplementary?

Suppl Fig2C - did METTL3 KD in control cell line (step without MYCN) have any effect on viability? This data should also be shown

Fig 4C/supplementary figure 4 - pathway analysis of trunk NC genes that are co-bound by MYCN and METTL3 is very limited. Could this be expanded on?

4G/H - changes in m6A enrichment and RNA expression are shown in shep MYCN dox cells but not shep controls. Do you see the same pattern in non MYCN expressing shep?

** As a service to authors, EMBO Press provides authors with the possibility to transfer a manuscript that one journal cannot offer to publish to another EMBO publication or the open access journal Life Science Alliance launched in partnership between EMBO Press, Rockefeller University Press and Cold Spring Harbor Laboratory Press. The full manuscript and if applicable, reviewers' reports, are automatically sent to the receiving journal to allow for fast handling and a prompt decision on your manuscript. For more details of this service, and to transfer your manuscript please click on Link Not Available. **

To
The Editor,
The EMBO Journal

Dear Dr. Klimmeck,

Thank you for sending us the comments and decision on our manuscript (EMBOJ-2023-115931). My apologies for the delayed response to your decision email. The delay was primarily due to our thorough review of the comments before formulating our reply.

We are pleased to learn that all three reviewers found our paper relevant, and they believe that the results will be of considerable interest to the field. Reviewer 3 has deemed our paper acceptable for publication in EMBO, while the other two reviewers provided valuable criticism and suggestions. We believe these comments are addressable and will enhance the overall quality of our manuscript.

We appreciate your prior discussion with EMBO sister journal LSA Editor Dr. Eric Sawey regarding the possibility of transferring the manuscript to LSA. However, we kindly ask for your reconsideration of our manuscript, highlighting the following aspects that we believe merit a second evaluation of our submission.

I. Reviewer 1 acknowledged that we have addressed an important question in the field: how "the core METTL3/METTL14-containing m⁶A modifying complex exhibits gene-specific regulatory functions." While our data suggest that the oncogenic transcription factor MYCN could guide the METTL3/14-containing m⁶A complex to specific genes, Reviewer 1 noted that the experimental evidence provided was not sufficient to substantiate our claim. We appreciate these constructive comments and agree with the reviewer. We are actively working on addressing these concerns and are confident that we can strengthen our manuscript to corroborate our claims.

II. Reviewer 2, like Reviewer 1, provided constructive and well-directed questions that, if addressed, will significantly improve the quality of our manuscript. Reviewer 2 recognized the potential of our study to "advance our understanding of the molecular mechanisms in neuroblastoma" and highlighted the prospect of "a possible avenue for therapeutic interventions." However, concerns were raised about certain approaches, and a more rigorous validation of the therapeutic aspects was recommended. We are currently collaborating to further validate our observations on METTL3 inhibition-mediated therapeutic approaches using in vivo experimental model of neuroblastoma.

III. We have meticulously reviewed all major and minor comments provided by the three reviewers, and we are confident in our ability to address them comprehensively.

Overall, since the reviewers' comments are progressive and within the scope of our expertise, we kindly request you to reconsider your decision. We believe that revising our manuscript will enhance the appeal of our observations to the wider scientific community in both the fields of epitranscriptomics and neuroblastoma.

We look forward to your response.

Best regards,
Tanmoy Mondal

Dear Dr Mondal,

Thank you for your letter concerning our decision on your manuscript EMBOJ-2023-115931, and re-emphasizing the value of your findings for the community. Please accept my apologies for following up with protraction due to detailed internal discussions and other tasks in the office related to new primary submissions. I have now carefully assessed your rebuttal and discussed these in detail with the editorial team. I am afraid that considering all information at hand, we have concluded to maintain our decision that we cannot pursue your manuscript further at the EMBO Journal.

As detailed in our decision letter the referees raised major critique on both mechanistic as well as applied aspects of your study, which made us conclude that the analysis is overall too preliminary for what we need to expect for continued consideration at the EMBO Journal. I see from your current letter that you are in the process of addressing the major issues raised. However, since the outcome of these experiments is entirely open and in the absence of a more detailed annotation of your revision, I am afraid we cannot commit to proceeding with consideration of your study at the EMBO Journal.

I can offer to assess a fully revised version of the manuscript editorially, should that be of your interest, but again cannot commit to the outcome of such reevaluation.

I regret not to have better feedback for you. I appreciate that you approached us further regarding this decision and again apologize for the delay.

Kind regards,

Daniel Klimmeck

Daniel Klimmeck, PhD
Senior Editor, The EMBO Journal

** As a service to authors, EMBO Press provides authors with the possibility to transfer a manuscript that one journal cannot offer to publish to another EMBO publication or the open access journal Life Science Alliance launched in partnership between EMBO Press, Rockefeller University Press and Cold Spring Harbor Laboratory Press. The full manuscript and if applicable, reviewers' reports, are automatically sent to the receiving journal to allow for fast handling and a prompt decision on your manuscript. For more details of this service, and to transfer your manuscript please click on Link Not Available. **

Point-by-point answer to the Reviewers' comments.

We thank the Reviewers and the Editor for providing valuable suggestions on our manuscript. We have addressed these comments comprehensively in this file, as well as in the main text and added figures. Please find below the reviewers' comments (**in black**) and our point-by-point responses (**in blue**). In the rebuttal section, we have included data from all the newly performed experiments conducted during the revision.

Referee #1

(Report for Author)

In their manuscript, Thombare and colleagues explore the functions of the N6-methyladenosine (m6A) complex in neuroblastoma development. The authors identified a role of m6A in regulating HOX gene expression to allow differentiation into sympathetic neurons. Overexpression of MYCN in trunk neural crest cells (tNCC) leads to an undifferentiated cell state due to recruitment of enhanced deposition of m6A specifically into posterior HOX genes. Inhibition of m6A in both MYCN-overexpressing cells and neuroblastoma cells with MYCN amplifications induces differentiation and DNA damage.

Several recent studies demonstrated that all internal adenosines within consensus DRACH motifs are converted to m6A during transcription, unless they are close to a splice site, where exon junction complexes prevent methylation (He et al, 2023; Uzonyi et al, 2022; Yang et al, 2022). Thus, it is currently largely unknown how the core METTL3/METTL14-containing m6A modifying complex exhibits gene-specific regulatory functions. Unfortunately, the question of how the m6A complex is recruited to specific MYCN targets is not directly addressed in this manuscript. As described below in more detail, my main concerns are two-fold:

We sincerely appreciate the reviewer for highlighting several crucial aspects, questions, and criticisms in our manuscript. The feedback provided has significantly aided us in enhancing our work. In the following sections, we have meticulously addressed each concern raised by the reviewer, and we have detailed them below.

First, the functional relevance of m⁶A in the HOX genes in the presence or absence of MYCN could simply reflect differentiation stages and therefore, it remains unclear whether loss of m⁶A in the HOX genes is cause or consequence of differentiation.

Answer: We thank the reviewer for raising this significant and intriguing question. In response to the query, we have employed several approaches outlined below:

I) To investigate whether the loss of m⁶A methylation in *HOX* genes, particularly *HOXC9*, is a consequence of differentiation or if the absence of m⁶A has a causal effect on differentiation, we targeted the m⁶A demethylase FTO to the m⁶A site of the *HOXC9* transcript using gRNAs. Our observations revealed the removal of m⁶A from *HOXC9*, leading to increased *HOXC9* expression and induced differentiation of MYCN-amplified SK-N-BE(2) cells. This data suggests that the specific loss of m⁶A in *HOXC9* plays a causal role in differentiation, which has been integrated into the revised manuscript as **Figure 5C and D** (attached below for quick perusal).

Figure for reviewers removed; see Figure 5 in the manuscript.

Figure 5 from the revised manuscript. (C) Illustration describing the recruitment of dCasRx-FTO at target RNA. Representative IF showing expression of PRPH (green) and TUBB3 (red) in SK-N-BE(2) cells expressing dCasRx-FTO Mutant (catalytically dead-H231A and D233A mutant)/dCasRx-FTO WT (wild-type) with either non-template control (NTC gRNA) or HOXC9 guide RNAs (HOXC9 gRNA-1, HOXC9 gRNA-2). Dox induction was performed for 72 h followed by 3 days RA mediated differentiation in the presence of Dox. Box-whisker plot shows neurite length. Data are from three independent experiments. Two-way ANOVA with Šidák's multiple comparisons test was used (**** $p < 0.0001$, ns- nonsignificant). Scale bar represents 50 μm . (D) Representative IF showing expression of HOXC9 (green) in SK-N-BE(2) cells in the same condition as detailed above in (C). Box-whisker plot shows HOXC9 intensity normalized with DAPI intensity. Signal intensity measurements were taken from over 1000 cells. Data are from three independent experiments. Two-way ANOVA with Šidák's multiple comparisons test was used (** $p < 0.01$, **** $p < 0.0001$, ns- nonsignificant). Scale bar represents 50 μm .

II) We also observed MYCN-mediated METTL3 recruitment using SHEP^{MYCN} cells. SHEP cells are mesenchymal type NB cells and are resistant to differentiation signals such as the presence of retinoic acid (PMID: 26245651). So, such a system helps uncouple the MYCN/METTL3-mediated effect of m⁶A methylation on HOX genes from the MYCN-mediated differentiation effect. Using ChIP-seq we have shown evidence that in SHEP cells MYCN could recruit METTL3 over HOX genes (**Supplementary Figure S4I**). In the revised version we provide further evidence that recruitment of METTL3 by MYCN induces m⁶A methylation in *HOXC8* and *HOXC9* by RIP-seq (**Figure 4I**). We have validated higher m⁶A methylation over posterior *HOXC* genes by RIP-qPCR as well (**Figure 4I** in the revised manuscript, attached below). This data argues that MYCN and METTL3 crosstalk-mediated m⁶A modification is independent of differentiation.

Figure for reviewers removed; see Figure 4 in the manuscript.

Figure 4 from the revised manuscript. (I) Left: Browser screenshot showing m^6A RIP-seq tracks at 3'UTR of *HOXC8* and *HOXC9* genes in *SHEP^{MYCN}* cells before and after Dox induction for 24 h. Right: m^6A RIP-qPCR data showing enrichment of both *HOXC8* and *HOXC9* in *SHEP^{MYCN}* cells before and after Dox induction for 24 h. Data are represented as a percentage of input. IgG was used as a negative control (n=3). Two-tailed paired *t*-test was used (***p* < 0.001, *****p* < 0.0001).

III) In MYCN-overexpressing SHEP cells, we show that 30% of METTL3/MYCN co-bound genes are positive for m^6A (**Figure 4L**). We observe that METTL3/MYCN co-bound genes have a higher level of METTL3 enrichment (**Figure 4K**). Furthermore, METTL3/MYCN co-bound m^6A positive genes exhibit a higher m^6A density at the 3' UTR compared to other m^6A positive genes (**Figure 4M**). The number of m^6A peaks/genes is higher in METTL3/MYCN co-bound genes compared to the rest of m^6A positive genes (**Figure 4N**). This data suggests that MYCN-mediated METTL3 recruitment could promote m^6A modification not only at the level of *HOX* genes but globally.

Figure for reviewers removed; see Figure 4 in the manuscript.

Figure 4 from the revised manuscript. (K) Left: Distribution of METTL3 ChIP signal in a metagene profile. The data is centered at the transcription start site (TSS) [-1 kb to +1 kb], at genes that are co-bound by METTL3 and MYCN or bound by METTL3 only in *SHEP^{MYCN}* cells after 24 h Dox induction. **(L)** Venn diagram comparing m^6A+ and METTL3-MYCN co-bound genes *SHEP^{MYCN}* cells after Dox 24h induction. METTL3 and MYCN co-bound regions were determined using the ChIP-seq experiments. **(M)** Metagene analysis showing relative m^6A peak density at genes co-bound by METTL3 and MYCN or the rest of m^6A containing genes in *SHEP^{MYCN}* cells after Dox induction. **(N)** Box-whisker plot showing the number of m^6A peaks/genes that are co-bound by METTL3

and MYCN (median =2) or the rest of m⁶A containing genes (median =1) in SHEP^{MYCN} cells after Dox induction. Whiskers indicate the 1st to 99th percentiles, and the 1% of outliers are shown as individual dots. Statistical analysis was performed using the Wilcoxon test (** $p < 0.01$).

In the revised manuscript, we also show that MYCN overexpression in SHEP cells promotes a higher interaction between METTL14 and H3K36me3. H3K36me3 is generally present on the gene body regions of active genes, and the interaction between METTL14 and H3K36me3 has been proposed as a mechanism that guides the m⁶A modifying machinery to perform co-transcriptional m⁶A modification (PMID: 30867593, Huilin Huang et al, Nature; 2019). Our data suggest that MYCN, by recruiting METTL3 to gene promoters, enhances the interaction of the m⁶A modifying machinery with H3K36me3, as evidenced by the increased H3K36me3 and METTL14 PLA signal (**Figure 4O** in the revised manuscript, attached below).

Figure for reviewers removed; see Figure 4 in the manuscript.

Figure 4 from the revised manuscript. (O) PLA in SHEP^{MYCN} cells with or without Dox induction for 24 h depicting METTL14 and H3K36me3 PLA signal (green) in the nucleus (marked by DAPI). The negative

control shows PLA with only the H3K36me3 antibody. Signal intensity measurements were taken from over 50 cells. Data are presented as box-whisker plot from three independent experiments. Statistical analysis was performed using an unpaired t-test (**** $p < 0.0001$). Scale bar represents 10 μ m.

To summarize, our data suggest that MYCN-mediated guidance of METTL3 could promote m⁶A methylation on MYCN/METTL3 co-bound genes. This is consistent with earlier studies, such as those involving the transcription factor SMAD2 (PMID: 29489750, Bertero A et al, Nature 2018), which can recruit METTL3 in a manner similar to MYCN. Our study indicates that lineage-specific transcription factors play a crucial role in METTL3 recruitment in a gene-specific manner. We provide evidence that MYCN-mediated METTL3 recruitment enhances interaction with H3K36me3 (as described above) to facilitate co-transcriptional m⁶A deposition.

However, once METTL3 is recruited, the next step involves determining how the m⁶A modifying machinery selects which DRACH-like motifs to methylate (only 5% of DRACH sequences are methylated), given that the number of DRACH motifs exceeds the identified m⁶A sites (PMID: 36705538, He P et al, 2023 Science). The elegant studies highlighted by the reviewer (PMID: 36705538, He P et al, 2023 Science; PMID: 36599352, Uzonyi et al, 2022, Mol Cell; and PMID: 36550132, Yang et al, 2022, Nature Communications) address this question of m⁶A site selection. They suggest that the m⁶A motif located close to the splice site

is protected from methylation by exon junction complexes, which block the association of METTL3 with RNA (Yang et al, 2022, Nature Communications), thus providing a mechanism for the selective methylation of m⁶A sites out of the many motifs typically present in the transcript.

Overall, we believe we have addressed the reviewer's concerns regarding MYCN-mediated METTL3 recruitment in the revised manuscript. We also think we have effectively integrated our findings with existing knowledge on the mechanism of METTL3 recruitment and m⁶A site selection in a gene-specific manner. We have discussed these points (indicated in the discussion section of the revised manuscript marked with red) in light of the reviewer's comments and based on data obtained from newly performed experiments.

Second, why HOXC8 and HOXC9 are particularly important and whether and how they interact to induce differentiation is not addressed.

Answer: This is an interesting point raised by the reviewer. We have addressed this issue in the revised manuscript using the following approach:

I) We observed that the promoters of deregulated genes following MYCN overexpression in SAP were enriched with HOXC9 motifs, suggesting that the down-regulation of HOXC9 expression could be critical in the undifferentiated phenotype observed in MYCN-overexpressing cells (**Figure 5A** in the revised manuscript, attached below). The down-regulation of HOXC9 expression in MYCN induced SAP was detected using RNA-seq (**Figure 3H**), RT-qPCR (**Figure 3H** in the revised manuscript, attached below), immunoblot (Supplementary Figure 3A), and IF (**Figure 3G**). To further address the importance of HOXC9, we performed HOXC9 overexpression in MYCN-SAP cells and observed that overexpression of HOXC9 could rescue the MYCN-mediated undifferentiated state (**Figure 5B** in the revised manuscript, attached below). This confirms that HOXC9 expression is critical in the tNCC-SAP-SN differentiation steps as described in our study.

Figure for reviewers removed; see Figure 5 in the manuscript.

Figure 5 from the revised manuscript. (A) Top 10 transcription factor binding motifs enriched in the promoter region of the DEGs (Flag- MYCN overexpressed Dox- vs. Dox+) in SAP. (B) IF showing expression of PRPH (green) and TUBB3 (red) in Flag-MYCN overexpressed (Dox+, from day 5 onwards) SN stage cells with either control (Ctrl) or HOXC9 overexpression (OE). HOXC9 was OE was performed from day 9 of differentiation. Box-whisker plots show the quantification of the neurite length, TUBB3, and PRPH intensity. Scale bar represents 100 μ m. Experiments were performed in three independent biological replicates. Unpaired t-test was used (** $p < 0.001$, **** $p < 0.0001$).

Figure for reviewers removed;
see Figure 3 in the manuscript.

Figure 3 from the revised manuscript. (H) Left: Differentially expressed posterior HOXC genes between control (Dox-) and Flag-MYCN overexpression (Dox +, from day 5 onwards) in SAP. The expression values were determined from RNA-seq data. Right: Relative mRNA expression of HOXC8 and HOXC9 in SAP following Flag- MYCN overexpression (Dox +, from day 5 onwards) and in control (Dox-). GAPDH was used to normalize the qPCR data. Data are shown as mean \pm SD of three replicates. Statistical analysis was performed using an unpaired t-test, ** $p < 0.01$, *** $p < 0.001$.

II) We observed that METTL3 knockdown (KD) in MNA SK-N-BE(2) NB cells could enhance the differentiation of these cells (**Figure 2J**). We wanted to determine if this phenotype in METTL3 KD cells is due to the upregulation of HOXC9 expression. To test this, we depleted HOXC9 by shRNA in the METTL3 KD cells and observed a reversal of the differentiation phenotype. This data again points to the critical role of HOXC9 in the differentiation of MNA SK-N-BE(2) NB cells (**Figure 5E and F** in the revised manuscript, please see **page 23** of rebuttal). This important point was also raised by reviewer 2 as well.

III) In our earlier submitted version, we presented data suggesting that a significant number of genes deregulated following METTL3 KD are HOXC9 targets (20%) (**Supplementary Figure S5E**). These common genes (deregulated by METTL3 KD and HOXC9 targets) are enriched in pathways related to neuronal differentiation (**Supplementary Figure S5E**).

IV) We also presented PLA data indicating that HOXC8 and HOXC9 could interact with each other and, when overexpressed together in MYCN-amplified SK-N-BE(2) cells, could synergistically induce differentiation (**Supplementary Figure S5F-H**). In the revised version, we performed further ChIP for HOXC8 and HOXC9 following their overexpression and found that they co-occupy similar target genes with known roles in neuronal differentiation, such as PRPH, SEMA3D, and NRCAM (**Supplementary Figure S5I** in the revised manuscript, attached below).

Figure for reviewers removed; see Appendix Figure S5

Supplementary Figure S5 from the revised manuscript. (I) Myc-tag ChIP qPCR data, represented as percentage input over selected genes in SK-N-BE(2) overexpressing either Myc-tagged HOXC8, HOXC9, or mock. Data are shown as mean \pm SD from three independent biological replicates. Statistical analysis was performed using two-way ANOVA with Dunnett's

multiple comparisons test ($p < 0.05$, ** $p < 0.01$).*

Altogether, we believe we have addressed the issues regarding the importance of posterior *HOX* genes in the differentiation of MYCN-overexpressing cells.

Specific comments:

1. As a model system this study used tNCC derived from human embryonic stem cells (hESC). The tNCC can be further differentiated into sympathoadrenal progenitors (SAP) and sympathetic neuron (SN). This differentiation process correlated on with down-regulation of METTL3 and METTL14 on the protein level. The authors should also show the expression levels of the other complex members of the protein complex.

Answer: We have now included additional data on WTAP and RBM15, which are part of the METTL3/METTL14 complex. We observe that their expression patterns are similar to those of METTL3 and METTL14 (**Figure 1C**).

2. Why are HOXC8 and HOXC9 more strongly affected than other posterior HOX genes? Is the distribution of DRACH motifs different across HOX genes?

Answer: Considering the reviewer's suggestion, we have investigated the number of DRACH-like motifs in the *HOXC* gene cluster, including *HOXC8* and *HOXC9*. We did not observe any difference in the number of DRACH motifs present over *HOXC* genes, such as *HOXC8* and *HOXC9*, suggesting that the presence of a higher number of DRACH motifs might not be the reason for the differential presence of m⁶A peaks over these genes (data provided below for the reviewer's perusal). This observation aligns with the publication mentioned by the reviewer, where m⁶A sites are typically more abundant than the detected m⁶A peaks in a transcript. The exon junction complex provides a selection mechanism for m⁶A site methylation (PMID: 36705538, He P et al., 2023 Science; PMID: 36599352, Uzonyi et al., 2022, Mol Cell; and PMID: 36550132, Yang et al., 2022, Nature Communications).

Therefore, we argue that the number of m⁶A sites in the HOX transcript may not be the limiting factor for having m⁶A peaks. Instead, our data indicate that MYCN-mediated METTL3 recruitment could be a critical factor for m⁶A modification in *HOXC8* and *HOXC9* genes. This has been emphasized using data obtained from both tNCC and MYCN-overexpressing SHEP cells (**Figure 4F and 4I; Supplementary Figure 4I**).

3. Figure 3F,G: The down-regulation of *HOXC8* and *HOXC9* is not very convincing by IF. The Western blots should be shown as well and both differentiation stages (SAP and SN) should be

shown. It is also unclear why and how efficiently the MYCN over-expressing cells can differentiate into SAPs and SNs.

Answer: Considering the reviewer's suggestion, we have now performed a western blot on MYCN-overexpressing SAP cells, and we observed decreased expression of HOXC9 (**Supplementary Figure S3A** in the revised manuscript please see attached below). Additionally we have verified decreased expression of HOXC9 in MNA NB tumors by western blot (**Figure 2C** in the revised manuscript please see **page 19** of rebuttal)

We attempted to use several antibodies for HOXC8, but we were unable to obtain the correct sized band for HOXC8 in the western blot, so we did not include it. However, the downregulation of both HOXC8 and HOXC9 following MYCN overexpression in SAP cells was detected by RT-qPCR in the earlier submitted version (**Figure 3H**, in the revised manuscript). Additionally, during the revision, we performed RNA-seq in the MYCN overexpressing SAP cells, which again showed downregulation of HOXC8 and HOXC9 at the RNA level (**Figure 3H**). In the revised version, we show the upregulation of HOXC9 expression following METTL3 knockdown (KD) in SAP cells in the western blot as well (**Supplementary Figure S10** in the revised manuscript please see attached below). Upregulation of HOXC8 and HOXC9 in the METTL3 KD cells is also detected at the RT-qPCR level (**Supplementary Figure S1L**).

Figure for reviewers removed; see Appendix Figure S3 in the manuscript.

Supplementary Figure S3 from the revised manuscript. (A) Representative immunoblot showing expression of HOXC9 in SAP following Flag- MYCN overexpression (Dox+, from day 5 onwards) and in control (Dox-). Vinculin was used as a loading control. The values below indicate the fold change in levels of HOXC9. The experiments were repeated three times.

Figure for reviewers removed; see Appendix Figure S1 in the manuscript.

Supplementary Figure S1 from the revised manuscript. (O) Representative immunoblot shows the levels of HOXC9, in SAP with either TetO shCtrl or TetO shM3-2. Vinculin was loading control. The values below the blots indicate the fold change (normalized to loading control) in the levels of HOXC9.

We described in the earlier submitted version of the manuscript that at the SN stage, the expression of HOXC9 is already very low (**Supplementary Figure S6E**, in the revised version), which is why we did not perform any western blots at this stage from MYCN-

overexpressing cells. As described above, although our RNA-seq data showed downregulation of HOXC9 at the SAP stage in MYCN-overexpressing cells (**Figure 3H**), we did not observe any change in already lowly expressed HOXC9 at the SN stage.

We appreciate the reviewer's insight, and we acknowledge that the IF data presented regarding HOXC8/9 expression in the earlier submitted version could be improved to more convincingly show the downregulation of these proteins. Reviewer 2 also provided valuable feedback on the IF data. We realized that not normalizing the HOXC8/9 intensity with DAPI caused higher variation. The plotted data was obtained from a large number of cells (in the range of 1000), and the variability in expression between individual cells contributed to large error bars in the bar plots we presented earlier. We also realized that plotting the data in a bar plot did not provide a clear visualization of the range of data points obtained from a large number of cells. In the revised version, we have provided DAPI-normalized data for the HOXC8/9 intensity and presented the data as box-whisker plots, where the data ranges are visible. We have implemented this change throughout the manuscript for consistency in data presentation. We believe the reviewer will find these new plots convincing, as they show statistically significant differences in HOXC8/9 expression in both MYCN overexpression (**Figures 3F and 3G**) and METTL3 KD conditions (**Supplementary Figure S10**).

Regarding the second part of question 3 on “how efficiently the MYCN overexpressing cells can differentiate,” in the revised manuscript, we performed RNA-seq in MYCN-overexpressing SAP cells to check if critical SAP marker genes were differentially expressed following MYCN expression. We observed differential expression of several known SAP genes, such as *ASCL1* and *ISL1* (**Supplementary Figure S3C**), suggesting that MYCN overexpression perturbs the core SAP gene expression signature. This is consistent with the lack of differentiation of these MYCN-overexpressing cells, as indicated by RNA-seq data from MYCN-overexpressing SAP and SN cells, which suggest that genes related to neuronal differentiation were downregulated following MYCN overexpression (**Supplementary Figure S3D and S3E**).

4. Figure 4: Since m6A is co-transcriptionally installed nearly by default, provided the adenosine is located within a DRACH motif, it is not surprising that MYCN and METTL3 at least partially co-localize. To show that MYCN guides m6A deposition, the authors should test whether MYCN-targets are enriched in DRACH motifs.

It is also unclear why MYCN needs to recruit the METTL3 to reduce HOX gene expression in undifferentiated cells. It seems that they should be repressed by default in these cells.

We thank the reviewer again for bringing up these interesting questions. We have tried to address both issues below in two sections (1 & 2).

Section 1:

As described above (and briefly reiterated here for the reviewer's perusal in the context of question 4), we show that METTL3/MYCN co-bound m⁶A positive genes exhibit higher m⁶A density at the 3' UTR compared to other m⁶A positive genes (**Figure 4M**). Additionally, m⁶A peaks/genes were higher in METTL3/MYCN co-bound m⁶A genes compared to the rest of the m⁶A positive genes (**Figure 4N**). In both tNCC and MYCN overexpressed SHEP cells, MYCN/METTL3 co-bound genes showed higher levels of METTL3 enrichment at the promoter compared to genes bound only by METTL3 (**Figure 4N**). This data suggests that MYCN-mediated METTL3 recruitment could promote m⁶A modification (**Figures 4K-N**).

In the revised manuscript, we also show that MYCN overexpression in SHEP cells promotes higher interaction between METTL14 and H3K36me3. H3K36me3 is generally present in the gene body regions of active genes, and interaction between METTL14 and H3K36me3 has been proposed as a mechanism guiding the m⁶A modifying machinery to carry out co-transcriptional m⁶A modification (PMID: 30867593, Huilin Huang et al., Nature, 2019). Our data suggest that MYCN recruits METTL3 to the gene promoter, enhancing the interaction of the m⁶A modifying machinery with H3K36me3, as visualized by enhanced H3K36me3 and METTL14 PLA signals (**Figure 4O**). Our data are consistent with earlier studies where a similar mechanism has been proposed for other transcription factors such as SMAD2 (PMID: 29489750, Bertero A et al., Nature, 2018), which could recruit METTL3 in a manner comparable to MYCN. Overall, we show that lineage-specific transcription factors like MYCN play an important role in METTL3 recruitment in a gene-specific manner to facilitate m⁶A modification.

We also tested the possibility of METTL3 and MYCN interaction as a consequence of the co-transcriptional process. To test this, we blocked transcription elongation with Flavopiridol (FP) and performed METTL3 and MYCN co-immunoprecipitation. We observed efficient blocking of transcription following FP treatment in MYCN-overexpressing SHEP cells, but this did not affect the MYCN and METTL3 interaction as detected by co-immunoprecipitation assay

(**Figure 4B** and **Supplementary Figure S4B** in the revised manuscript and attached below). We further tested by ChIP-qPCR if METTL3 and MYCN recruitment were altered following transcription block by FP treatment and found that METTL3 and MYCN recruitment was unaltered on selected genes, including HOXC8/9 (**Supplementary Figure S4J** in the revised manuscript and attached below). These data suggest that MYCN-mediated initial recruitment of METTL3 to gene promoters is independent of the co-transcriptional process.

Figure for reviewers removed; see Figure 4 in the manuscript.

Figure 4 from the revised manuscript. (B) Co-IP of METTL3 or MYCN from lysates of SHEP^{MYCN} after Dox induction for 24 h and treatment with 300 nM flavopiridol (FP) for 1 h, blotted with MYCN or METTL3 antibodies. IgG served as a negative control.

Figure for reviewers removed; see Appendix Figure S4 in the manuscript.

Supplementary Figure S4 from the revised manuscript. (J) MYCN, METTL3, IgG ChIP-qPCR data, represented as percentage input over selected genes in Dox induced SHEP^{MYCN} cells treated with or without FP. Data are shown as mean \pm SD from three independent biological replicates. Unpaired t-test was

used (ns- nonsignificant).

Next, considering the reviewer's suggestion, we checked the number of DRACH motifs present in MYCN/METTL3 co-bound m⁶A positive genes versus the rest of the m⁶A genes (the grouping of the genes is the same as presented in **Figures 4L-N**). We did not see any difference in the number of DRACH motifs in MYCN/METTL3 co-bound m⁶A positive genes versus the rest of the m⁶A genes (data is pasted below for the reviewer's perusal). This data suggests that the number of m⁶A motifs present in the MYCN/METTL3 co-bound genes might not be a critical driving factor for the m⁶A positivity of these genes.

Rebuttal Figure 2. Number of DRACH motifs per kb in m⁶A positive genes that were METTL3-MYCN co-bound or rest of the m⁶A containing genes in SHEP^{MYCN} cells (Dox+).

Section 2:

The reviewer also raised an interesting question: “It is also unclear why MYCN needs to recruit METTL3 to reduce *HOX* gene expression in undifferentiated cells. It seems that they should be repressed by default in these cells.”

We observed that *HOXC8/9* genes were enriched with MYCN and METTL3 at the ES cell stage, albeit at a lower level compared to tNCC. In ES cells, these genes are transcriptionally inactive, which is also visualized by the complete lack of the active chromatin mark H3K27ac.

We wanted to check if there are more regions in ES cells where METTL3/MYCN is present at the gene promoters, but they lack H3K27ac using the ChIP-seq data. We identified more such genes genome-wide with MYCN and METTL3 enrichment but lacking H3K27ac at the promoter in ES cells. The METTL3-MYCN co-bound genes without H3K27ac were associated with pathways related to neuronal projection and axonogenesis. This data suggests that MYCN and METTL3 pre-mark these genes in a transcription-independent manner (evident by the lack of H3K27ac and low RNA expression) and these genes are probably regulated in an m⁶A-dependent manner later during development. This data also indicates that in a transcription-independent manner, MYCN and METTL3 could be recruited to developmentally regulated genes. Further studies are needed to understand the mechanisms that drive such pre-marking of MYCN and METTL3 over developmental genes and how this pre-marking influences m⁶A levels at later developmental stages. We have provided this data for the reviewer’s consideration (**data related to this section is pasted below**).

Rebuttal Figure 3. (A) Number of genes that were METTL3-MYCN co-bound with/without H3K27ac in hESC. (B) Expression [average \log_2 (CPM)] of genes that were METTL3-MYCN co-bound with/without H3K27ac in hESC. Expression values were obtained from RNA-seq data. (C) Top enriched terms associated with genes that were METTL3-MYCN co-bound with/without H3K27ac in hESC.

5. The finding that inhibition of m6A deposition using the small inhibitor STM2457 is sufficient to enhance differentiation in MYCN over-expressing or MYCN-amplified neuroblastoma cells is interesting. The authors should provide some information with regard to toxicity of the inhibitor and the combination of STM2457 and doxorubicin in some tested normal populations of tNCC, SAP and SN.

Answer: In response to the reviewer’s suggestion, we treated tNCC with a combination of STM2457 and doxorubicin, but observed no effect on the growth of these cells. We tested the same dose of STM2457 and doxorubicin on the MYCN NB PDX cell line, which showed susceptibility to combination treatment (**Supplementary Figure S6I and S6J** in the revised manuscript, attached below).

Figure for reviewers removed;
see Appendix Figure S6 in the
manuscript.

Supplementary Figure S6 from the revised manuscript. (J) Bar plot shows the cell viability of tNCC treated with DMSO, STM2457, Doxorubicin, or a combination of STM2457 and Doxorubicin for 72 hours. Data are presented as mean \pm SD from three independent experiments. Unpaired t-test was used, ns- nonsignificant.

In the revised manuscript, we conducted *in vivo* experiments in mice and zebrafish xenograft models to evaluate the efficacy of the combination treatment. We used NSG mice with tumors resulting from subcutaneous injection of the MNA PDX cell line (COG-N-415). Xenografted mice were treated with either vehicle control, a combination of STM2457 and doxorubicin, or single drugs. Our findings indicate that STM2457, when combined with doxorubicin, significantly reduced tumor volume compared to single drug treatments (**Fig. 6H** in the revised manuscript, please see **page 27** of rebuttal). Importantly, none of the drugs or the combination had a significant effect on mouse body weight (**Fig. 6H**), suggesting that the combination treatment is well tolerated in this model.

We also observed these drug combinations were well tolerated in the zebrafish NB model. Overall, these data suggest that METTL3 inhibition may represent an effective therapeutic approach for treating NB.

Referee #2

(Report for Author)

The paper by Thombare and colleagues investigates the impact of the m6A modification on neuroblastoma, the most common extracranial childhood cancer. Focusing on the METTL3 methyltransferase m6 writer, the study suggests that m6A modification plays a crucial role in regulating the expression of HOX genes in neural crest cells, influencing their differentiation into sympathetic neurons. Furthermore, the authors suggest a potential therapeutic approach for neuroblastoma, particularly in cases with MYCN amplification/overexpression, as inhibiting/knocking down METTL3 induces differentiation, and may increase the vulnerability to chemotherapy. Finally, the authors suggest that MYCN and METTL3 physically interact to methylate MYCN targets and control HOX transcript stability via m6A. Overall, the findings have the potential to advance our understanding of the molecular mechanisms in neuroblastoma and the authors propose a possible avenue for therapeutic interventions. However, there are multiple shortcomings in this manuscript that need to be addressed. They are as follows:

We appreciate the reviewer's valuable suggestions and critical comments. In the revised version, we have made efforts to address these issues, aiming to enhance the quality of the data presented in the manuscript. We believe that we have successfully addressed the concerns raised by the reviewer.

1) The authors write: "Overall, higher expression of METTL3/14, differential gene expression along with m6A RIP- seq data in hESC and tNCC suggest a critical role for m6A in tNCC." However, based on the levels of the m6A writers and number of m6A peaks, one cannot suggest a critical role for m6A - please tone down this statement.

Answer: Considering the reviewer's suggestion we have re-written this sentence as follows

"Overall, higher expression of METTL3/14, differential gene expression along with m⁶A RIP- seq data in hESC and tNCC suggest that m⁶A may have a role in the tNCC differentiation."

2) Regarding Figure 1D - please show the P value for this comparison. Furthermore, while the authors suggest that the number of m6A peaks is higher in tNCC compared to hESCs, they do not clearly state whether this can be simply explained by a higher number of peaks in the same/overlapping transcripts or in fact some transcripts that are not m6A modified in hESC are

modified in tNCCs (or other way round) upon lineage commitment. Please present a Venn diagram for m⁶A-modified transcripts in hESC and tNCCs - I cannot find these data in the current manuscript.

Answer: In the revised version, we have incorporated the p-value for the indicated **Figure 1D** (m⁶A peaks tNCC vs hESC) as recommended by the reviewer.

Additionally, we have expanded our analysis to compare m⁶A positive genes in hESC vs tNCC. We observed not only an increase in the number of m⁶A peaks but also identified a different set of genes that showed a gain of m⁶A peaks during the differentiation from hESC to tNCC. This additional data is now presented in the revised version as **Figure 1G**.

3) The experiments presented throughout the manuscript are performed using only one shRNA. Given that key claims are being made based on METTL3 knockdown data, it would be important to perform the key experiments using another independent hairpin.

Answer: We concur with the reviewer's perspective, and we have made every effort to incorporate data from METTL3 KD using two independent shRNAs, particularly focusing on the METTL3 KD-mediated differentiation and cell growth phenotype, which are pivotal experiments. The newly generated data are now included in **Figures 1I, M, and Figure 2J** (depicting the differentiation phenotype following METTL3 KD), as well as **Supplementary Figure S2D** (illustrating the cell growth phenotype following METTL3 KD).

Although we previously had data from two shRNAs for some conditions, we did not present it in the earlier submitted version due to the lack of such data across multiple conditions. However, in the revised version, we have conducted additional experiments to provide this data, ensuring validation of the main claim using two independent hairpins.

Moreover, in the earlier submitted version, we included data from METTL3 inhibition by STM2457. We demonstrated that METTL3 inhibition leads to the differentiation of MYCN NB cells (**Figure 6D**), mirroring the METTL3 KD-mediated differentiation phenotype. The data from METTL3 inhibition further corroborate the findings from METTL3 KD experiments.

4) In Figure 2B, the authors suggest that HOXC8/9 are downregulated in tumours with MYCN amplifications. It would be important to test whether or not the HOXC8 and HOXC9 protein levels are also affected.

Answer: Using publicly available data (PMID: 33627664), we demonstrate a reduction in HOXC9 protein levels in MNA tumors compared to non-MNA tumors in the available public dataset (**Figure 2C** in the revised version, attached below). However, this dataset did not include information on HOXC8 expression, so we were unable to include it in our analysis.

Furthermore, to validate our findings, we utilized a validation cohort consisting of MYCN-amplified high-risk and MYCN non-amplified low-risk tumors. Through western blot analysis, we observed a down-regulation of HOXC9 expression in MYCN-amplified tumors compared to non-amplified NB tumors (**Figure 2C** in the revised version, attached below).

Figure for reviewers removed; see Figure 2 in the manuscript.

Figure 2 from the revised manuscript. (C) Left: Box-whisker plots show HOXC9 protein levels in non-MNA and MNA NB patients (R2 data set). Right: Immunoblot shows the levels of HOXC9 in NB patient samples. MYCN status, risk stratification, stage, and HOXC9 expression levels determined by RNA sequencing (expression score) are also provided. HSP90 was used as a loading control.

Despite our efforts, we encountered challenges in detecting HOXC8 expression using various antibodies in western blot analysis. Consequently, we refrained from including this data. However, at the RNA level, downregulation of both HOXC8 and HOXC9 expression was evident in MYCN-amplified NB tumors (**Figure 2B**).

5) In Figure 2 the authors go on to perform RNA-seq in SK-N-BE cells upon METTL3 knockdown as well as m6A-seq in SK-N-BE cells. However, they do not provide any global analyses of the data. It would be important to show graphs/plots depicting the expression change between METTL3 KD against control, for non-modified and m6A-modified transcripts. Is there a statistically significant upregulation of m6A-modified transcripts upon

METTL3 KD? Is there a difference in expression of non-modified transcripts upon METTL3 KD?

Answer: Taking into account the reviewer's suggestion, we have investigated the expression of m⁶A-positive and m⁶A-negative transcripts in METTL3 KD versus control in SK-N-BE2 cells. Our analysis revealed a significant up-regulation of m⁶A-positive transcripts following METTL3 KD, consistent with the established role of m⁶A modification in RNA stability regulation. We have included this data as **Supplementary Figure S2B** in the revised manuscript (attached below).

Figure for reviewers removed; see Appendix Figure S2 in the manuscript.

*Supplementary Figure S2 from the revised manuscript. (B) Box plot shows fold change in RNA expression after METTL3 KD (TetO shCtrl vs. TetO shM3-1) in SK-N-BE(2) cells. Genes are categorized based on the presence of m⁶A into m⁶A+ or genes lacking m⁶A (m⁶A-). Wilcoxon test was used (**** $p < 0.0001$).*

6) Is global m⁶A-transcript stability and translational efficiency affected upon METTL3 depletion in SK-N-BE cells? It would be important to perform SLAM-seq and RIBO-seq experiments to address this.

Answer: We appreciate the reviewer's suggestion to conduct global analyses to examine RNA stability and translation efficiency following METTL3 KD, utilizing techniques such as SLAM-seq and RIBO-seq. Indeed, numerous studies, as referenced (PMID: 37202476, PMID: 27558897, PMID: 31964509), have elucidated the global functions of METTL3-mediated m⁶A modification in RNA stability regulation.

Consistent with these findings, our study revealed a significant up-regulation of m⁶A positive transcripts following METTL3 KD, highlighting the role of m⁶A modification in RNA stability regulation (**Supplementary Figure S2B** in the revised manuscript). However, given our focus on *HOXC* transcripts, we opted not to conduct genome-wide experiments, partly due to resource constraints and also because prior studies have extensively demonstrated m⁶A-dependent regulation of RNA stability on a global scale.

In our study, we focused on investigating the stability of *HOXC8/9* transcripts post METTL3 KD, as evidenced by our earlier findings from 3 days post METTL3 KD (**Figure 2H** in the

revised manuscript, attached below). In the revised version, we have extended this analysis to include stability data after 6 days of METTL3 KD as well, yielding consistent results with those obtained after 3 days of METTL3 KD (**Figure 2H**). Furthermore, we demonstrated that METTL3 inhibition with STM2457 also resulted in an increased half-life of *HOXC8/9* transcripts, corroborating our METTL3 KD findings (**Supplementary Figure 6D** in the revised manuscript, attached below).

Figure for reviewers removed; see Figure 2 in the manuscript.

Figure 2 from the revised manuscript. (H) Stability of *HOXC8* and *HOXC9* transcripts detected by RT-qPCR following Actinomycin D (10 µg/ml) mediated transcription blocking for the time points indicated in control (*TetO shCtrl*) and METTL3 KD (*TetO shM3-1*) SK-N-BE(2). Assay was conducted following 3 and 6 days of doxycycline (Dox) addition. Line plots present the quantification of remaining levels of *HOXC8* and *HOXC9* transcript at the indicated time points. Half-life ($t_{1/2}$) values are also denoted. Data are presented as mean ± SEM. Two-way ANOVA with Šídák's multiple comparisons test was employed (* $p < 0.05$, ** $p < 0.01$, *** $p < 0.001$, **** $p < 0.0001$)

Supplementary Figure S6 from the revised manuscript. (D) Stability of *HOXC8* and *HOXC9* transcripts detected by RT-qPCR after Actinomycin D (10 µg/ml) mediated transcription blocking for the time points indicated in SK-N-BE(2) cells treated with either DMSO or STM2457. Line plots presenting the quantification of remaining levels of *HOXC8* and *HOXC9* transcript at the indicated time points ($n=3$). $t_{1/2}$ values are also denoted. Two-way ANOVA with Šídák's multiple comparisons test was employed and data are shown as mean ± SEM of three replicates. (* $p < 0.05$, ** $p < 0.01$, **** $p < 0.0001$).

Figure for reviewers removed; see Supplementary Figure S6 in the manuscript.

Considering the reviewer's suggestion, we also sought to determine if METTL3 KD affects the translation efficiency of *HOXC8/9* transcripts. However, polysome profiling in control and METTL3 KD cells did not reveal any change in the polysome association of *HOXC8* and *HOXC9* transcripts, indicating that m⁶A modification in these transcripts may have little role in translation regulation (**Supplementary Figure S2C** in the revised manuscript, attached below). Consequently, as our key genes were not regulated in a translation-dependent manner

following METTL3 KD, we refrained from conducting further global analyses such as RIBO-seq.

Figure for reviewers removed; see Appendix Figure S2 in the manuscript.

Supplementary Figure S2 from the revised manuscript. (C) Left: Polysome profile in Control and METTL3 KD SK-N-BE(2) cells. Right: Enrichment (% Input) of HOXC8, HOXC9, and GAPDH transcripts present in the polysome fraction of 72 h Dox induced

control and METTL3 KD SK-N-BE(2) cells. Data are presented as mean \pm SD from two independent experiments. Statistical analysis was performed using two-way ANOVA with Dunnett's multiple comparisons test (ns-nonsignificant).

We hope that the reviewer will appreciate our rationale for not performing additional global analyses such as SLAM-seq and RIBO-seq within the scope of our study, which primarily focuses on elucidating m⁶A-dependent regulation of *HOX* transcripts and the cross-talk between MYCN and METTL3 in regulating m⁶A modification of *HOXC* transcripts.

7) The authors state: "In particular, we observed the increased stability of HOXC8 and HOXC9 mRNA following METTL3 KD in SK-N-BE(2) cells (Fig. 2G)". The stability of HOXC9 is very marginally affected and the conclusions need to be toned down.

Answer: Considering the reviewer's suggestion we have toned down the conclusion and rewritten it as "We observed a moderate increase in the stability of *HOXC8* and *HOXC9* mRNA following induction of METTL3 KD by Dox addition for both 3 and 6 days in SK-N-BE(2) cells (**Fig. 2H**)."

We appreciate the reviewer for bringing this to our attention. In response to the reviewer's suggestion, we have thoroughly re-evaluated the data and conducted new experiments. Initially, our stability data for HOXC8 and HOXC9 transcripts following METTL3 KD was limited to 3 days (as presented in **Figure 2H** of the revised manuscript). While the data itself remains unchanged, we have adjusted the scale on the Y-axis to enhance the visibility of the difference between control and METTL3 KD-induced changes in stability. Moreover, we have now calculated the half-life for these transcripts, a detail that was lacking in the earlier version. Notably, the calculated half-life values, clearly depicted in the plot, indicate a higher stability following METTL3 KD.

Furthermore, during the revision process, we conducted an RNA stability experiment following 6 days of METTL3 KD. Consistently, this new data reaffirms the increased stability of *HOXC8* and *HOXC9* transcripts. We have also calculated the half-life for the day 6 data and integrated it into the plot, as presented in **Figure 2H**.

8) It is suggested that upregulation of HOXC8 and HOXC9 may mediate differentiation of SK-N-BE cells upon METTL3 knockdown. Can knockdown of HOXC8 and/or HOXC9 rescue the phenotype resulting from METTL3 knockdown?

Answer: We appreciate the reviewer for highlighting this crucial aspect. In response to this suggestion, we have conducted knockdown (KD) experiments targeting HOXC9 in METTL3-depleted cells to investigate whether the differentiation phenotype mediated by METTL3 depletion could be reversed.

We performed HOXC9 KD in METTL3-depleted cells. Our results indicate that METTL3 depletion leads to an upregulation of HOXC9 protein levels, and this effect can be abolished by HOXC9 shRNA. Importantly, knocking down HOXC9 in METTL3-depleted cells effectively prevented the differentiation of these cells. These findings strongly suggest that the increase in HOXC9 expression contributes to the differentiation phenotype observed in MYCN NB cells upon METTL3 depletion. We have incorporated this new data in **Figure 5E-F** in the revised manuscript (please see attached below).

Figure for reviewers removed; see Figure 5 in the manuscript.

Figure 5 from the revised manuscript. (E) Representative immunoblot showing expression of HOXC9 in Control (TetO shCtrl) and METTL3 KD (TetO shM3-1, TetO shM3-2) SK-N-BE(2) cells, along with shRNA mediated KD of HOXC9 (shHOXC9). Dox induction was performed for 72 h. Vinculin was loading control. The values below indicate the fold change in levels of HOXC9. The experiments were repeated three times. (F) IF showing expression of PRPH (green) and TUBB3 (red) in TetO shCtrl, TetO shM3-1, and TetO shM3-2 SK-N-BE(2) cells in similar condition as described in (E), except Dox was added for 24 h after HOXC9 shRNA transduction followed by 3 days RA mediated differentiation in the presence of Dox. Box-whisker plot shows the quantification of the neurite length. Experiments were performed in three independent biological replicates. Two-way ANOVA with Šidák's multiple comparisons test was used (* $p < 0.05$, **** $p < 0.0001$). Scale bar represents 50 μm .

9) The data shown in Figure 3F-G appear to have huge error bars. How can these differences be statistically significant? Please explain how this was achieved? The same comment pertains to results shown in Figure 5E.

Answer: We acknowledge the reviewer's observation regarding the insufficient clarity of the IF data presented for HOXC8/9 expression in the earlier version of the manuscript. Upon reflection, we recognized that not normalizing the HOXC8/9 intensity with DAPI contributed to variability in the data. Additionally, the large error bars in the previous bar plots were influenced by the variability in expression between individual cells, as the data was obtained from a significant number of cells (>1000).

To address these concerns, we have made significant improvements in data presentation in the revised version. Specifically, we have normalized the HOXC8/9 intensity with DAPI and presented the data as box-whisker plots, allowing for a clearer visualization of the range of data points obtained from a large number of cells. This adjustment has been applied consistently throughout the manuscript whenever similar comparisons were made.

We believe that these enhancements in data presentation will provide the reviewer with more convincing evidence, particularly regarding the statistically significant differences. Details regarding the statistical methods employed are indicated in the figure legends. Furthermore, our validation efforts, including the confirmation of HOXC9 downregulation following MYCN overexpression and upregulation of HOXC9 protein levels in METTL3 KD conditions, are consistent with the IF data. This validation was achieved through supplementary experiments, as depicted in **Supplementary Figure S3A** and **S10** (please see **page 10** of rebuttal).

Moreover, downregulation of both *HOXC8* and *HOXC9* at the RNA level has been confirmed using RNA-seq and RT-qPCR in MYCN overexpressing SAP cells, as shown in **Figure 3H** of the revised manuscript (please see **page 7** of rebuttal).

Similar improvements in data presentation have been implemented for data related to Figure 5E (revised manuscript **Figure 6E**), ensuring consistency in the presentation of results throughout the manuscript.

10) The data presented in Figure 4 suggest that MYCN and METTL3 physically interact- please validate this interaction by performing immunoprecipitation experiments. Can MYCN pull down METTL3, and vice versa?

Answer: We appreciate the reviewer's suggestion regarding this experiment.

In response to the reviewer's recommendation, we have conducted a METTL3 and MYCN Co-Immunoprecipitation experiment, successfully validating the interaction between MYCN and METTL3 using this approach. We have integrated this new data into **Figure 4A** of the revised manuscript (attached below).

Figure for reviewers removed; see Figure 4 in the manuscript.

Figure 4 from the revised manuscript. (A) Co-IP of METTL3 or MYCN from lysates of Flag-MYCN overexpressing SHEP cells (SHEP^{MYCN}) after Dox induction for 24 h, blotted with MYCN or METTL3 antibodies. IgG served as a negative control.

11) They conclude that: "...METTL3 inhibitors may represent efficacious therapeutic agents in the treatment of NB". However, in Figure 5 the authors use unusually high concentration of METTL3 inhibitor (i.e. 10 microM). Why? It is much higher compared to the concentration that compromises AML cells (Nature 593, 597-601; 2021).

Answer: We explored a range of concentrations for STM2457 and determined that the effective concentration for treating NB cells in vitro was higher than what was previously reported for AML cells. After initial observations, we proceeded with a concentration of 10 microM STM2457 for our experimental setup. Interestingly, as we prepared our initial draft and subsequent revisions, several publications emerged discussing the therapeutic use of STM2457

for various cancers such as ovarian cancer and breast cancer. These studies utilized STM2457 concentrations in the range of 10-20 microM, consistent with our chosen concentration.

In the revised manuscript, we have incorporated data from *in vivo* treatment of a mouse NB PDX model (**Figure 6H** in the revised manuscript, please see attached below). It's important to note that for *in vivo* mouse PDX treatment experiments with STM2457 alone or in combination with doxorubicin, we utilized a STM2457 dosing of 50 mg/kg, consistent with the dosing reported in AML studies.

While discussing the potential for more effective METTL3 inhibitors, it's worth mentioning a recent report suggesting the development of a novel and more potent METTL3 inhibitor, STM3006. However, despite its efficacy *in vitro* at concentrations ten times lower than STM2457, STM3006 faces challenges related to bioavailability, rendering it unsuitable for *in vivo* studies. Therefore, the pursuit of a more potent METTL3 inhibitor with improved *in vivo* pharmacokinetic properties remains a promising avenue for future research.

12) The authors suggest that METTL3 inhibition combined with chemotherapy may be a therapeutic strategy in neuroblastoma. However, METTL3 inactivation alone causes hematopoietic stem cell failure and doxorubicin is toxic too. If there is a synergistic interaction between METTL3 inhibition and doxorubicin, as the authors suggest, then such treatment is highly likely to have deleterious consequences to normal tissues/organs. What is the therapeutic window for such treatment? It would be important to test this in normal (non-cancer) counterparts of neuroblastoma cells. It would also be important to perform a pilot study with METTL3 inhibition + doxorubicin to investigate whether this impacts on mouse health and survival.

Answer: We deeply appreciate the reviewer for highlighting these critical issues, which prompted us to conduct a new round of experiments. We have incorporated the findings into the revised manuscript as outlined below.

Firstly, we evaluated the combination of STM2457 and Doxorubicin in tNCC, serving as the normal cell counterpart of NB cell lines (**Supplementary Figure S6J** in the revised manuscript, **page 16** of rebuttal). The combination dose mirrored that used for *in vitro* treatment of the MYCN NB PDX cell lines (**Supplementary Figure 6I**). Encouragingly, we found that the combination had no adverse effects on the growth of tNCC.

A recent study examining the effects of STM2457 treatment on normal hematopoiesis (PMID: 37464070) concluded that pharmacological METTL3 inhibition yielded milder, more manageable effects compared to genetic METTL3 knockout studies. Notably, earlier AML studies with STM2457 (Nature 593, 597-601; 2021) reported transient and rapidly reversible hematopoietic effects, with no observed long-term toxicity or impacts on normal blood counts in mice after pharmacological METTL3 inhibition using STM2457.

In response to the reviewer's suggestion, we conducted an *in vivo* experiment using a mouse MYCN NB PDX model to assess the therapeutic window for STM2457 and combination therapy. Detailed drug dosing information is provided in the revised manuscript, with STM2457 dosing consistent with previous reports (AML studies, Nature 593, 597-601; 2021). Our data indicate that the combination of STM2457 and doxorubicin was more effective in reducing tumor size in MYCN NB PDX compared to either drug alone. Furthermore, we monitored mouse body weight during the treatment period and observed no significant differences between vehicle and treatment groups, suggesting that the drug combination is well tolerated in the mouse model. These new findings are included in **Figure 6H** and attached below

Figure for reviewers removed; see Figure 6 in the manuscript.

Figure 6 from the revised manuscript. (H) *Left: Cartoon demonstrating the experimental strategy used for the mouse in vivo experiment performed with patient-derived xenograft (PDX) cells. MNA COG-N-415x, PDX cells were injected into NSG mice. Once tumors reached 170 mm³ mice were allocated into four treatment groups (n = 4-6 per group) and treated for 14 days with vehicle (20% hydroxypropyl-beta cyclodextrin) daily, STM2457 (50 mg/kg in vehicle) daily, doxorubicin (0.2 mg/kg in vehicle) every three days or a combination of STM2457 and doxorubicin at the same doses. Line plots show tumor volume (middle) and body weight (right) in the treatment groups. Data are presented as mean ± SEM. Statistical analysis was conducted using a two-way ANOVA with Tukey's post hoc test (***) $p < 0.001$, **** $p < 0.0001$, ns- non-significant).*

Additionally, we found that these drug combinations were well tolerated in a zebrafish NB model, with combination treatment demonstrating an overall better treatment response (**Supplementary Figure S6K**) and attached below.

Figure for reviewers removed; see Figure Appendix Figure S6 in the manuscript.

Supplementary Figure S6 from the revised manuscript. (K) SK-N-BE(2)-GFP cells labeled with DiI (red dye) were injected into the perivitelline space of wild type zebrafish, 48 h post fertilization. They received treatments with either solvent (0.2% DMSO), STM2457 100 μ M, doxorubicin 5 μ M, or a combination of both with the same doses for 72 h. Images were taken at 0 h and 72 h after treatment and were analyzed through Image J measuring tumor area using macro with the same settings on every image. Waterfall plots show percent change in tumor size and scatter dot plot shows tumor growth as fold change (72 h/ 0 h). Data are

presented as mean \pm SEM. Statistical analysis was performed using one-way ANOVA with Dunnett's multiple comparisons test (* $p < 0.05$).

In conclusion, through the implementation of new experiments, the incorporation of additional data, and the meticulous addressing of concerns surrounding treatment efficacy and safety, we have strengthened the quality of our findings. These efforts support the case for considering METTL3 inhibition as a potential therapeutic strategy for neuroblastoma.

Referee #3

(Report for Author)

Overall this manuscript is well written, presents novel mechanistic data and in my opinion is suitable for publication in EMBO.

We are pleased to learn that the reviewer found our manuscript suitable for publication in EMBO. We extend our gratitude to the reviewer for their valuable suggestions and comments. In response, we have diligently incorporated and addressed these suggestions in the revised manuscript, as outlined below.

Minor comments:

Suppl 1J-Q - the role of metal KD in hESC & NC - this feels like important data that is very relevant and should not be buried in the supplementary?

Answer: We appreciate the reviewer for bringing this to our attention. In accordance with the reviewer's suggestion, we have relocated significant data, such as **Figures 1I, J, and L**, from the supplementary figures to the main Figure 1 in the revised manuscript.

Suppl Fig2C - did METTL3 KD in control cell line (step without MYCN) have any effect on viability? This data should also be shown

Answer: We have incorporated data from METTL3 knockdown (KD) in SHEP cells without MYCN induction as **Supplementary Figure 2F**. Interestingly, we observed that METTL3 KD alone, without MYCN induction, did not have any effect on cell growth.

Fig 4C/supplementary figure 4 - pathway analysis of trunk NC genes that are co-bound by MYCN and METTL3 is very limited. Could this be expanded on?

Answer: Following the reviewer's suggestion, we have integrated this analysis into the revised manuscript as a pathway analysis on METTL3 and MYCN co-bound genes in tNCC. Our findings reveal that these genes are enriched with biological processes related to neuronal projection development and axonogenesis. Detailed data can be found in **Supplementary Figure S4F**.

4G/H - changes in m⁶A enrichment and RNA expression are shown in shep MYCN dox cells but not shep controls. Do you see the same pattern in non MYCN expressing shep?

Answer: We investigated m⁶A enrichment over the *HOXC8* and *HOXC9* transcripts in both MYCN-induced and non-induced cells using m⁶A RIP-qPCR. Our findings revealed higher m⁶A enrichment over these transcripts in MYCN-induced cells, as shown in **Figure 4I**. This data was included in the earlier submitted version.

In response to the reviewer's suggestion, in the revised version, we conducted m⁶A RIP-seq, which again demonstrated a consistent pattern of higher m⁶A enrichment following MYCN overexpression over the *HOXC8* and *HOXC9* genes (**Figure 4I**, please see **page 4** of rebuttal).

Furthermore, we examined the level of *HOXC8* and *HOXC9* transcripts in METTL3-depleted cells without MYCN expression using qRT-PCR. Surprisingly, the transcript levels of *HOXC8* and *HOXC9* remained unchanged. However, METTL3 KD in combination with MYCN induction led to increased expression of *HOXC8* and *HOXC9* transcripts, as depicted in **Figure 4J** (please see **page 13** of rebuttal).

Dear Dr Mondal,

Thank you for submitting your revised manuscript (EMBOJ-2023-115931R1-Q) to The EMBO Journal, as well as for your patience with our response at this time of the year. Your amended study was sent back to the referees for their re-evaluation, and we have received recommendations from two of them, which I enclose below. As you will see, the experts stated that the work has been substantially improved by the revisions and they are now in favour of publication. We have in addition checked your response to reviewer #1 editorially and found it to be satisfactory.

Thus, we are pleased to inform you that your manuscript has been accepted in principle for publication in The EMBO Journal.

We now need you to take care of a number of minor issues related to formatting and data presentation as detailed below, which should be addressed at re-submission.

Please contact me at any time if you have additional questions related to below points.

As you might have seen on our web page, every paper at the EMBO Journal now includes a 'Synopsis', displayed on the html and freely accessible to all readers. The synopsis includes a 'model' figure as well as 2-5 one-short-sentence bullet points that summarize the article. I would appreciate if you could provide this figure and the bullet points.

Thank you for giving us the chance to consider your manuscript for The EMBO Journal. I look forward to your final revision.

Again, please contact me at any time if you need any help or have further questions.

Kind regards,

Daniel Klimmeck

>> Please provide up to five keywords for your study.

>> Add a completed Author Checklist.

>> Please provide the main manuscript text as .docx file.

>> Author Contributions: add information on author contributions. Note that CRediT has replaced the traditional author contributions section as of now because it offers a systematic machine-readable author contributions format that allows for more effective research assessment. and use the free text boxes beneath each contributing author's name to add specific details on the author's contribution.

More information is available in our guide to authors.
<https://www.embopress.org/page/journal/14602075/authorguide>

>> Rename the current 'Competing Interests' section to 'Disclosure and Competing Interests Statement'.

>> Figures: the main figures should be uploaded as individual, high resolution figure files. The figure legends should be removed from the main figures and added to the manuscript text.

>> Section order should be corrected as follows: title page with complete author information, abstract, keywords, introduction, results, discussion, methods, data availability section, acknowledgements, disclosure and competing interests statement, references, main figure legends, tables, expanded figure legends.

>> Reagents and Tools Table: needs to be uploaded in the correct format using the template provided on our webpage. Rename the excel file "Dataset EV1", and to add a legend to the file.

>> Funding: information on funding needs to be entered as part of the Acknowledgments section; and in addition completed in our online system.

>> Appendix file: the supplemental figures should be renamed "Appendix Figure S1" etc. The supplemental methods should remain in the appendix file as "Appendix Supplementary Methods", given that they contain a number of schemes. The appendix will need a table of contents including page numbers on its first page.

>> Please provide source data for the study as to the separate request e-mail by my colleague Hannah Sonntag.

>> Data availability section: deposit the sequencing data in a public data repository and make them publicly accessible, annotating dataset identifiers and hyperlinks.

>> Recheck publication status of the BioRxiv citations Wenninger et al (2022), Vaid et al (2022), and adjust in case of formal journal publication.

>> BioRender: Add a separate "Graphics" section at the end of the Methods using this format: Graphics: (some of the... OR Figure #... OR synopsis) Graphics were created with BioRender.com.

>> Consider additional changes and comments from our production team as indicated below:

- Data Availability Section:

1. Please note that the specific URL for GEO dataset GSE244473 is not provided in the data availability statement.
2. Please note that reviewer access code for "GSE244473" dataset is not provided in the data availability statement.

- Figure legends:

1. Please note that the legends for figures 1-7, Supplementary figure(s) 1-6 do not contain a common figure title. This needs to be rectified.
2. Please note that the exact p values are not provided in the legends of figures 1D, L, M; 2B, H, I, J; 3D-I; 4I, J, N, O; 5B, C, D, F; 6A-H; Supplementary figure(s) 1A, E, G, L-O; 2B, D-E; 3B; 4A-B; 5D, H-I; 6A-E, H, K
3. Please indicate the statistical test used for data analysis in the legends of figures 1H, J; 2E; 5A, Supplementary figure(s) 1B, F, I; 2A, I; 3D-E; 4F; 5B, E; 6F-I
4. Please note that for the figures 2C, Supplementary figure(s) 2D, 5B """" has not been defined in the figure legends. Please rectify this in the legends as applicable. Please also indicate the statistical test used for data analysis and the exact p-values for the same in the legend.
5. Please note that the box plots need to be defined in terms of minima, maxima, centre, bounds of box and whiskers, and percentile in the legends of figures 2C, I, J; 3D-G; 4N, O; 5B, C, D, F; 6A-G; Supplementary figure(s) 1M-O; 2B; 4A-B; 5H; 6A-B
6. Please note that information related to n is missing in the legends of figures 2H; 4N; Supplementary figure(s) 2B; 3B; 5B; 6K
7. Please note that n=2 in Supplementary figure(s) 2C
8. Although 'n' is provided, please describe the nature of entity for 'n' in the legends of figures 2B, C; 3I; 4I; Supplementary figure(s) 5D, Supplementary figure(s) 3B, 5B
9. Please note that the error bars are not defined in the legends of figures 2I; 3I; 4I; 6G.
10. Please note that the scale bar needs to be defined for figures 5A
11. Please note that scale bar and its definition are missing for figures 6L.

Referee #2:

I am quite happy with the way the authors addressed my comments, and I have no further criticisms to make.

Referee #3:

Overall the experimental data in the revised version of this manuscript significantly adds to the authors finding of a novel association between MYCN and METTL3 in the regulation of neural crest and neuroblastoma differentiation.

The authors have provided comprehensive answers based on additional experimental data to all of the concerns raised by the reviewers.

While the focus of this paper is on the more mechanistic side, these findings are also of potential translation relevance in an area of unmet need.

I would recommend this paper for publication

The authors addressed the remaining editorial issues.

Dear Dr Mondal,

Thank you for submitting the revised version of your manuscript. I have now evaluated your amended manuscript and concluded that the remaining minor concerns have been sufficiently addressed.

I am thus pleased to inform you that your manuscript has been accepted for publication in the EMBO Journal.

Related I would like to hereby ask your consent on keeping the referee figures included in this file.

On a different note, I would like to alert you that EMBO Press offers a format for a video-synopsis of work published with us, which essentially is a short, author-generated film explaining the core findings in hand drawings, and, as we believe, can be very useful to increase visibility of the work. Please see the following link for representative examples and their integration into the article web page:

<https://www.embopress.org/doi/full/10.15252/embojournal.2019103932>

Best regards,

Daniel Klimmeck

Daniel Klimmeck, PhD
Senior Editor
The EMBO Journal
EMBO
Postfach 1022-40
Meyershofstrasse 1
D-69117 Heidelberg
contact@embojournal.org
Submit at: <http://emboj.msubmit.net>